

# A Statistical Analysis of TIR Anomalies extracted by RST in Relation

# with Earthquake in Sichuan Area with Use of MODIS LST Data

*Ying Zhang[1,2], Qingyan Meng[1]*

1. Institute of Remote Sensing and Digital Earth, Chinese Academy of Sciences, Beijing, 100101, China

2.  University of Chinese Academy of Sciences, Beijing, 100101. China

**Abstract:** There is a long history for research of earthquake prediction, but weakness of traditional approaches to study seismic hazard have been more and more evident. Remote sensing and earth observation technology, which is a new method that can instantly acquire a large area of abnormal information caused by earthquakes, is believed to be the key to the breakthrough of the bottleneck in the study of earthquake prediction. A multi-parametric approach seems, instead, to be the most promising approach in order to increase reliability and precision of short-term seismic hazard forecast, and Thermal Infrared (TIR) anomaly is an important part of the earthquake precursors. Though many scientists have studied the correlation among TIR anomalies identified by the Robust Satellite Techniques (RST) methodology and single earthquake, there is few study to extract the TIR anomalies in long period and large study area. Moreover, a statistical analysis of TIR anomalies in relation with earthquake is needed to determine whether there is the existence of TIR anomalies before earthquake. In this paper, a refined RST data analysis and Robust Estimator of TIR Anomalies (RETIRA) index were used to extract the TIR anomalies from 2002 to 2018 in Sichuan area with use of Moderate-resolution Imaging Spectro-radiometer (MODIS) Land Surface Temperature (LST), and the earthquake catalog were also used to study the correlation between TIR anomalies and occurrences of earthquake. Most of the thermal infrared anomalies correspond to earthquakes, and statistical methods are used to prove that there is a correlation between the extracted thermal infrared anomalies and earthquakes. And this is the first time to evaluate earthquakes prediction ability with use of PPV, FDR, TPR and FNR, the statistical result shows that the prediction ability of RST in Sichuan area is limited.

Key Words: Thermal Infrared Anomalies, Land Surface Temperature, MODIS, Earthquake

## 1. Introduction

There are numerous observations of surface temperature changes prior to Earth's crust earthquakes(Tronin, Hayakawa et al., 2002). Nowadays, Thermal Infrared Remote Sensing has taken to be a new method for seismic precursors detecting. Anomalous thermal infrared emissions have been widely detected by satellite sensors before the major earthquakes(Piroddi, Ranieri et al., 2014). Several studies discovered space-time anomalies in TIR and outgoing longwave satellite imagery, from weeks to days, before and after earthquakes (Wang, 1984, Gorny, Salman et al., 1988, Qiang, Xu et al., 1991, Tronin, 1996, Tramutoli, Bello et al., 2001, Ouzounov and Freund, 2004, Tramutoli, Corrado et al., 2015). Identification of thermal infrared (TIR) precursors as pre-seismic signal has gained support over world, especially in Russia, China, India, United States, Italy, and Saraf et al. observed such short-term anomalies around the epicentral region for earthquakes in India, Algeria, Iran, China, Pakistan and Indonesia through NOAA-AVHRR, Terra/Aqua-MODIS and



passive microwave DMSP-SSM/I satellite data and call them 'transient TIR anomalies (Saraf, Rawat et al., 2009).

There is few data analysis techniques that can isolate residual TIR variations, potentially associated with earthquake occurrence, from the normal variability of TIR signal due to other causes(Tramutoli, Cuomo et al., 2005). But, more than 10 years (since 2001) of application of the general Robust Satellite Techniques (Tramutoli, 1998, Valerio, 2005, Tramutoli, 2007) methodology to this issue, have shown the ability of this approach to discriminate anomalous TIR signals possibly associated with seismic activity from normal fluctuations of Earth's thermal emission related to other causes(e.g., meteorological) independent of the earthquake occurrences(Eleftheriou, Filizzola et al., 2016). RST is based on the Robust AVHRR techniques (RAT), which is proposed for environmental monitoring with the use of NOAA/AVHRR observations(Tramutoli, 1998). And since then most of the announced RAT applications have demonstrated their reliability as well their exportability on different satellite sensors and geographic areas, so RAT evolved into RST(Tramutoli, 2007). RST contains two main steps: the first RST requirement is the characterization of behavior in normal conditions; the second step is the establishment of change detection criteria which should be specified for each considered phenomenon class and chosen technology as well as for the time and place of the observation(Tramutoli, 2007).

With use of RST, many researchers have extracted TIR anomalies (in the following all the TIR anomalies refer to the thermal infrared anomalies extracted by RST) in different earthquakes, and have analyzed the space-time distribution of TIR anomalies. Using MODIS LST data, Pergola studied the 6 April 2009 Abruzzo earthquake found that spatially extended and time persistent TIR anomalies (with RETIRA>3) appear in some space-time correlation with earthquakes of different magnitude occurred in Italy in the considered period (15 March- 15 April) and since seven days before the Abruzzo main shock(Pergola, Aliano et al., 2010), while Mebrouk studied 21 May 2003 Boumerdes earthquake and detected a thermal anomaly persisting for a week during the month preceding the earthquake(Bellaoui, Hassini et al., 2017). Many researchers also used data of other satellites, Aliano used 8 years of Meteosat TIR observations to analyze 21 May 2003 Boumerdes/Thenia(Algeria) earthquake found that the area of interest was affected by significant positive thermal anomalies (S/N>2.5-3) about one month before the main shock(Aliano, Corrado et al., 2007), and M.Lisi studied 6 April 2009 Abruzzo earthquake with the use of NOAA/AVHRR TIR observations and TIR anomalies were identified in some space-time correlation with Abruzzo earthquake epicenter between 30 March and 1 April(Lisi, Filizzola et al., 2010). Genzano have studied the 2009 Abruzzo with use of different satellite data(5 years of MSG/SEVIRI, 15years of NOAA/AVHRR and 8 years of EOS/MODIS), t no similar results have been observed in confutation(Genzano, Corrado et al., 2010). Besides the analysis for TIR anomalies in single earthquake, Tramutoli et, al. have studied the causes of TIR anomalies, they performed a test over an area affected by variable gas emissions to study the correlation between TIR anomalies and seismicity, and found that the general gas dispersion models and the spatial features follow the hypothesis of a strict relation between greenhouse gas releases and TIR anomalies related to seismic activity(Tramutoli, Aliano et al., 2013).

Nowadays, some researchers have done a long-term statistical analysis to determine the correspondence between TIR anomalies and earthquakes. Genzano used data of GMS-5/VISSR TIR measurements to study earthquakes with M>4 occurred in a wide area around Taiwan, in the month of September from 1995 to 2002, and the false positive rate remained zero when the earthquakes

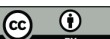



with M>4 or M>4.5 were considered, the false positive rate remained less than 6% when the M >5 is applied(Genzano, Filizzola et al., 2015). Tramutoli et al. studied the earthquakes with M>4.0 in Italy from July 2012 to June 2013 and the testing area was Italian southern Apennines, Po Plain, they found that the false positive rate was lesser than 33% while the missing rate is up to 67%(Tramutoli, Corrado et al., 2015). Eleftheriou et al. studied the earthquakes occurred in Greece in period 2004-2013 with use of TIR images acquired by MSG/SEVIRI, more than 93% of all identified TIR anomalies occurred in the prefixed space-time window around time and location of occurrence of earthquakes (with M>4) and the overall false positive rate is <7%(Eleftheriou, Filizzola et al., 2016). It seems that RST is an effect method to extract TIR anomalies before earthquakes, but there is no such study for the mainland of China

However, some researchers have proved that some single earthquake results are unreliable. Some so-called TIR anomalies are caused by weather anomalies, which are not related to earthquakes. An instance is that Matthew et al. studied the Gujarat (India) earthquake of 2001, and he found that the previous study, which indicated there was TIR anomalies before the earthquake, was not reliable. They concluded that there was no robust evidence for the existence of LST anomalies prior to the 2001 Gujarat earthquake and cloud covering was one possible cause for the anomalies(Blackett, Wooster et al., 2011). So, a rigorous rule of thermal anomaly judgement and long period statistical analysis are necessary.

In this paper, RST will be applied for the area ($27°N - 37°N, 97°E - 107°E$) to study a mountainous area of China. The long-term analysis (from Sep. 2002 to Mar. 2018) will conform the correspondence between TIR anomalies and earthquakes. Based on the statistical results, the earthquake prediction ability of RST will be evaluated in this paper.

2. Study area

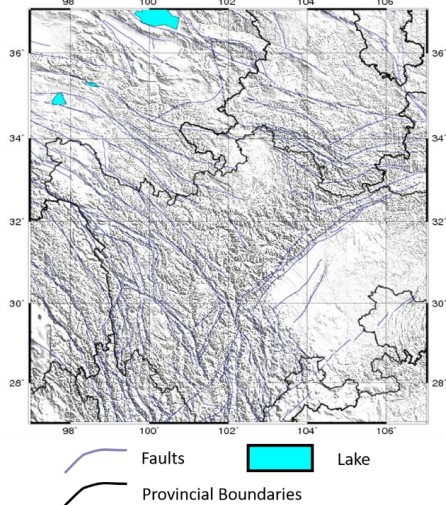

Fig.1 The distribution of faults in East-Southern Gansu province and its neighbor regions.

East-southern Gansu province and its neighbor regions are selected to study the correlation of TIR anomalies and earthquakes from Sep. 2002 to Mar. 2018. As it is shown in Fig.1, the scope of



the study area is $27^{\circ}N \ to \ 37^{\circ}N, \ 97^{\circ}E \ to \ 107^{\circ}E$. The region is located at the junction of Gansu, Qinghai and Sichuan, it is also the intersection of the North of North-South seismic belt and Kuma seismic belt, moreover, structures in this area are complex and strong earthquakes are frequent(Yang, Zhang et al., 2002). The area is on the eastern edge of Tibet Plateau, belonging to the upper part of

5    the rhombic block in the eastern-southern Gansu province. The Xian River fault, Longmen Shan Fault and Anning River fault are connected here, and the shape of the structure is like a 'Y'. This kind of geomorphology contains abundant plate tectonics, and the Longmen Shan fault in the direction of NNE becomes the steep slope of the Southeast Sichuan Basin and the erosion plateau in the northwest of the study area.

3.  Data and method

3.1  Data introduction

MODIS data is used to calculate the TIR anomalies, and the earthquake information got from the China Seismic Information (http://www.csi.ac.cn/) is used for statistical study in this paper.

The MODIS instrument is operating on both Terra and Aqua Spacecrafts. It has a viewing swath width of 2330 km and views the entire surface of the Earth every one to two days. Its detectors measure 36 spectral bands between 0.405 and 14.385 μm, and it acquires data at three spatial resolutions – 250m, 500m, 1000m. In this study, nighttime MODIS Land Surface Temperature daily data (MYD11C1) is used for the extraction of TIR anomalies. Because LSTs data are susceptible to

solar radiation during the daytime, the nighttime data is selected. The LSTs data is retrieved at 5600m by the generalized split-window algorithm. In the day/night algorithm, daytime and nighttime LSTs are retrieved from pairs of day and night MODIS observations in seven TIR bands. Moreover, the daily nighttime Cloud Mask data (MYD35L2) is used for excluding the LST data covered by the cloud. The resolution of Cloud Mask data is 250m and 1000m, so the resolution have

to be downscaled to correspond with the LST data.

The China Seismic Information has recorded earthquakes of whole China since1965. The total number of the recorded earthquakes is 898035. Earthquake caused by block movement and crust compression is a kind of severe geological movement, earthquakes are the instantaneous bursts of accumulated energy, and they may lead to large area of TIR anomalies. The earthquake happens out

of the study area will also cause the TIR anomalies near the boundaries in study area, so, for the previous analysis, the earthquakes happened in $25^{\circ}N \ to \ 40^{\circ}N, \ 95^{\circ}E \ to \ 110^{\circ}E$ will be used to study the TIR anomalies in $27^{\circ}N \ to \ 37^{\circ}N, \ 97^{\circ}E \ to \ 107^{\circ}E$. But earthquakes caused by ground subsidence, human factors and so on will not cause TIR anomalies, so the earthquakes with depth=0 should be excluded. Moreover, Tronin indicated that the anomaly was sensitive to crustal

earthquakes with a magnitude more than 4.7 and for distance of up to 1000km(Tronin, Hayakawa et al., 2002). So, we select the earthquake with M$\geq$ 3.5 and depth >0 happened in the area of $25^{\circ}N \ to \ 40^{\circ}N, \ 95^{\circ}E \ to \ 110^{\circ}E$ for study, and after screening, the total number of earthquakes satisfied conditions is 3615.

3.2  RST Methodology

The RST approach is based on a multi-temporal analysis of historical data set of satellite observations acquired in similar observational conditions(Eleftheriou, Filizzola et al., 2016). Because surface environment is relatively constant, the location of high temperature and low temperature is also relatively steady. Along with the change of time, infrared brightness temperature





will change, but it changes in slow speed and small scale with obvious seasonal characteristics. Put aside influence of meteorological factors and earthquake thermal infrared anomaly, the brightness temperature in the same area and same period of the year has strong stability and regularity. So, the basic theory of RST is that the background field is constructed to extract the thermal anomalies, and the mean and variance of the land surface temperature are used to measure the degree of thermal infrared anomalies.

This method consists of three main steps, it is as follows:

● Pre-processing

RST is to construct a reference, which is regarded to be at normal state under no influence from other factors, and to measure and extract the anomalies at corresponding time. $V(r,t)$ is LST data in location $r$ at time $t$. So, the first step is to eliminate the data affected by clouds, and to remove outliers.

➢ To eliminate the influence of day to day climatological changes or season time drifts, a pre-processing will be conducted for the daily LST data:

$$\Delta V(r,t) = V(r,t) - V(t) \quad (1)$$

$\Delta V(r,t)$ is difference between value of LST acquired at time t in location r and its spatial average $V(t)$ computed on the investigated area considering only belonging to the same class, while in this study area all the pixels belong to the land class.

➢ To build cloud-mask with use of Cloud Mask data (MYD35L2). In order to be sure that only cloud-free radiances contribute to the computation of reference fields, not only those pixels but also the 24 ones in a $5 \times 5$ box around it (very often belonging cloud edges) have been excluded by the following computations of reference fields(Eleftheriou, Filizzola et al., 2016).

$$A_1(r,t) = \begin{cases} 0, if\ the\ location\ r\ was\ affected\ by\ clouds\ at\ time\ t \\ 1, otherwise \end{cases} \quad (2)$$

➢ To build an outlier-mask.

Apart from the influence of clouds, there are still many other factors will change the LST, for instance, the extreme weather (blizzard, foehn effect and so on), human's activity and forest fire. These factors will lead to rapid and dramatic changes in large area of LST data. And these anomalies should be excluded from the construction of background filed and the extraction of TIR anomalies.

$$A_2(r,t) = \begin{cases} 0, if\ the\ data\ in\ location\ r\ at\ time\ t\ was\ an\ outlier, \\ 1, otherwise \end{cases} \quad (3)$$

The detailed steps for calculating $A_3$ will be shown in next step.

● Computing Reference Fields (RF)

➢ The $\mu_{\Delta V}(r, \tau, \Delta T)$ is the mean of location r for time series T and variance $\sigma_{\Delta V}^2(r, \tau, T)$ is applied at the time $\tau$, by using homogeneous historical records collected under the temporal constraint $t \in T$ in the past (t<$\tau$), and the $V_{REF}(r, \tau, \Delta T)$ is the background field.

$$A(r,T) = A_1(r,t) * A_2(r,t) * A_3(r,t) \quad (4)$$

$$V_{REF}(r, \tau, T) \equiv \frac{\sum_{\forall t \in T}[\Delta V(r,t) \cdot A(r,t)]}{\sum_{\forall t \in T} A(r,t)} \quad (5)$$

$$\sigma_{\Delta V}^2(r, \tau, T) \equiv \frac{\sum[\Delta V(r,t) \cdot A(r,t) - \mu_{\Delta V}(r,\tau)]^2}{\sum_{\forall t \in T} A(r,t)} \quad (6)$$



➢ To compute the outlier-mask, and this method is aimed to eliminate the abnormal significant data caused by the non-seismic factors. First step is to find reference fields of minima $V_{min}(r,\tau,T)$ and maxima $V_{max}(r,\tau,T)$ to be used at the time $\tau$.

$$\Delta V_{min}(r,\tau,T) \equiv \min\{\Delta V(r,t_1),\cdots,\Delta V(r,t_N)\} \ \forall t \in T, A(r,T)=1 \quad (7)$$

$$\Delta V_{max}(r,\tau,T) \equiv MAX\{\Delta V(r,t_1),\cdots,\Delta V(r,t_N)\} \ \forall t \in T, A(r,T)=1 \quad (8)$$

Where a modified outlier-mask $A_3(r,t)=A(r,T)\cdot\delta(r,t,\tau)$ have been introduced in order to avoid contributions from accidental minima or maxima. And because the blizzard, forest fire and the large area of clouds usually cause the abnormal increase or decrease in LST with a magnitude that far bigger than the change caused by earthquakes. These data should not be included for the calculation of reference field. As shown by ALIANO et al. and GENZANO et al., spatial distribution of clouds, over a thermal heterogeneous scene, can significantly change the value of $\Delta V$ in the cloud-free pixels(Saraf, Rawat et al., 2009). The large area of clouds will bring a *cold spatial average effect* to the computation of reference fields, so when $V(r,t) < \mu_V - 2*\sigma_V$ (here, $\mu_V$ is the temporal average and the $\sigma_V$ is its standard), the value of these pixels will be excluded(Eleftheriou, Filizzola et al., 2016). Moreover, even if not producing a *cold spatial average effect,* an extended cloud coverage can determine values of V(t) and then of the considered signal $\Delta V(r,t)$ scarcely representative of the actual conditions of cloud-free pixels, so when the cloudy fraction of land portion of the scene is $> 80\%$, then all the pixels have to be excluded from the computation of reference fields(Eleftheriou, Filizzola et al., 2016).

The term $\delta_1(r,t,\tau)$ and $\delta_2(r,t,\tau)$ are defined by the expression:

$$\delta_1(r,t,\tau) = \begin{cases} 0, if\ V(r,t) - \mu_V(r,\tau,T) < -2*\sigma_V(r,t,\tau) \\ 1, otherwise \end{cases} \quad (9)$$

$$\delta_2(r,t,\tau) = \begin{cases} 0, if\ cloudy\ fraction\ of\ land\ portion\ of\ scene\ is\ > 80\% \\ 1, otherwise \end{cases} \quad (10)$$

For the outliers caused by the forest fire or blizzards and other factors, $\delta_3$ is used to remove them (with k >=2, in this study the k is set to be 4), and its expression is as follows:

$$\delta_3(r,t,\tau) = \begin{cases} 1, if\ |V(r,t) - \mu_V(r,\tau,T)| < k\sigma_V(r,t,\tau) \\ 0, otherwise \end{cases}$$

$\delta(r,t,\tau) = \delta_1(r,t,\tau) * \delta_2(r,t,\tau) * \delta_3(r,t,\tau)$ computed by using an iterative k$\sigma$-clipping technique which starts by computing $\delta(r,t,\tau)$ on the base of the first determination of $\mu_V(r,\tau,T)$ and $\sigma_V^2(r,\tau,T)$, continues by updating their values by using only space/time locations with $\delta(r,t,\tau)=1$, as follows:

$$\mu'^2_{\Delta V}(r,\tau,T) \equiv \frac{\sum_{\forall t\in T}[\Delta V(r,t)\cdot A(r,t)]}{\sum_{\forall t\in T}A(r,t)} \quad (11)$$

$$\sigma'^2_{\Delta V}(r,\tau,T) \equiv \frac{\sum[\Delta V(r,t)\cdot A(r,t) - \mu_{\Delta V}(r,\tau)]^2}{\sum_{\forall t\in T}A(r,t)} \quad (12)$$

The process should be iterated until no further exclusions, are determined by the use of the latest determination of δ (Tramutoli, 1998).This process should be paid more attention, because in the past papers, this process is always ignored.

● Change-detection step

➢ To compute the Index of change of the Environment (ALICE), and the bigger the absolute value is, the more obvious the anomaly is $\otimes_{\Delta V}(r,\tau,T)$ is the ALICE of location r at

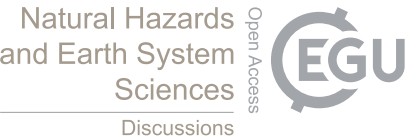

time τ, which is belonged to the time series T.

$$\otimes_{\Delta V}(r,\tau,T) = \frac{[\Delta V(r,\tau) - V_{REF}(r,\tau,T)}{\sigma_{\Delta V}(r,\tau,T)} \qquad (13)$$

➤ To determine whether the $\otimes_{\Delta V}(r,\tau,T)$ is affected by cloud. In the results, it can be easily concluded that some area in certain time will lack of data, and for these situations we have to implement a special value to inform that this data is affected by clouds and it should not be analyzed in the following part.

### 3.3 Identification of TIR anomalies

After the calculation of $K(r,\tau,T)$, the next step is to determine whether it is the TIR anomalies and to correlate the TIR anomalies with earthquakes. In this paper, the $K(r,\tau,T)$ bigger than the threshold means that there is a TIR anomaly, moreover, other conditions will be applied to conform the correlation between the TIR anomalies and earthquakes. For $K(r,\tau,T)$ and $eq(r,t)$, only these cases obey the following conditions, can be concluded that the TIR anomaly is related to $eq(r,t)$:

1) The ALICE $\otimes_{\Delta V}(r,\tau,T)$ >2. In ELEFTHERIOU's study, the threshold was set to 4, however, from the point of view of mathematical statistics, when the value is greater than two times the standard deviation, it has already belonged to the abnormal category, so in this study, the threshold is set to 2.

2) The $V(r,t)$ is surely not be blocked by clouds or affected by other factors.

3) *Spatial persistence* The TIR anomalies are gathering together, and they are not isolated being part of a group covering at least $150km^2$ within an area of $1° * 1°$ (400 pixels in the images).

4) Temporal persistence There are at least one more TIR anomaly appearing after the first TIR anomaly in 7 days.

5) The TIR anomalies appears 30 days before or 15 days after the $eq(r,t)$(Eleftheriou, Filizzola et al., 2016) .

6) The shortest distance for one point in the TIR anomalies group to the epicenter of $eq(r,t)$ is less than $R_D = 10^{0.43M}$.

The cases that the TIR anomalies satisfy conditions 1), 2), 3) and 4) but do not satisfy at least one of 5) and 5) mean that there are TIR anomalies but no corresponding earthquake. And other kinds of cases are of no TIR anomalies.

### 4. Results and analysis

A comprehensive statistical analysis and the TIR extraction results are shown in this chapter. In chapter 4.1, a statistical analysis is conducted to describe to the basic seismological conditions in the study area. And the statistical results for the correlation between earthquakes and TIR anomalies are shown and analyzed in chapter 4.2. Finally, an analysis of earthquake prediction ability for RST will be shown in chapter 4.3.

### 4.1 A basic statistical analysis for earthquakes in study area

Before studying the correlation between TIR anomalies and earthquakes, a simple analysis for



the temporal and spatial characteristic of earthquakes is conducted.

Firstly, the temporal distribution shows that the seismicity in 2008 is the most active from 2002 to 2018, and the seismicity after 2008 is obviously more frequent and violent than it before 2008. The bottom of Fig.2 indicates that there are 3615 earthquakes in the study area, $3.5 \leq M \leq 4$ is 2262, $4 \leq M \leq 5$ is 1124, $5 \leq M \leq 6$ is 198, $6 \leq M \leq 7$ is 26 and $7 \leq M \leq 8$ is 5. So the study area is a region with severe earthquake activity. And, as it is shown in upper of Fig.2, the average frequency in period A (from 2002.09 to 2007.12) is about 78. But total number of earthquakes in 2008 increases to 981 including the May 12 2008 Ms8.0 Wenchuan Earthquake, which is the most serious earthquakes in China in recent years. Though, the frequency decreases a lot after 2008 (the average frequency in this period is 243), it is still much higher than the frequency in period A. The temporal distribution tells that the seismic activity before 2008 is relatively weak, but in 2008, the seismic activity is extremely intense and reached the peak value, after 2008 the seismicity in this area still maintains a relatively strong state.

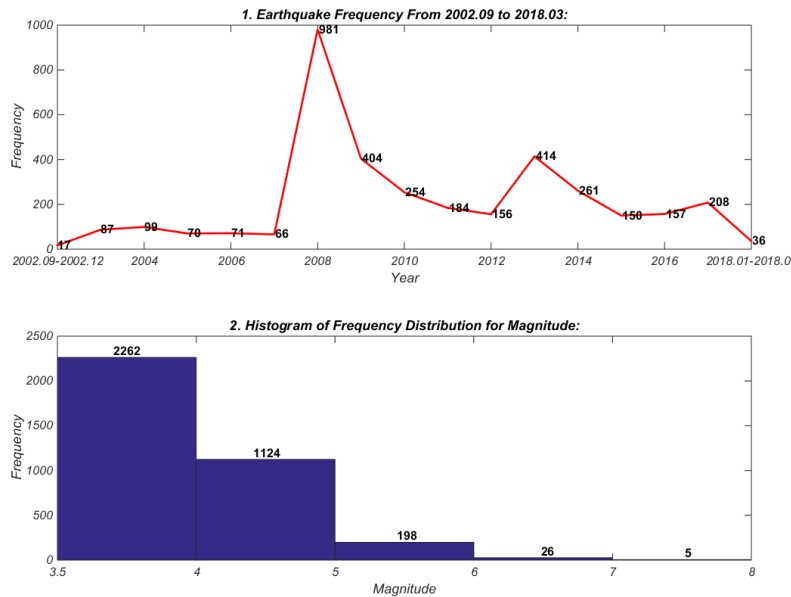

Fig.2 The temporal distribution from 2002.09-2018.03 of earthquakes with $M \geq 3.5$ in study area, and the distribution of seismic frequency along with the magnitude of the earthquakes.

**Table.1 The catalog of earthquakes with $M \geq 5.0$ before year 2008.**

| Date | Latitude\°N | Longitude\°E | Depth\km | Magnitude |
| --- | --- | --- | --- | --- |
| **2003.07.21** | 25.95 | 101.23 | 6 | 6.4 |
| **2003.10.16** | 25.92 | 101.30 | 5 | 6.2 |
| **2003.10.25** | 38.35 | 100.93 | 13 | 6.1 |
| **2003.08.18** | 29.57 | 95.60 | 33 | 6 |
| **2003.10.25** | 38.32 | 100.97 | 10 | 6 |
| **2006.07.19** | 33.03 | 96.35 | 30 | 5.9 |
| **2002.12.14** | 39.82 | 97.33 | 22 | 5.8 |

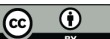



| 2005.08.05 | 26.55 | 103.15 | 21 | 5.6 |
|---|---|---|---|---|
| 2006.03.30 | 35.50 | 95.40 | 18 | 5.6 |
| 2003.11.13 | 34.75 | 103.93 | 10 | 5.5 |
| 2006.07.22 | 28.02 | 104.13 | 9 | 5.5 |
| 2006.08.25 | 28.03 | 104.01 | 7 | 5.5 |
| 2006.06.21 | 33.07 | 104.90 | 15 | 5.4 |
| 2006.07.18 | 33.07 | 96.28 | 20 | 5.4 |
| 2007.07.22 | 38.35 | 101.30 | 19 | 5.3 |
| 2004.09.07 | 34.73 | 103.92 | 19 | 5.2 |
| 2005.09.05 | 27.18 | 103.72 | 10 | 5.2 |
| 2003.08.21 | 27.42 | 101.27 | 5 | 5.1 |
| 2003.10.17 | 25.97 | 101.27 | 7 | 5.1 |
| 2003.11.01 | 25.93 | 101.22 | 3 | 5.1 |
| 2004.11.27 | 25.17 | 98.02 | 12 | 5 |
| 2005.01.05 | 32.28 | 101.55 | 5 | 5 |
| 2005.03.15 | 25.07 | 99.08 | 2 | 5 |
| 2006.07.23 | 33.03 | 96.05 | 30 | 5 |

And another evidence is shown in the Table.1, we have counted the earthquake with $M \geq 5.0$ in period B (from 2002.09 and 2007.12). There are 229 earthquakes with $M \geq 5.0$, but the total number in period A is 24, which is accounted for 10.48%, while the duration of period A is accounted for 33.87% of the total time (period A + period B). Moreover, there is no earthquakes with $M \geq 6.5$ in period A, but there are 14 earthquakes with $M \geq 6.5$ in period B. All these evidence tell us that the seismic activity in period B is much more violent and frequent.

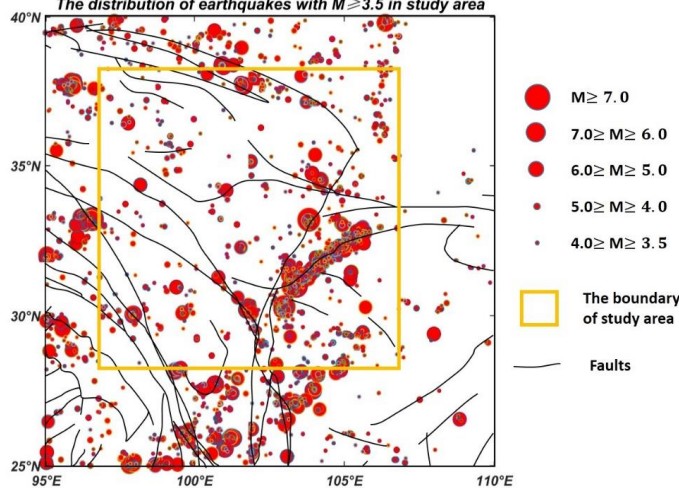

Fig.3 the spatial distribution of earthquakes in study area, and the brown rectangle refers to the study area ($27^{\circ}N$ to $37^{\circ}N$, $97^{\circ}E$ to $107^{\circ}E$). The reason showing the earthquakes out of study area is that the earthquakes closed to the study area may also cause TIR anomalies in study area.





Fig.3 shows the spatial distribution of earthquakes. The result indicates that the earthquakes mainly gather in the west and center of study area ($25^\circ$N to $40^\circ$N, $95^\circ$E to $110^\circ$E) where are mountainous and the earthquakes are mainly aggregated along faults, while the spatial distribution in the east and Sichuan Basin is much more sparse. There is a clustering phenomenon centered on earthquakes with M$\geq$ 6, the reason is that earthquakes usually occur in faults of active geological movements.

The purpose to study the temporal and spatial characteristic of earthquakes is for a general understanding of the seismic activities in study area. But there is another important reason which is to avoid the large accumulation of earthquakes in a short time and a small area, and that will lead to the same TIR anomaly corresponds to too many earthquakes, this phenomenon will excessively increase the statistical results in the latter part. It is found that there are about 233 earthquakes after the 12th May 2008 Ms8.0 Wenchuan earthquake, and the locations of these earthquakes are close to epicenter of Wenchuan earthquake. In chapter 4.2, the statistical analysis will be divided into two part, one is with the earthquakes in period C (from 2008.04 to 2008.07), and the other is without the period C.

### 4.2 The statistical analysis for the correlation between TIR anomalies and earthquakes

In this section, the TIR anomalies are extracted and a statistical analysis aiming at studying the correlation between TIR anomalies and earthquakes with M$\geq$ 4 is also been conducted. The way to judge the TIR anomalies is strictly conformed the rules mentioned in chapter 3.3.

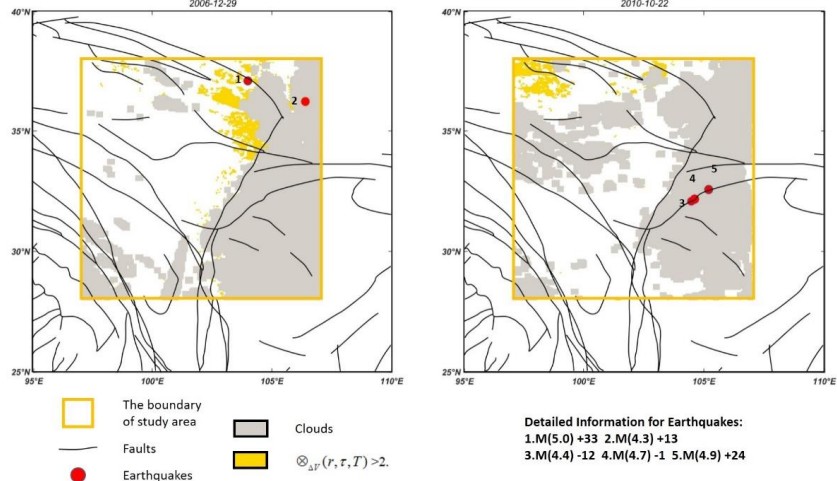

Fig.4 Two instance for the correlation between TIR anomalies and earthquakes, and the left is the TIR anomaly in 2006.12.29 corresponding to two earthquakes and the right is the TIR anomaly in 2010-10-22 corresponding to three earthquakes.

As they are shown in the Fig.4, the TIR anomalies are extracted by the RST and identification rules are applied to determine the correlation between TIR anomalies and earthquakes. After





Fig.5 Correlation analysis among TIR anomalies and earthquakes with M ≥ 4.0 in Sichuan area from Sep. 2002 to Mar. 2018. All 61 TIR anomalies are identified, and per row corresponds to one TIR anomaly. The first day of the TIR anomaly is shown in yellow, while the red cells are the following persistence. The cells in the blue rectangle mean that this day is affected by a large area of clouds, and the green cell is the final limitation for each anomaly. Cells with numbers indicate days of occurrence, and magnitude, of seismic events. The rows in the dotted box mean that there is no earthquakes corresponding to this TIR anomaly.



extracting, the total number of TIR anomalies is 61 and the correlation result is shown in the Fig.5. Considering the examples reported in Fig.4, which are summarized in row 18 and 33, the cells in yellow corresponding to the first day of TIR anomaly are 2006-12-29 and 2010-10-22 respectively. It can be concluded from the Fig.5 that 35 TIR anomalies correspond to earthquakes, while the other

26 do not, which are rows 1, 2, 3, 4, 5, 7, 8, 9, 10, 11, 14, 15, 16, 17, 19, 20, 21, 22, 23, 35, 41, 43, 44, 49, 51 and 52. The correlation rate is 57.4%. And as it is shown in Fig 6 that most TIR anomalies appear before earthquakes.

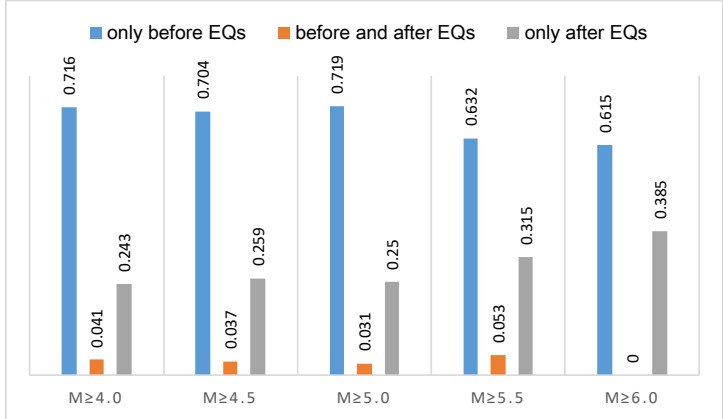

Fig.6 Distribution of TIR anomalies with respect to the earthquakes occurrence for different class of magnitude.

However, as it has been mentioned in section 4.1 that period C may increase the total number of TIR anomalies and the correlation rate a lot. So, the experiment without period C is also been

conducted, and the number of TIR anomalies is 60, while the correlation rate is 56.7%, both of which are roughly the same as the former result. Theoretically, in period C, the frequency of earthquakes is high and the magnitude of earthquakes is large, which should generate a lot of TIR anomalies and have good correspondences with earthquakes. But there is only one TIR anomaly corresponding to 5 earthquakes, Fig.7 may indicate the reason. It indicates that earthquakes are

gathering in several faults, but the spatial location of these faults are always blocked by the clouds, and the percentage is bigger than 90%. With the long time and large area of clouds blocking, the TIR anomalies caused by the earthquakes in period C may cannot be extracted by RST.

For a more comprehensive analysis for Fig.5, there are 23 TIR anomalies in period A with 19 corresponding to no earthquakes, while there are 38 TIR anomalies in period B with only seven

corresponding to no earthquakes and the correlation rate reaches 82% which is much higher than 57%. Fig.2 and Fig.8 can explain this phenomenon, in period A, the magnitudes of earthquake intensity are small, the frequency is low, and nearly half of the earthquake occurs in the cloudy region or its adjacent area, so the earthquakes are difficult to correspond to the extracted anomalies, and some potential anomalies may have not been extracted because of clouds. As for the results in

period B, the frequency and magnitude of earthquakes in the sparsely clouds areas increase a lot, thermal anomalies are more likely to be extracted and TIR anomalies are more likely to correspond to earthquakes.





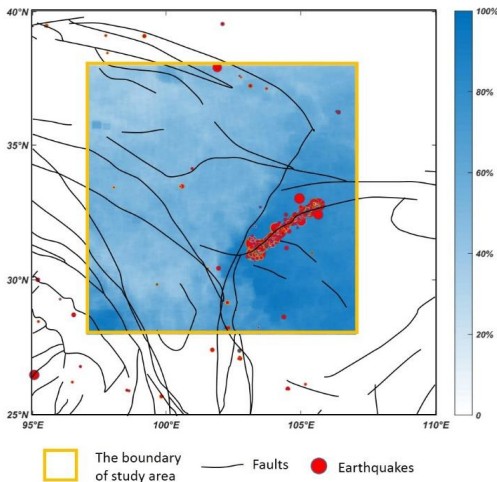

Fig.7 The distribution of earthquakes in period C and the frequency of each pixel being blocked by cloud in period C, and the higher values mean that the pixels are more frequent blocked by clouds.

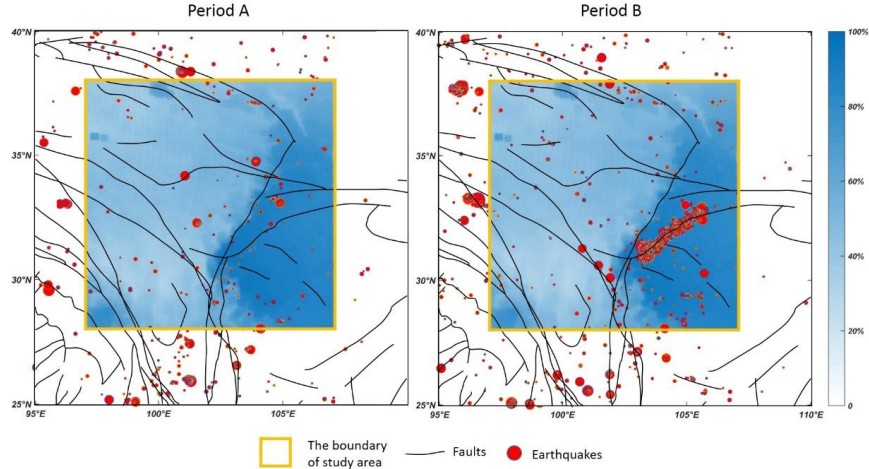

Fig.8 The distribution of earthquakes in period A, B and the frequency of each pixel being blocked by cloud in period C, and the higher values mean that the pixels are more frequent blocked by clouds.

4.3 The evaluation of earthquake prediction ability for RST

10      Aiming at evaluating the earthquake prediction ability of RST in Sichuan area, only the true positive rate for correspondence between TIR anomalies and earthquakes with M≥ 4.0 is far from enough. So four types of data are counted and four types of rate are calculated:

     TP1: True Positive 1, the total number of TIR anomalies corresponding to earthquakes.

     FP: False Positive, the total number of TIR anomalies corresponding to no earthquakes.

15      TP2: True Positive 2, the total number of earthquakes corresponding to TIR anomalies.



FN: False Negative, the total number of earthquakes corresponding to no TIR anomalies.

PPV: The rate of TIR anomalies which correspond to earthquakes to the total TIR anomalies.

FPR: The rate of TIR anomalies which correspond to no earthquakes to the total TIR anomalies.

TPR: The rate of earthquakes which correspond to TIR anomalies to the total earthquakes.

FNR: The rate of earthquakes which correspond to no TIR anomalies to the total earthquakes.

**Table.2 The statistical result of earthquakes with M≥ 5.0**

| **M ≥ 5.0** | TP1 | FP | TP2 | FN |
|---|---|---|---|---|
| | 19 | 42 | 32 | 218 |
| | PPV:31.1%   TP1/(TP1+FP) | | TPR:12.8%   TP2/(TP2+FN) | |
| | FDR:68.9%   FP/(TP1+FP) | | FNR:87.2%   FN/(TP2+FN) | |

For a more accurate understanding of the 8 parameters, an example is shown in Table 2. The
example studies the earthquakes with M ≥ 5.0, and result indicates that there are 61 (TP1+FP) TIR
anomalies appearing in study duration, and 19 (TP1) of them correspond to earthquakes while the
other 42 (FP) do not, the probability of exact correspondence between the TIR anomalies and
earthquakes is 31.1% (PPV) while the probability of no correspondence is 68.9% (FPR). Moreover,
there are 250 (TP2+FN) earthquakes with  M ≥ 5.0 in the study area, and 32 (TP2) of them
correspond to TIR anomalies while the other 218 (FN) correspond to no TIR anomalies, the
probability of exact correspondence between the earthquakes and TIR anomalies is 12.8% (TPR)
while the probability of no correspondence is 87.2% (FNR). We have calculated the earthquakes
with M≥ m  (m={3.5, 3.6, 3.7,…,7.8, 7.9, 8.0}). And both the experiments with and without period
C are calculated, the results show that they are roughly the same, so in this section, we only discuss
the result with period C.

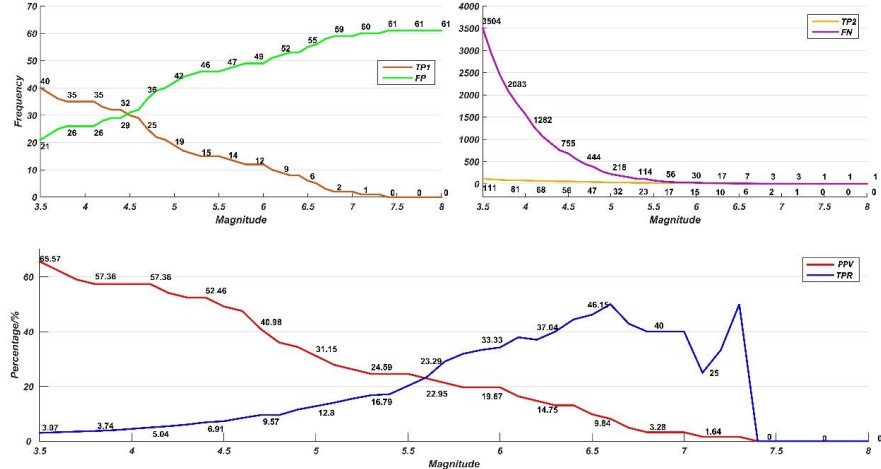

Fig.9 The statistical result of earthquakes including the period C (from 2008.04 to 2008.07), and the
curve TP1, FP, TP2, FN, PPV and TPR are corresponding to the example in Table 2.

**Table.2 The detailed results of earthquakes including the period C (from 2008.04 to 2008.07).**

| M | TP1 | FP | TP2 | FN | PPV | FPR | TPR | FNR |
|---|---|---|---|---|---|---|---|---|
| 3.5 | 40 | 21 | 111 | 3504 | 0.656 | 0.344 | 0.031 | 0.969 |



| | | | | | | | | |
|---|---|---|---|---|---|---|---|---|
| 3.6 | 38 | 23 | 100 | 2933 | 0.623 | 0.377 | 0.033 | 0.967 |
| 3.7 | 36 | 25 | 90 | 2454 | 0.590 | 0.410 | 0.035 | 0.965 |
| 3.8 | 35 | 26 | 81 | 2083 | 0.574 | 0.426 | 0.037 | 0.963 |
| 3.9 | 35 | 26 | 76 | 1807 | 0.574 | 0.426 | 0.040 | 0.960 |
| 4 | 35 | 26 | 74 | 1563 | 0.574 | 0.426 | 0.045 | 0.955 |
| 4.1 | 35 | 26 | 68 | 1282 | 0.574 | 0.426 | 0.050 | 0.950 |
| 4.2 | 33 | 28 | 62 | 1068 | 0.541 | 0.459 | 0.055 | 0.945 |
| 4.3 | 32 | 29 | 59 | 909 | 0.525 | 0.475 | 0.061 | 0.939 |
| 4.4 | 32 | 29 | 56 | 755 | 0.525 | 0.475 | 0.069 | 0.931 |
| 4.5 | 30 | 31 | 54 | 685 | 0.492 | 0.508 | 0.073 | 0.927 |
| 4.6 | 29 | 32 | 51 | 550 | 0.475 | 0.525 | 0.085 | 0.915 |
| 4.7 | 25 | 36 | 47 | 444 | 0.410 | 0.590 | 0.096 | 0.904 |
| 4.8 | 22 | 39 | 40 | 379 | 0.361 | 0.639 | 0.095 | 0.905 |
| 4.9 | 21 | 40 | 36 | 276 | 0.344 | 0.656 | 0.115 | 0.885 |
| 5 | 19 | 42 | 32 | 218 | 0.311 | 0.689 | 0.128 | 0.872 |
| 5.1 | 17 | 44 | 30 | 183 | 0.279 | 0.721 | 0.141 | 0.859 |
| 5.2 | 16 | 45 | 28 | 152 | 0.262 | 0.738 | 0.156 | 0.844 |
| 5.3 | 16 | 45 | 27 | 127 | 0.262 | 0.738 | 0.175 | 0.825 |
| 5.4 | 15 | 46 | 23 | 111 | 0.246 | 0.754 | 0.172 | 0.828 |
| 5.5 | 15 | 46 | 19 | 75 | 0.246 | 0.754 | 0.202 | 0.798 |
| 5.6 | 14 | 47 | 17 | 56 | 0.230 | 0.770 | 0.233 | 0.767 |
| 5.7 | 13 | 48 | 16 | 39 | 0.213 | 0.787 | 0.291 | 0.709 |
| 5.8 | 12 | 49 | 15 | 32 | 0.197 | 0.803 | 0.319 | 0.681 |
| 5.9 | 12 | 49 | 15 | 30 | 0.197 | 0.803 | 0.333 | 0.667 |
| 6 | 12 | 49 | 13 | 25 | 0.197 | 0.803 | 0.342 | 0.658 |
| 6.1 | 10 | 51 | 11 | 18 | 0.164 | 0.836 | 0.379 | 0.621 |
| 6.2 | 9 | 52 | 10 | 17 | 0.148 | 0.852 | 0.370 | 0.630 |
| 6.3 | 8 | 53 | 8 | 12 | 0.131 | 0.869 | 0.400 | 0.600 |
| 6.4 | 8 | 53 | 8 | 10 | 0.131 | 0.869 | 0.444 | 0.556 |
| 6.5 | 6 | 55 | 6 | 7 | 0.098 | 0.902 | 0.462 | 0.538 |
| 6.6 | 5 | 56 | 5 | 5 | 0.082 | 0.918 | 0.500 | 0.500 |
| 6.7 | 3 | 58 | 3 | 4 | 0.049 | 0.951 | 0.429 | 0.571 |
| 6.8 | 2 | 59 | 2 | 3 | 0.033 | 0.967 | 0.400 | 0.600 |
| 6.9 | 2 | 59 | 2 | 3 | 0.033 | 0.967 | 0.400 | 0.600 |
| 7 | 2 | 59 | 2 | 3 | 0.033 | 0.967 | 0.400 | 0.600 |
| 7.1 | 1 | 60 | 1 | 3 | 0.017 | 0.983 | 0.250 | 0.750 |
| 7.2 | 1 | 60 | 1 | 2 | 0.017 | 0.983 | 0.333 | 0.667 |
| 7.3 | 1 | 60 | 1 | 1 | 0.017 | 0983 | 0.500 | 0.500 |
| 7.4 | 0 | 61 | 0 | 1 | 0.000 | 1.000 | 0.000 | 1.000 |
| 7.5 | 0 | 61 | 0 | 1 | 0.000 | 1.000 | 0.000 | 1.000 |
| 7.6 | 0 | 61 | 0 | 1 | 0.000 | 1.000 | 0.000 | 1.000 |
| 7.7 | 0 | 61 | 0 | 1 | 0.000 | 1.000 | 0.000 | 1.000 |
| 7.8 | 0 | 61 | 0 | 1 | 0.000 | 1.000 | 0.000 | 1.000 |
| 7.9 | 0 | 61 | 0 | 1 | 0.000 | 1.000 | 0.000 | 1.000 |





| 8 | 0 | 61 | 0 | 1 | 0.000 | 1.000 | 0.000 | 1.000 |

As it is shown in Fig.9, PPV declines as the magnitude increases, while FP obviously increases, this phenomenon indicates that with the increase of the magnitude, the number of TIR anomalies corresponding to the earthquakes decreases. TPR and FN decrease steadily, because the total number
of earthquake samples is decreasing.

But, what needs more attention is the rates (PPV, TPR). Firstly, a general perceptual analysis shows that PPV decreases steadily with the increasing of M, while TPR is increasing when M≤ 6.6 and M = 7.2, 7.3, the maximum of TPR is 50% when M=6.6 and 7.3, and the TPR decreases when M= 6.7~7.1. When M is 3.5 and 4.0, PPV is 65.6% and 57.4%, it means that when there is a
10 TIR anomaly appearing, there is a possibility of 65.6% (57.4%) that earthquakes with M≥3.5 (4.0) will happen. When the M is 5, 6, 7, the E is 31.1%, 19.7%, 3.3%, and these are much lower than PPV of M=3.5 and 4.0. It is concluded from the change of PPV curve that where there is a TIR anomaly, there will be more than 50% possibility of an earthquake with M≥3.5 (4.0) in study area. However, this does not mean that when there is a TIR anomaly, there will be strong earthquakes
with M≥5.0 in the study area. On the contrary, the probability of earthquakes with high magnitude is still not high.

The curve of TPR tells the probability of a TIR anomaly when earthquakes occur. When M=3.5 the TPR is 3.1%, and with the increasing of M, TPR increases steadily, but it keeps in a low state when M ∈ [3.5, 5.4] and the TPR are lower than 20%. The results show that the earthquakes with
20 low magnitude have relatively low possibility (less than 20%) to correspond to TIR anomalies, however, the earthquakes with M ≥ 6.0, which are very destructive, have a relative high correspondence. The high correspondence is of great significance to earthquake prediction. It tells us that a considerable number of the destructive earthquake is more likely to be predicted by this method.
According to two results, it can be concluded that, when a TIR anomaly occurs, there is a 57.3% possibility of an earthquake with M≥4.0, and when there is a strong earthquake with M≥6.0, more than 1/3 of the earthquakes correspond to thermal anomalies. Most TIR anomalies correspond to the earthquakes with M≥4.0, however when M≥6.0, the PPV is relatively low, and that means the false alarm rate for strong earthquakes is high. For TPR, it is increasing with the magnitude, and
when M=6.6 and 7.3 it is 50%. It can be concluded from the curve TPR that the greater the magnitude of an earthquake is, the more likely it is to be predicted by this method. But, in fact the PPV and TPR are low, or in other words the FPR and FNR, which are negative for the prediction ability of RST, are really high. All in all, the false alarm rate for M≥4.0 is 42.6%, with the increasing of M the FPR will become higher, and the missing rate for M≥4.0 is 96%, it seems that
when M<5.5, there is no obvious correlation between TIR anomalies and earthquakes, though TPR will increase when M is becoming bigger, while the maximum of TPR is still 50%, which is also an unsatisfied value. So, the prediction ability of RST in Sichuan area is limited.

5. Discussion
To compare the results with the previous similar studies, the summary of the four similar studies is shown in Table.3. It is obvious that PPV of this paper is relatively lower than others, so it is important to verify its actual added value in comparison with a random alarm function (see for instance Eleftheriou). The detailed method is accessible in chapter 3.4 of (Eleftheriou, Filizzola et



al., 2016), and the result is shown in Fig.10. When M≥ 3.5, the point is at the upper part of the random guess, which means that there is no obvious correlation between TIR anomalies and earthquakes with M≥ 3.5, it seems that it is of a casual correlation. When M≥ 4.0 and M≥ 4.5 these two points are still very close to the line, but it is at the lower part, and that means that a non-casual correlation actually exists among the extracted TIR anomalies and earthquakes (M≥ 4.0 and ≥ 4.5). However the correlation is not strong. The result in this paper is different from Eleftheriou's study, in her study, there is a much more obvious and strong correlation between the TIR anomalies and earthquakes. The cause may be that, as it being shown in Fig.8, east and southeast corner of the study area is always blocked by the clouds, and total time taken up is over 90%, at the same time there are many earthquakes happening in this area, and these earthquakes cannot be related to TIR anomalies because of the lack of data, and they are inevitably counted into FNR, which is v in Molchan error analysis, making the correlation weaker. For the M≥ 5.0, 5.5 and 6.0, the points are obvious under the random guess, with the increasing of M, the non-causal correlation is becoming stronger.

Tronin indicated that the anomaly was sensitive to crustal earthquakes with a magnitude more than 4.7 and for distance of up to 1000km(Tronin, Hayakawa et al., 2002). But in this study, when M=4.7, the TPR is 9.6% and the FNR is 90.4%, and the point in Fig.10, when M≥ 4.5, is very close to the random guess, so the statistical result does not support the opinion that TIR anomaly is sensitive to the earthquakes with M ≥ 4.7. When M≥ 5.9, the earthquakes seem to be sensitive to TIR anomalies according to the Table.3. The reason for this failure to conform to previous conclusion may be the regional structure and geological movement, cloud cover, the effectiveness of method extracting TIR anomalies and so on. Further study is needed.

Table.3 A general summary for research studying the statistical correlation between TIR anomalies and earthquakes.

| Author's name | Data Source | Study area | Duration | PPV |
|---|---|---|---|---|
| Genzano | GMS-5/VISSR | Taiwan | 1995.09-2002.09 | 100% (M≥ 4 or 4.5) 94% (M≥ 5.0) |
| Tramutoli | | Italian southern Apennines | 2012.07-2013.06 | 67% |
| Eleftheriou Alexander | MSG/SEVIRI | Greece | 2004-2013 | 93% (M≥ 4.0) |
| Ying Zhang | MODIS | China, Sichuan Area | 2002.09-2018.03 | 57.4% (M≥ 4.0) |

We have counted the total number of TIR anomalies and number of FP in each month of two studies. And it is found that the TIR anomalies are gathering in November, September, January and February in Eleftheriou's study, while in this paper, TIR anomalies are gathering in November, September and January. A line which means the percentage of area not blocked by clouds in Sichuan area is also presented in Fig.11. In this paper, there is a significant positive proportional correlation between the number of TIR anomalies and the area not blocked by clouds. When the percentage is high, the number of TIR anomalies is high, and when the percentage is low, the number of TIR anomalies is low. So there might be many TIR anomalies related to earthquakes, but they are blocked by the clouds and cannot be extracted. However, what is the true cause, blocked by clouds, caused



by seasonal weather or else? It is remained to be solved. Moreover, there is another interesting phenomenon that many TIR anomalies corresponding to no earthquakes are gathering in November and September, and both months are not cloudy and are in cold weather. So, why many FP are gathering in these two months is also remained to be studied.

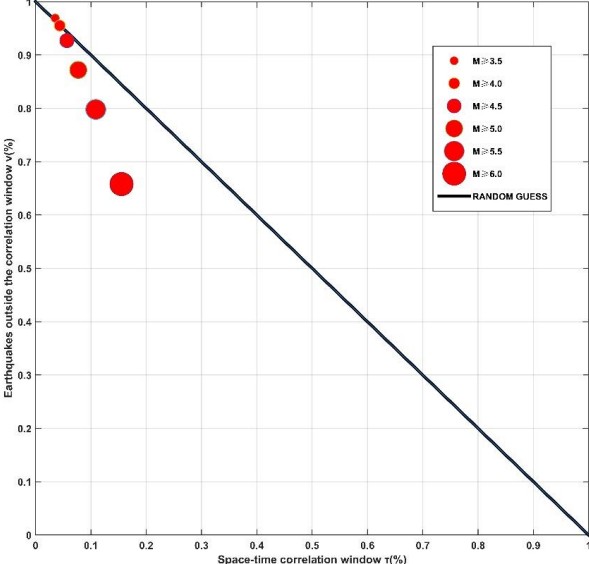

Fig.10 Molchan error diagram analysis computed for different class of magnitude and TIR anomalies on the study period (2002.09-2018.03), and the red circles refer to earthquakes occurred before and after the appearances of TIR anomalies.

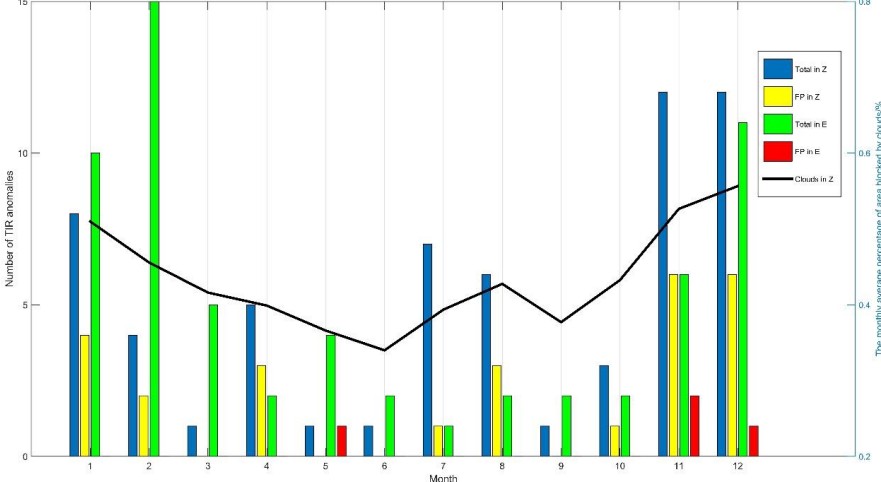

Fig.11 The monthly average percentage of area not blocked by clouds in study region in Zhang's study, or in other words, in this study. And the bar chart is the numbers of different kind of TIR anomalies. 'Total in Z' is the total number of TIR anomalies in this study, 'FP in Z' is the number of TIR anomalies corresponding to no earthquakes in this study. 'Total in E' is the total number of TIR

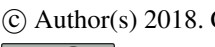



anomalies in Eleftheriou's study, 'FP in E' is the number of TIR anomalies corresponding to no earthquakes in Eleftheriou's study.

6. Conclusion

(1) 18 years statistical analysis for the correlation between earthquakes and TIR anomalies shows that 57.4% TIR anomalies correspond to the earthquakes with $M \geq 4.0$ in Sichuan area, and the bigger the M is, the more likely the earthquakes will correspond to TIR anomalies. The low PPV and TPR may be due to the large areas in the study region are covered by clouds all year round.

(2) The low PPV and TPR suggest that the earthquakes prediction ability in Sichuan area with use of RST is limited. For the strong earthquakes with $M \geq 6.0$, though the false alarm rate is high, the missing rate is relatively low. RST applied to the study area and has certain prediction ability for strong earthquakes.

(3) There is no obvious correlation between the earthquakes with $M < 5.0$ and the TIR anomalies extracted by RST in Sichuan area. However, the underlying causes of this situation need to be further studied.

(4) RST put up in this paper or in Eleftheriou's study is still seriously affected by the clouds and the seasons. It is necessary to improve and optimize algorithms and statistical methods to exclude the influence of clouds and seasons.

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
