# Peer review of "A Statistical Analysis of TIR Anomalies Extracted by RST in Relation to an Earthquake in the Sichuan Area using MODIS LST Data"

_Natural Hazards and Earth System Sciences, 2018_

## Referee Comment (RC1) · X. Lu (Referee) · 19 Sep 2018

This paper is very meaningful and interesting which studied the relationship between TIR anomalies and earthquakes in Sichuan area from 2002 to 2018. It has a very long study period and a detailed statistics. I have some suggestions below which I feel would strengthen the paper.

Page 1 Line 25 in abstract – PPV, FDR, TPR and FNR are best added their full names. Line 35-The "TIR" might be repeated with the "outgoing longwave satellite".

Page 2 Line 34- What's the meaning of the "t no similar results"?

Page 3 * Add the names of the main faults in Fig.1. An overall map of China and Sichuan could also be considered.

Page 4 * There are three spatial resolutions -250m, 500m, 1000m of the MODIS data, which one was used in your paper? Line 32- How did you get this conclusion-"But earthquakes caused by ground subsidence, human factors and so on will not cause TIR anomalies"?

Page 6 * Maybe I did not understand the calculation process, but the right sides of the formula (11) and (12) are the same as the right sides of the formula (5) and (6). What's the purpose for repeating the same expression?

Page 7 Line 14- "In ELEFTHERIOU's study" should add the reference. Line 29- "one of 5) and 5)" is "one of 5) and 6)"?

Page 9 Line 3- Period A is from 2002.09 to 2007.12, is Period B also from 2002.09 to 2007.12? Line 9- I cannot find the brown rectangle.

Page 12 Line 31- "Thermal anomalies are more likely to be extracted and TIR anomalies are more likely to correspond to earthquakes". Do you have a standard of the thermal anomaly? Just as the size and the amplitude of the anomaly. *There are 61 anomalies with period C, but 60 anomalies without period C because of the more cloud in period C in study area, however, little difference of clouds in Fig.7 and Fig.8 can be detected. Should you make the difference more obvious?

Page 14 * In Fig.9, the percentage of PPV and TPR with magnitude $\geq 7.4$ is 0, is there no TIR anomalies for these earthquakes? I have read some papers that there were TIR anomalies when Wenchuan 8.0 earthquake occurred. * In 4.3 The evaluation of earthquake prediction ability for RST and 5 Discussion, the description is complicated, can you simplified them slightly?

Page 16 Line 37- "the prediction ability of RST in Sichuan area is limited". I think this sentence is too absolute. Many earthquakes occurred in Longmenshan fault in

Sichuan area had been studied by scientists, and got some anomalies before or after the earthquake. For example, Singh (2010) had found precursory signals using satellite and ground data associated with the Wenchuan earthquake of 12 May 2008. Zhang Yuan-sheng (2010) also detected the TBB anomalies before Wenchuan earthquake; and Zhang Xuan (2013) had analyzed the thermal infrared anomaly before the Lushan 7.0 earthquake. So, maybe the data was not suitable for this region, not simply because of the cloud. This should be described in the paper.
* * *

---

## Referee Comment (RC2) · Anonymous Referee #2 · 26 Sep 2018

I find the topic of the paper very interesting and promising. In my opinion it can be considered for publication but there are many issues that must be addressed. The paper needs to be rewritten in better English as it is incomprehensible in many cases, distracting from the authors' primary contribution. I suggest the authors substantially revise, through elimination of ancillary matters (reduce the extend of the text and try to eliminate any repetitions). In addition, some suggestions that I hope to help authors: Fig.1: The faults cannot be well discriminated. You should change the color or the width of the symbol. Page 3: You repeat the same information about the study area. You should fix it. Page 3, last line: What do you mean with the word "scope"? Page 4, first line: The coordinates have been given in the previous page. Page 4, lines 30-35: I

think it would be more helpful to give a distance radius. Page 4, line 42: Please give the year of Eleftheriou et al publication. Page 6, lines 8-10: "And because the blizzard, forest fire and the large area of clouds usually cause the abnormal increase or decrease in LST with a magnitude that far bigger than the change caused by earthquakes." I think you must provide at least one reference for this statement. In your methodology section it is not clear whether you use RETIRA index or not. In the abstract you write: "In this paper, a refined RST data analysis and Robust Estimator of TIR Anomalies (RETIRA) index were used to extract the TIR anomalies from 2002 to 2018 in Sichuan 20 area with use of Moderate-resolution Imaging Spectro-radiometer (MODIS) Land Surface Temperature (LST),.........." but I cannot find any reference to RETIRA index in the main text. I think that RST methodology must be re-written in order to become clear where the already known RST RETIRA methodology stops and how your refinements have been implemented. Page 6, lines 37-38: The sentence is ....strange??? Eleftheriou et al., 2016 applied the RST RETIRA index and not the ALICE index. As far as I know, these are two different indexes. In pages 6 and 7 you mention ALICE. Page 7: The conditions need to be refined using better syntax. Page 7, line 29: Please correct the sentence: one of 5) and 5) mean that there are TIR anomalies but no corresponding earthquake. Page 7, line 30: What do you mean? Please explain. Page 8, first paragraph: Please refine your English or exclude this paragraph. In my opinion, Fig. 2 is enough. Fig. 3: Its caption needs refinement. I propose to move this fig and the first two paragraphs of page 10 in the STUDY AREA Section. Fig. 4: Please, rewrite the caption of the Figure. Page 13, line 12: " ....is far from enough". I don't understand. Is it good? Is it Very good? or the opposite? Section 4.3: (The evaluation of earthquake prediction ability for RST). It is very complicated. I think that you must make this section more attractive and easy to be read. Page 19: "Conclusions" and not "Conclusion". Using bullets from the first line is not recommended. And of course one simple question: Many researchers and among them Eleftheriou et al., use to call "thermal anomaly" the pixels with $4\sigma\Delta\hat{\text{I}}d'(x,y) > 4$. The fact that you selected a threshold equal to 2 instead of 4, makes the results comparable?

---

## Referee Comment (RC3) · Anonymous Referee #3 · 3 Oct 2018

Dear Authors,

It was pleasure to read this paper. It could be a contribution to the actual knowledge in the field of possible precursor signals of the earthquake activity, but I have a number of severe reservations right now.

The goal of your paper it is quite clear, namely to conduct a correlation analysis among earthquakes occurred in Sichuan region (China) and satellite TIR anomalies highlighted by RST methodology and RETIRA index.

Main comments

Whole paper needs to be rewritten in a better English. In particular some sections, like the paragraph 3.2 (RST methodology) or paragraph 5 (Discussion), are not clear and only after various readings the paper can be understood.

The major part of citations should be revised, both in the form (sometime given name is used instead of surname like at line 6 of page 2) and in content (some wrong citations have been used or some important citations miss, as for RETIRA index).

To identify thermal anomalies possibly related to impending earthquakes, you used LST (Land Surface Temperature) products retrieved by the radiance collected by MODIS sensor on board of the polar satellites EOS/AQUA and EOS/TERRA.
Taking in mind that Authors who proposed the RST approach shown the advantages offered by the use of sensors onboard of geostationary platforms instead of sensor onboard of polar satellite packages (see the paper Filizzola, C., N. Pergola, C. Pietrapertosa, and V. Tramutoli (2004), Robust satellite techniques for seismically active areas monitoring: a sensitivity analysis on September 7, 1999 Athens's earthquake, Phys. Chem. Earth, 29(4–9), 517–527, doi:10.1016/j.pce.2003.11.019), since 2004 the major part of RST applications to thermal monitoring of seismogenic areas have been carried out using TIR satellite records acquired by sensors onboard of geostationary satellites (as also you have reported in the your paper). Now, my question is why you prefer to use EOS/MODIS data instead that TIR records collected by sensor on geostationary platforms (e.g. the Japanese MTSAT satellite)?
Moreover, LST (Land Surface Temperature) products have been take in account. LST products are very useful to reduce variability of atmospheric water vapor, but in the computation of LST several approximations are necessary (e.g. emissivity, total water vapour content, ecc.), which should produce errors (also of 4-5 K degree) in the satellite LST estimations. Taking in mind, that thermal anomalies possibly related to seismic activity are of low intensity, wrong LST estimation could mask and/or generate false anomalies. Have you an idea of the impact of this errors on the your analysis?

Earthquake catalogue (China Seismic Information; http://www.csi.ac.cn/) used to verify possible correlation with TIR anomalies is inaccessible. Please provide a correct URL. Anyway, consulting a different seismic catalogue, i.e. UGSG catalogue (https://earthquake.usgs.gov/earthquakes/search/), using a similar criteria (M≥3.5; Depth >0; region from 25°N to 40°N and from 95°E to 110°E; time since August 1, 2002 up to April 15, 2018) I found 2369 earthquakes, respect to 3615 seismic events reported in the your paper. A comparable numbers of seismic events, i.e. 3828, is obtained when the USGS catalogue is consulted starting by 1965. Have you use seismic data from 2002 or from 1965? In the first case (i.e. 2002) how you explain this difference (2369 vs 3615)? In the second case (i.e. 1965), because MODIS data are available since 2002, how is possible found some relations among TIR anomalies and earthquakes (before 2002)? In this last case, please provide a correct analysis.

About the performed correlation analysis among the appearances of TIR anomalies and earthquake occurrences, you should be in mind that working in the optical band, a wide presence of meteorological clouds, as well as the

lack of satellite data, do not allow to give continuity to the observations, which is necessary to identify possible TIR anomalies or to fully appreciate a possible space–time persistence of previously occurred TIR anomalies, producing in this way a possible overestimation of missed events. Please, consider this suggestion and provide a more convincing analysis. As consequence also your conclusions should be reconsidered.

Specific comments

Page 1 - Lines 18-19; In the abstract, you announced that a refined RST data analysis and Robust Estimator of TIR Anomalies (RETIRA) index were used, but in the text I have not read any new improvements to the RST methodology, if not those reported in Eleftheriou et al. (2016). Otherwise please explain better the refinements made to the RST technique. Moreover, add the reference of RETIRA index.

P1 - L25; Please provide the complete name of PPV, FDR, TPR and FNR.

P1 - L26; The sentence "the prediction ability of RST in Sichuan area is limited" is too strong!

P4 - L34-36; I not understand the sense of this sentence "Moreover, Tronin indicated that the anomaly was sensitive to crustal 1000kmearthquakes with a magnitude more than 4.7 and for distance of up to 1000km" in this position.

P5 - L14-24; Cloudy pixel, as well as pixels declared as edge clouds, should be exclude before the computation of ΔV(r,t) otherwise effects due to cloudiness are not removed and false TIR anomalies could be generate.

P5 - L25-31; How you identify the extreme weather events (e.g. blizzard)?

P6 - L9-13; The reference Saraf et al. (2009) is correct? I not found no mention about effects of cloudy pixels on ΔV in this publication.

P6 - L13-20; *Cold spatial average effect as reported in* Aliano et al. (2009), Genzano et al. (2010) and Eleftheriou et al. (2016) could affect the whole TIR scene. Rightly you have take in account this effect, but in opposite way of Genzano at al. (2015) or Eleftheriou et al. (2016) you work at pixel level instead of whole scene level. In this way, the above mentioned effect could be not removed in the computation of reference fields.

P6 - L32-35; The sentence "This process should be paid more attention, because in the past papers, this process is always ignored." is wrong. In all applications of RST approach, kσ-clipping method (always applied) guarantee to remove outlier (i.e. extreme events) from the computation of reference fields. Please, consider to rewrite better the sentence.

P6-P7 (Change detection step); Although you have announced the computation of ALICE index, the index reported in the equation 13 should be the RETIRA index (correct equation can be found in Filizzola et al. 2004; Tramutoli et al. 2005). Please, correct it.

P7 L9-23; The criteria used to identify TIR anomalies are the same introduced for the first time in Genzano et al. (2015) and in Eleftheriou et al. (2016) in order to indentify Significant Sequences of TIR Anomalies (SSTAs). Please consider to call it in similar way, mentioning these two publications.
Moreover, starting from a mathematical point of view you have consider to set a threshold K equal 2, if you have a normal distribution (Gaussian) a 2 times the standard deviation could be sufficient to identify anomalies. In addition, RETIRA index (as well as ALICE index) give the possibility to evaluate in term of SIgnal-Noise ratio (S/N) the intensity of anomalies (see Tramutoli at al. 2001 or Tramutoli et al. 2005 for more details). Please, take in mind this suggestions when choose threshold k.

P9 L3; Period B is the same of period A (i.e. from2002.09 and 2007.12)?

P10 Fig.4; Figure shown are not a good example of TIR anomalies possibly associated to earthquakes. In the example on the right part (TIR map of 2010/10/22) earthquakes seems not satisfy the rules announced in chapter 3.3. Moreover, to show the whole sequence of TIR anomalies, not only one day with TIR anomalies, could help the reader to better understand the concept of Significant Sequence of TIR Anomalies.

P11 Fig.5; As reported in the caption "The cells in the blue rectangle mean that this day is affected by a large area of clouds, ...", now, some days with TIR anomalies belonging to several sequences of TIR anomalies (i.e. 2, 10, 27, 32, 35, 36, 37, 45, 50, 58, 59) are affected by a wide cloudy coverage, all this lets thinks that TIR anomalies due to meteorological effects are not removed from the analysis (as suggested in Eleftheriou et al., 2016).

P12 L21-33. Rightly, you are reported that cloudy coverage could prevent to observe with continuity the presence of TIR anomalies, this is a intrinsically limitation of satellite technologies which work in the optical band, and not of RST methodology. Please revise your sentences.

P13-16 Paragraph 4.3 (The evaluation of earthquake prediction ability for RST); The performed analysis not have any sense if carried out in this way. Mainly, the analysis on the rate of earthquakes which correspond ("TPR") or not ("FNR") to TIR anomalies it is very complicated to perform, because gaps in observations, due to the lack of satellite data or to a wide presence of meteorological clouds make impossible to give a continuity to the observations, which is necessary to identify possible TIR anomalies or to fully appreciate a possible space–time persistence of previously occurred TIR anomalies, as consequences the relation one to one (earthquake-TIR anomalies) that you are looking is corrupted by this limitation. Anyway, before to comment the results of a some kind of sensitivity analysis this circumstance should be announced.

---

## Author Comment (AC1) · 8 Oct 2018

Many thanks for your comments.

The following is my reply for your comments and questions.

Page 1 Line 25: The full names of PPV, FDR, TPR and FNR will be added.

Page 1 Line 35: "outgoing longwave" will be deleted.

Page 2 Line 34: "t no similar" will be corrected, it is "not similar".

Page 3 Fig.1: the names of the main faults will be added, and i will think about adding the overall map of China and Sichuan in this Fig.

Page 4 Line 18: In this study, we used the 560m data.

Page 4 Line 32: What I want to express is that earthquakes caused by human factors, such as housing collapse, will not cause the TIR anomalies, so the earthquakes with depth=0 will be excluded. I will search the related papers online and add the reference, if there is no such papers I will delete this sentence.

Page 6 formula (11) and (12): yes, they are the same. But, they have different meaning. In (5) and (6) the left of these formulas are the final calculated values used for finding the ALICE. And the (11) and (12) is an iterative process, the values will be changed with the iterative process, in other words, the A in formula (11) and (12) are different. If it is necessary I can change the A for A` in formula (11) and (12).

Page 7 Line 14: the reference will be added.

Page 9 Line 3: Period B is 2008.01 to 2018.03, and in Page 9 Line 3, it should be "Period A" not "Period B", I will correct it.

Page 9 Line 9: It is an orange rectangle, I will correct it.

Page 12 Line 31: the standard identification I have shown in chapter 3.3, these rules are quoted form "Long-Term RST analysis of anomalous TIR sequences in relation with earthquakes occurred in Greece in the period 2004-2013".

Page 13 Fig.7 and Fig.8: Sorry for my unclear expression. There is little difference between in the distribution of clouds, but what I want to show is the relationship between the earthquakes distribution and clouds distribution. In period C, many earthquakes happened in the position that always blocked by clouds.

Page 14 Fig.9: Yes I have also read some papers that indicate the TIR anomalies before Wenchuan earthquakes, but may use different data, different method and different identification rules for TIR anomalies. In this paper, I do not mean that there is no TIR anomalies before Wenchuan Earthquake, and I cannot give the results that there is or there is not a TIR anomaly corresponding to some earthquake. This article is only to evaluate and study whether the results extracted by RST with use of MODIS LST data are effective.

Page 16 Line 37: The expression in my paper is unclear. I did not negate the validity of Thermal Infrared for earthquake prediction studies, nor did I negate the validity of this method. I just want to say things based on the results, the prediction ability of RST method and MODIS data in Sichuan area is limited, it may be caused by weather, data, identification of TIR anomalies or others.

---

## Author Comment (AC2) · 9 Oct 2018

Many thanks for your comments.

Firstly, I will rewrite it in English and invite a native speaker to polish my writing. Thanks for your suggestion.

Fig.1: I will make Fig.1 more clearly.

Page 3: I will make the description clearer and more concise.

Page 3: "scope" is the "range", I will modify the expression.

Page 4 Line 1: I will delete the repeated information.

Page 4 Line 30-35: I will modify the expression.

Page 4 Line 42: I will add the year.

Page 6 RETIRA: Sorry for that, I have written it wrong, I used ALICE but no RETIRA, I will correct it. And as you have mentioned, there is some confusing use of ALICE and RETIRA in paper, I will check them carefully and correct the wrong places.

Page 7: I will rewrite the conditions.

Page 7 line 29: "one of 5) and 6). There are TIR anomalies, and these TIR anomalies correspond to no earthquakes.

Page 7 Line 30: And the other TIR earthquakes correspond to no TIR anomalies.

Page 8 First Paragraph: I will consider your suggestion.

Fig.3: I will consider your suggestion.

Fig.4: I will rewrite the caption.

Page 13 Line 12: it means "It is not good, it cannot fully describe the prediction ability".

Section 4.3: I will try to add some fig or others to make it more easy to understand.

Question: The criterion for determining TIR anomalies is relatively subjective, and there is no universal standard to judge whether there is TIR anomalies. For instance, $R_I$ values have been classified as 'anomalous' pixels for different threshold: >2.0, >2.5, >3.0 and >3.5(Tramutoli, Cuomo et al. 2005) ; $\geq 2.0$ and $\geq 3.0$ (Genzano, Aliano et al. 2007); $\geq 2.0$, $\geq 2.5$ and $\geq 3.0$ (Pergola, Aliano et al. 2010). In ELEFTHERIOU's study, the threshold was set to 4, however, from the point of view of mathematical statistics, when the value is greater than two times the standard deviation, it has already belonged to the abnormal category, so in this study, the threshold is set to 2.

Genzano, N., C. Aliano, C. Filizzola, N. Pergola and V. J. T. Tramutoli (2007). "A robust satellite technique for monitoring seismically active areas: The case of Bhuj–Gujarat earthquake." **431**(1): 197-210.

Pergola, N., C. Aliano, I. Coviello and C. Filizzola (2010). "Using RST approach and EOS-MODIS radiances for monitoring seismically active regions: a study on the 6 April 2009 Abruzzo earthquake." Natural Hazards & Earth System Sciences **10**(2): 239-249.

Tramutoli, V., V. Cuomo, C. Filizzola, N. Pergola and C. J. R. S. o. E. Pietrapertosa (2005). "Assessing the potential of thermal infrared satellite surveys for monitoring seismically active areas: The case of Kocaell (Izmit) earthquake, August 17, 1999." **96**(3): 409-426.

---

## Author Comment (AC3) · 9 Oct 2018

Thanks for your comments. The supplement is my reply.

Please also note the supplement to this comment:
https://www.nat-hazards-earth-syst-sci-discuss.net/nhess-2018-214/nhess-2018-214-AC3-supplement.pdf
* * *

---

## Author Comment (AC4) · 11 Oct 2018

Sincerely thanks for your suggestions and questions.

I do have learned a lot from your comments.

Firstly, I am sorry that I have expressed my object of this paper unclearly. I am not going to evaluate whether RST is valid for earthquake prediction, or if the MODIS LST data is suitable for the extraction of TIR anomalies. I just want to study and evaluate prediction ability of a certain method (RST) applied in a certain area (SIchuan) with use of a certain data (MODIS). If the PPV and TPR are high, it indicates that the prediction ability of this method applied in this area with this data is very well. But if the PPV and TPR is low, they indicate that the prediction ability is limited. The limited prediction ability may be caused by cloud, lack of data, method effectiveness or precision of the data, and, what is the key cause, that has not been studied in this paper. I will check my article carefully and modify the inappropriate, confusing sentences and words. .

Main comments 1,2: Whole paper needs to be rewritten in a better English. In particular some sections, like the paragraph 3.2 (RST methodology) or paragraph 5 (Discussion), are not clear and only after various readings the paper can be understood. The major part of citations should be revised, both in the form (sometime given name is used instead of surname like at line 6 of page 2) and in content (some wrong citations have been used or some important citations miss, as for RETIRA index).

R: The major part of citations should be revised, both in the form (sometime given name is used instead of surname like at line 6 of page 2) and in content (some wrong citations have been used or some important citations miss, as for RETIRA index).

I will rewrite my paper in English and invite a native speaker to help me polish it. And I will check my paper and revise the citations.

Main comments 3: To identify thermal anomalies possibly related to impending earthquakes, you used LST (Land Surface Temperature) products retrieved by the radiance collected by MODIS sensor on board of the polar satellites EOS/AQUA and EOS/TERRA.

Taking in mind that Authors who proposed the RST approach shown the advantages offered by the use of sensors onboard of geostationary platforms instead of sensor onboard of polar satellite packages (see the paper Filizzola, C., N. Pergola, C. Pietrapertosa, and V. Tramutoli (2004), Robust satellite techniques for seismically active areas monitoring: a sensitivity analysis on September 7, 1999 Athens's earthquake, Phys. Chem. Earth, 29(4–9), 517–527, doi:10.1016/j.pce.2003.11.019), since 2004 the major part of RST applications to thermal monitoring of seismogenic areas have been carried out using TIR satellite records acquired by sensors onboard of geostationary satellites (as also you have reported in the your paper). Now, my question is why you prefer to use EOS/MODIS data instead that TIR records collected by sensor on geostationary platforms (e.g. the Japanese MTSAT satellite)?

Moreover, LST (Land Surface Temperature) products have been take in account. LST products are very useful to reduce variability of atmospheric water vapor, but in the computation of LST several approximations are necessary (e.g. emissivity, total water vapour content, ecc.), which should produce errors (also of 4-5 K degree) in the satellite LST estimations. Taking in mind, that thermal anomalies possibly related to seismic activity are of low intensity, wrong LST estimation could mask and/or generate false anomalies. Have you an idea of the impact of this errors on the your

analysis?

R: Firstly, we can compare the MODIS with other kinds of thermal infrared satellite receiving instruments (GMS-5 and AVHRR), in the following table some characteristics have been shown.

**Table 1 The characteristics of three kinds of satellite thermal infrared data receiving instruments.**

| Parameter | GMS-5 | AVHRR | MODIS |
|---|---|---|---|
| Spatial resolution/km | 5 | 1 | 1 |
| Bit | 8 | 10 | 12 |
| Bands | 4 | 5 | 36 |
| Thermal Infrared Bands | 2 | 3 | 6 |
| Band Width | | 100 | 20 |
| Scan method | Line-scanning | Line-scanning | CCD-scanning |
| SNR | | 9~20 | >=500 |

From the table, it can be concluded that MODIS outperforms other sensors in terms of spatial resolution, data accuracy and Thermal infrared bands. Moreover, it can provide global daily nighttime data which is also suitable for TIR anomalies extraction.

Secondly, some researchers have published papers about the TIR anomalies before earthquakes with use of MODIS LST data or other LST data. D. Ouzounov found evidence for a thermal anomaly LST pattern that is apparently related to pre-seismic activity with use of MODIS LST data(Ouzounov and Freund 2004). And that is a strong evidence to prove that the MODIS LST is also can be used for TIR anomalies study. Moreover, other researchers have also conducted the studies with t MODIS LST data(Choudhury 2005, Panda, Choudhury et al. 2007, Pergola, Aliano et al. 2010). So, I think that MODIS LST data is also suitable for this study.

In this paper, the structure is intact, and to compare different data extraction TIR anomalies is not the purpose, which can be conducted in the coming research.

Main comments 4: Earthquake catalogue (China Seismic Information; http://www.csi.ac.cn/) used to verify possible correlation with TIR anomalies is inaccessible. Please provide a correct URL. Anyway, consulting a different seismic catalogue, i.e. UGSG catalogue (https://earthquake.usgs.gov/earthquakes/search/), using a similar criteria (M≥3.5; Depth >0; region from 25°N to 40°N and from 95°E to 110°E; time since August 1, 2002 up to April 15, 2018) I found 2369 earthquakes, respect to 3615 seismic events reported in the your paper. A comparable numbers of seismic events, i.e. 3828, is obtained when the USGS catalogue is consulted starting by 1965. Have you use seismic data from 2002 or from 1965? In the first case (i.e. 2002) how you explain this difference (2369 vs 3615)? In the second case (i.e. 1965), because MODIS data are available since 2002, how is possible found some relations among TIR anomalies and earthquakes (before 2002)? In this last case, please provide a correct analysis.

R: http://www.csi.ac.cn is the Chinese Official website to publish the Earthquake information. And I am sorry for that I cannot get on this website either and I do not know why. But I have saved the data downloaded from that website, and I will send it to you in the attachment. I will also search the data on USGS to confirm if they are different. I will answer you this question later.

Main comments 5: About the performed correlation analysis among the appearances of TIR anomalies and earthquake occurrences, you should be in mind that working in the optical band, a wide presence of meteorological clouds, as well as the lack of satellite data, do not allow to give continuity to the observations, which is necessary to identify possible TIR anomalies or to fully appreciate a possible space–time persistence of previously occurred TIR anomalies, producing in this way a possible overestimation of missed events. Please, consider this suggestion and provide a more convincing analysis. As consequence also your conclusions should be reconsidered.

R: A wide presence of meteorological clouds, lack of satellite data surely will influence the TPR or FNR. However, I think we should not to remove the earthquakes influenced by clouds or lack of data. I think this analysis is a simulated forecast, the prediction ability is to evaluate how many earthquakes can or cannot be predicted with this method and data. Surely, earthquakes are unable to be predicted when there is large wide of clouds, but in other words, this is also a great limitation and defect of the RST method or Thermal Infrared data, and how to solve these problems is the key to enhance the ability of earthquake prediction. Some earthquakes cannot be predicted by this method is the truth, we should not to avoid or remove them, on the contrary, the earthquakes correspond to no TIR anomalies should be counted in missed rate. But the high missing rate does not mean that the RST or MODIS data are not valid for earthquake prediction study, it reflects some limitations of this method and data in other aspects.

Specific comments

Page 1 - Lines 18-19; In the abstract, you announced that a refined RST data analysis and Robust Estimator of TIR Anomalies (RETIRA) index were used, but in the text I have not read any new improvements to the RST methodology, if not those reported in Eleftheriou et al. (2016). Otherwise please explain better the refinements made to the RST technique. Moreover, add the reference of RETIRA index.

R: sorry for that, there is no refinement for the RST, but the statistical method, I add PPV, FDR and FNR to evaluate the earthquakes prediction ability. And I will add reference.

P1 - L25; Please provide the complete name of PPV, FDR, TPR and FNR.

R: I will add the full name of PPV, FDR, TPR and FNR.

P1 - L26; The sentence "the prediction ability of RST in Sichuan area is limited" is too strong!

R: I will change the sentence. "The prediction ability of RST with use of MODIS in Sichuan area is limited."

P4 - L34-36; I not understand the sense of this sentence "Moreover, Tronin indicated that the anomaly was sensitive to crustal 1000kmearthquakes with a magnitude more than 4.7 and for distance of up to 1000km" in this position.

R: That is the reason why I choose earthquakes with M>=3.5, because those earthquakes have covered the earthquakes with M>=4.7.

P5 - L14-24; Cloudy pixel, as well as pixels declared as edge clouds, should be exclude before the computation of $\Delta V(r,t)$ otherwise effects due to cloudiness are not removed and false TIR anomalies

could be generate.

R: yes, I have eliminated the cloudy pixels and the edge clouds.

P5 - L25-31; How you identify the extreme weather events (e.g. blizzard)?

R: two ways. 1. The kσ-clipping processing can eliminate some influence of extreme weather. 2. By using satellite cloud map, wind direction map and meteorological knowledge, the meteorological data in this area are analyzed. For example, there is wide blizzard in this area from Jan. 2008 to Mar. 2008.

P6 - L9-13; The reference Saraf et al. (2009) is correct? I not found no mention about effects of cloudy pixels on ΔV in this publication.

R: Accepted, I have given the wrong reference, I will correct it.

P6 - L13-20; *Cold spatial average effect as reported in* Aliano et al. (2009), Genzano et al. (2010) and Eleftheriou et al. (2016) could affect the whole TIR scene. Rightly you have take in account this effect, but in opposite way of Genzano at al. (2015) or Eleftheriou et al. (2016) you work at pixel level instead of whole scene level. In this way, the above mentioned effect could be not removed in the computation of reference fields.

R: Maybe I have expressed it right, in this paper, when the cloudy fraction of the land portion of the scene is >80%, all the pixels in this scene will be removed from the calculation of background field. Maybe the formula 10) is confusing.

P6 - L32-35; The sentence "This process should be paid more attention, because in the past papers, this process is always ignored." is wrong. In all applications of RST approach, kσ-clipping method (always applied) guarantee to remove outlier (i.e. extreme events) from the computation of reference fields. Please, consider to rewrite better the sentence.

R: I will delete this sentence.

P6-P7 (Change detection step); Although you have announced the computation of ALICE index, the index reported in the equation 13 should be the RETIRA index (correct equation can be found in Filizzola et al. 2004; Tramutoli et al. 2005). Please, correct it.

R: I will check my paper and correct the confusing use of ALICE and RETIRA.

P7 L9-23; The criteria used to identify TIR anomalies are the same introduced for the first time in Genzano et al. (2015) and in Eleftheriou et al. (2016) in order to indentify Significant Sequences of TIR Anomalies (SSTAs). Please consider to call it in similar way, mentioning these two publications. Moreover, starting from a mathematical point of view you have consider to set a threshold K equal 2, if you have a normal distribution (Gaussian) a 2 times the standard deviation could be sufficient to identify anomalies. In addition, RETIRA index (as well as ALICE index) give the possibility to evaluate in term of SIgnal-Noise ratio (S/N) the intensity of anomalies (see Tramutoli at al. 2001 or Tramutoli et al. 2005 for more details). Please, take in mind this suggestions when choose threshold k.

R: I will rewrite the criteria, and many thanks for your advice to choose the threshold K.

P9 L3; Period B is the same of period A (i.e. from2002.09 and 2007.12)?

R: Period B is from 2008.01 to 2018.03, I will correct it.

P10 Fig.4; Figure shown are not a good example of TIR anomalies possibly associated to earthquakes. In the example on the right part (TIR map of 2010/10/22) earthquakes seems not satisfy the rules announced in chapter 3.3. Moreover, to show the whole sequence of TIR anomalies, not only one day with TIR anomalies, could help the reader to better understand the concept of Significant Sequence of TIR Anomalies.

R: I will make changes according to your opinion.

P11 Fig.5; As reported in the caption "The cells in the blue rectangle mean that this day is affected by a large area of clouds, ...", now, some days with TIR anomalies belonging to several sequences of TIR anomalies (i.e. 2, 10, 27, 32, 35, 36, 37, 45, 50, 58, 59) are affected by a wide cloudy coverage, all this lets thinks that TIR anomalies due to meteorological effects are not removed from the analysis (as suggested in Eleftheriou et al., 2016).

R: Yes, after the carefully examining, I find that I haven't eliminated the so-called TIR anomalies caused by large clouds cover. I will correct the statistical analysis in the paper, also the related contents in other parts of the paper.

P12 L21-33. Rightly, you are reported that cloudy coverage could prevent to observe with continuity the presence of TIR anomalies, this is a intrinsically limitation of satellite technologies which work in the optical band, and not of RST methodology. Please revise your sentences.

R: As what I have mentioned at the beginning of the reply, I will revise the sentence.

P13-16 Paragraph 4.3 (The evaluation of earthquake prediction ability for RST); The performed analysis not have any sense if carried out in this way. Mainly, the analysis on the rate of earthquakes which correspond ("TPR") or not ("FNR") to TIR anomalies it is very complicated to perform, because gaps in observations, due to the lack of satellite data or to a wide presence of meteorological clouds make impossible to give a continuity to the observations, which is necessary to identify possible TIR anomalies or to fully appreciate a possible space–time persistence of previously occurred TIR anomalies, as consequences the relation one to one (earthquake-TIR anomalies) that you are looking is corrupted by this limitation. Anyway, before to comment the results of a some kind of sensitivity analysis this circumstance should be announced.

R: As what I have explained at the beginning of the reply, this is simulated earthquake prediction, the limited prediction ability may be caused by   lack of data, clouds cover, method itself, the characteristics of the data, the limitation of RST or other things. No matter what is reason that makes some certain earthquakes cannot be predicted, some of earthquakes haven't been correlated with TIR anomalies is the truth, so they have to be counted in the missing rate. Our paper is not to study why these earthquakes are not corresponding to TIR anomalies, but just evaluate the prediction ability.

Another thing I want to explain that, in this study,    earthquakes and TIR anomalies are not one-to-one but many-to-many.

Choudhury, S. J. I. J. o. R. S. (2005). "Cover: Satellite detects surface thermal anomalies associated with

the Algerian earthquakes of May 2003." **26**(13): 2705-2713.

Ouzounov, D. and F. Freund (2004). "Mid-infrared emission prior to strong earthquakes analyzed by remote sensing data." Advances in Space Research **33**(3): 268-273.

Panda, S. K., S. Choudhury, A. K. Saraf and J. D. J. I. J. o. R. S. Das (2007). "MODIS land surface temperature data detects thermal anomaly preceding 8 October 2005 Kashmir earthquake." **28**(20): 4587-4596.

Pergola, N., C. Aliano, I. Coviello and C. Filizzola (2010). "Using RST approach and EOS-MODIS radiances for monitoring seismically active regions: a study on the 6 April 2009 Abruzzo earthquake." Natural Hazards & Earth System Sciences **10**(2): 239-249.

---

## Author Comment (AC5) · 15 Oct 2018

Dear Referee, I have some additions to my last reply. In "One year of RST based satellite thermal monitoring over two Italian seismic areas ", Tramutoli has also calculated the missing rate, and it is up to 67%. And I would like to send you the earthquake catalog downloaded from www.csi.ac.cn, but i cannot send you in this reply. So, would you please give me your e-mail? If you want my earthquakes catalog. And my email is zhangying2016@radi.ac.cn. Your Sincerely, Ying Zhang

---

## Author Response (AR1)

**Reviewer 1:**

**Point 1: Page 1 Line 25 in abstract – PPV, FDR, TPR and FNR are best added their full names.**

**Response 1:** The full names of PPV, FDR, TPR and FNR will be added.

**Change 1:** P1L24-This is the first time that the ability to predict earthquakes has been evaluated based on the positive predictive value (PPV), false discovery rate (FDR), true-positive rate (TPR) and false-negative rate (FNR).

**Point 2: Line 35-The "TIR" might be repeated with the "outgoing longwave satellite"**

**Response 2:** "outgoing longwave" will be deleted.

**Change 2:** P2L7-Several studies have detected space-time anomalies in TIR satellite imagery,

**Point 3: Page 3 * Add the names of the main faults in Fig.1. An overall map of China and Sichuan could also be considered.**

**Response 3:** Because there are abundant faults in the study area, the figure will be in a mass and be confusing when the names of faults are added. Moreover, the names of faults are not necessary in our study.

**Change 3:**

[Figure]

**Point 4: There are three spatial resolutions -250m, 500m, 1000m of the MODIS data, which one was used in your paper? Line 32- How did you get this conclusion-"But earthquakes caused by ground subsidence, human factors and so on will not cause TIR anomalies"?**

Response 4: The spatial resolution of our data is 5600m. And the line 32 I have deleted it in

the manuscript.

Change 4: P6L8-The LST data were retrieved at 5,600 m using the generalized

split-window algorithm. In the day/night algorithm, daytime and nighttime LSTs are

retrieved from pairs of day and night MODIS observations in seven TIR bands.

**Point 5: Page 6 * Maybe I did not understand the calculation process, but the right sides of the formula (11) and (12) are the same as the right sides of the formula (5) and (6). What's the purpose for repeating the same expression?**

Response 5: yes, they are the same. But, they have different meaning. In (5) and (6) the

left of these formulas are the final calculated values used for finding the ALICE. And the

(11) and (12) is an iterative process, the values will be changed with the iterative process,

in other words, the A in formula (11) and (12) are different. But the formula and method

are little bit confusing and hard to understand, so I changed the organization of method.

**Change 5:** P8L25

$$A(r, T) = A_1(r, t) * A_2(r, t) \qquad (9)$$

$$V_{REF}(r, \tau, T) \equiv \frac{\sum_{\forall t \in T}[\Delta V(r,t) \cdot A(r,t)]}{\sum_{\forall t \in T} A(r,t)} \qquad (10)$$

$$\sigma_{\Delta V}^2(r, \tau, T) \equiv \frac{\sum[\Delta V(r,t) \cdot A(r,t) - \mu_{\Delta V}(r,\tau)]^2}{\sum_{\forall t \in T} A(r,t)} \qquad (11)$$

**Point 6: "In ELEFTHERIOU's study" should add the reference. Line 29- "one of 5) and 5)" is "one of 5) and 6)"?**

**Response 6:** the reference will be added. And the 5) and 5) will be corrected to 5) and 6)

**Change 6:** P9L18-The RETIRA $\otimes_{\Delta V}(r, \tau, T)$ > 2. In Eleftheriou's study, the threshold was set at 4 (Eleftheriou et al., 2016) ;

P10L5-but do not satisfy at least either 5) or 6),

**Point 7: Page 9 Line 3- Period A is from 2002.09 to 2007.12, is Period B also from 2002.09 to 2007.12? Line 9- I cannot find the brown rectangle.**

**Response 7:** Period B is 2008.01 to 2018.03, and in Page 9 Line 3, it should be "Period A" not "Period B", I will correct it. And in the fig, it is an orange rectangle, I will correct it.

**Change 7:** P12L3-Further evidence is presented in Table 1, where earthquakes of M ≥ 5.0 that occurred during period A (from 2002.09 to 2007.12) are detailed.

P13L2-The orange rectangle represents the study area ($27°N$ to $37°N$, $97°E$ to $107°E$).

**Point 8: Page 12 Line 31- "Thermal anomalies are more likely to be extracted and TIR anomalies are more likely to correspond to earthquakes". Do you have a standard of the thermal anomaly? Just as the size and the amplitude of the anomaly.**

**Response 8:** the standard identification I have shown in chapter 3.3, these rules are quoted form "Long-Term RST analysis of anomalous TIR sequences in relation with earthquakes

occurred in Greece in the period 2004-2013".

**Change 8:** no changes.

**Point 9: There are 61 anomalies with period C, but 60 anomalies without period C because of the more cloud in period C in study area, however, little difference of clouds in Fig.7 and Fig.8 can be detected. Should you make the difference more obvious?**

Response 9: Sorry for my unclear expression. There is little difference between in the distribution of clouds, but what I want to show is the relationship between the earthquakes distribution and clouds distribution. In period C, many earthquakes happened in the position that always blocked by clouds.

**Change 9:** no changes.

**Point 10: Page 14 * In Fig.9, the percentage of PPV and TPR with magnitude _7.4 is 0, is there no TIR anomalies for these earthquakes? I have read some papers that there were TIR anomalies when Wenchuan 8.0 earthquake occurred. * In 4.3 The evaluation of earthquake prediction ability for RST and 5 Discussion, the description is complicated, can you simplified them slightly?**

Response 10: Yes I have also read some papers that indicate the TIR anomalies before Wenchuan earthquakes, but may use different data, different method and different identification rules for TIR anomalies. In this paper, I do not mean that there is no TIR anomalies before Wenchuan Earthquake, and I cannot give the results that there is or there is not a TIR anomaly corresponding to some earthquake. This article is only to evaluate and study whether the results extracted by RST with use of MODIS LST data are effective.

**Change 10:** no changes.

**Point 11: Page 16 Line 37- "the prediction ability of RST in Sichuan area is limited". I think this sentence is too absolute. Many earthquakes occurred in Longmenshan fault in Sichuan area had been studied by scientists, and got some anomalies before or after**

the earthquake. For example, Singh (2010) had found precursory signals using satellite and ground data associated with the Wenchuan earthquake of 12 May 2008. Zhang Yuan-sheng (2010) also detected the TBB anomalies before Wenchuan earthquake; and Zhang Xuan (2013) had analyzed the thermal infrared anomaly before the Lushan 7.0 earthquake. So, maybe the data was not suitable for this region, not simply because of the cloud. This should be described in the paper.

Response 11: The expression in my paper is unclear. I did not negate the validity of Thermal Infrared for earthquake prediction studies, nor did I negate the validity of this method. I just want to say things based on the results, the prediction ability of RST method and MODIS data in Sichuan area is limited, it may be caused by weather, data, identification of TIR anomalies or others.

Change 11: P23L10-As such, the prediction potential of RST using MODIS LST data in the Sichuan area is limited. However, it doesn't indicate that the RST is not effective for earthquake prediction, on the contrary, many other cases prove that this method is very effective for extracting TIR anomalies. The low PPV and TPR may be caused by the limitation of RST, nature of MODIS LST data, special topographic and weather background of study area, or something else.

**Reviewer 2:**

Point 1: The paper needs to be rewritten in better English as it is incomprehensible in many cases, distracting from the authors' primary contribution. I suggest the authors substantially revise, through elimination of ancillary matters (reduce the extend of the text and try to eliminate any repetitions).

Response 1: I have invited a native-speaker to help me polish the English and almost the whole paper have been modified.

Change 1: English Polishing.

**Point 2: Fig.1: The faults cannot be well discriminated. You should change the color or the width of the symbol.**

**Response 2:** I will make Fig.1 more clearly.

**Change 2:**

P5 Fig.1

[Figure]

**Point 3: Page 3, last line: What do you mean with the word "scope"?**

**Response 3:** "scope" is the "range", I will modify the expression.

**Change 3:** P5L6- As shown in Fig.1, the range of the study area is $27°N$ to $37°N$, $97°E$ to $107°E$.

**Point 4: first line: The coordinates have been given in the previous page.**

**Response 4:** I will delete the repeated information.

**Change 4:** P4L20- In this paper, RST is applied to a mountainous area in China.

**Point 5: Page 3: You repeat the same information about the study area.**

**Response 5:** I will simplify the information about the study area.

**Change 5:** P5L4-L15

The southeastern Gansu province and its neighboring regions were selected as the study area, to assess the correlation between TIR anomalies and earthquakes from September 2002 to March 2018. As shown in Fig.1, the range of the study area is $27°N \ to \ 37°N$, $97°E \ to \ 107°E$. The study region is located at the intersection of Gansu, Qinghai, and Sichuan provinces; it also includes the intersection of the northern section of the north-south seismic belt and the Kuma seismic belt. Structures in this area are complex and strong earthquakes are frequent (Yang, Zhang et al., 2002). The area is on the eastern edge of the Tibetan Plateau, belonging to the upper part of the rhombic block in the southeast Gansu province. The Xian River fault, the Longmen Shan Fault, and the Anning River fault intersect here, and the structure is Y-shaped. This type of geomorphology is widely encountered in plate tectonics, and the Longmen Shan fault in the north-northeast direction becomes a steep slope in the southeast Sichuan Basin and an erosion plateau northwest of the study area.

**Point 6: Page 4, lines 30-35: I think it would be more helpful to give a distance radius.**

**Response 6:** It is a good advice, however in this paper the latitude and longitude will make it more easy to understand and memorize.

**Change 6:** no changes.

**Point 7: Page 4, line 42: Please give the year of Eleftheriou et al publication.**

**Response 7:** I will correct it.

**Change 7:** P6L1- The RST approach is based on multi-temporal analyses of historical satellite observational datasets acquired under similar observational conditions (Eleftheriou et al., 2016).

**Point 8: Page 6, lines 8-10: "And because the blizzard, forest fire and the large area of clouds usually cause the abnormal increase or decrease in LST with a magnitude that far bigger than the change caused by earthquakes."**

Response 8: This sentence is a repeated information, and the meaning of this sentence

has been expressed separately in the preceding paper.

Change 8: This sentence has been deleted.

**Point 9: In your methodology section it is not clear whether you use RETIRA index or not. In the abstract you write: "In this paper, a refined RST data analysis and Robust Estimator of TIR Anomalies (RETIRA) index were used to extract the TIR anomalies from 2002 to 2018 in Sichuan 20 area with use of Moderate-resolution Imaging Spectro-radiometer (MODIS) LandSurface Temperature (LST)," but I cannot find any reference to RETIRA index in the main text.**

Response 9: I will add the reference of RETIRA.

Change 9: P9L3-The RETIRA (Robust Estimator of TIR Anomalies, Filizzola, 2004) must

be computed,

**Point 10: I think that RST methodology must be re-written in order to become clear where the already known RST RETIRA methodology stops and how your refinements have been implemented.**

Response 10: Sorry for my inaccurate expression, I have not done a refinement for RST,

and the RST methodology have been rewritten.

Change 10: P6 Chapter3.2

**Point 11: Page 6, lines 37-38: The sentence is : : :.strange???**

Response 11: I will correct it.

Change 11: P9L3-The RETIRA (Robust Estimator of TIR Anomalies, Filizzola, 2004) must

be computed, and the bigger the absolute value is, the more evident the anomaly is.

$\otimes_{\Delta V}(r,\tau,T)$ is the RETIRA of location r at time $\tau$, which belongs to the time series T.

**Point 12: Eleftheriou et al., 2016 applied the RST RETIRA index and not the ALICE index. As far as I know, these are two different indexes. In pages 6 and 7 you mention ALICE.**

**Response 12:** I have made a mistake, what I use in this paper is RETIRA and there is no

ALICE, I will correct it.

**Change 12:** All the ALICE have been replaced by RETIRA

**Point 13: Page 7: The conditions need to be refined using better syntax.**

**Response 13:** The conditions have been modified.

**Change 13:**

P9L18

1) The RETIRA $\otimes_{\Delta V}(r,\tau,T)$ > 2. In Eleftheriou's study, the threshold was set at 4 (Eleftheriou et al., 2016) ; however, from a statistical perspective, when the value is greater than two times the standard deviation, it already falls within the abnormal category. In this study, therefore, the threshold is set at 2.

2) The $V(r,t)$ is not blocked by clouds or affected by other factors.

3) Spatial persistence: The TIR anomalies cluster together and are not isolated, being part of a group covering at least 150 $km^2$ within an area of $1° * 1°$ (400 pixels in the images).

4) Temporal persistence: At least one more TIR anomaly appears within 7 days after the first TIR anomaly.

5) The TIR anomalies appear 30 days before or 15 days after the $eq(r,t)$ (Eleftheriou et al., 2016) .

6) The shortest distance from a given point in the TIR anomalies group to the epicenter of $eq(r,t)$ is less than $R_D = 10^{0.43M}$.

**Point 14: Page 7, line 29: Please correct the sentence: one of 5) and 5) mean that there are TIR anomalies but no corresponding earthquake.**

**Response 14:** The sentence will be corrected.

**Change 14:** P10L6- but do not satisfy at least either 5) or 6),

**Point 15: Page 7, line 30: What do you mean? Please explain.**

**Response 15:** The sentence has been modified.

**Change 15:** P10L6- There are also cases wherein no TIR anomalies occur.

**Point 16: Page 8, first paragraph: Please refine your English or exclude this paragraph. In my opinion, Fig. 2 is enough. Fig. 3: Its caption needs refinement. I propose to move this fig and the first two paragraphs of page 10 in the STUDY AREA Section.**

**Response 16:** I will refine this paragraph, and thanks for your advice, but I think it is necessary to summarize the Fig.2. The caption of Fig.3 will be refined, and thanks for your advice, but the purpose of the Fig.3 is to show that many earthquakes are gathering in certain regions, and the Study area section focused on the distribution of faults and the geologic background.

**Change 16:**

P10L19- First, the temporal distribution shows that the seismicity from 2002 to 2018 was most active in 2008, and that it increased in frequency and violence from that time. The bottom of Fig. 2 indicates that there were 3,615 earthquakes in the study area($3.5 \leq M \leq 4$, 2,262; $4 \leq M \leq 5$, 1,124; $5 \leq M \leq 6$, 198; $6 \leq M \leq 7$, 26; and $7 \leq M \leq 8$, 5. Therefore, the study area is characterized by severe seismic activity. As may be seen from the upper part of Fig. 2, the average earthquake frequency during period A (from 2002.09 to 2007.12) was around 78. However, the total number of earthquakes in 2008 increased to 981 including the May 12 2008 Ms 8.0 Wenchuan Earthquake, the most

serious earthquake in China in recent years. Although the frequency decreased substantially after 2008 (the average frequency during this period was 243), it remained much higher than it had been during period A. The temporal distribution indicates that seismic activity prior to 2008 had been relatively weak, but in 2008, the seismic activity was extremely intense and reached its peak. After 2008, seismicity in this area continued to maintain this intensity.

Fig. 2 The temporal distribution from 2002.09 to 2018.03 of earthquakes with $M \geq 3.5$ in the study area, and the distribution of seismic frequency with earthquake magnitude.

Fig. 3 The spatial distribution of earthquakes in the study area. The orange rectangle represents the study area ($27^\circ N \ to \ 37^\circ N$, $97^\circ E \ to \ 107^\circ E$). Earthquakes beyond the parameters of the study area are shown because earthquakes close to the study area may also cause TIR anomalies within the study area.

**Point 17: Fig. 4: Please, rewrite the caption of the Figure.**

**Response 17:** The caption will be rewritten.

**Change 17:**

Fig .4 Two examples of the correlation between TIR anomalies and earthquakes: on the left is the TIR anomaly recorded on 2006.12.29 that corresponded to two earthquakes, and on the right is the TIR anomaly recorded on 2010.10.22 that did not correspond to earthquakes.

**Point 18: Page 13, line 12: " : : :.is far from enough". I don't understand. Is it good? Is it Very good? or the opposite?**

**Response 18:** It is not good, and I will rewrite this sentence.

**Change 18:** P18L8- With the aim of evaluating the earthquake prediction potential of RST using MODIS LST data for the Sichuan area, the true-positive rate (TPR) of correspondence between TIR anomalies and earthquakes with $M \geq 4.0$ alone is insufficient. Therefore, four types of data are incorporated, with four types of ratio calculated as follows:

**Point 19: Section 4.3: (The evaluation of earthquake prediction ability for RST). It is very complicated. I think that you must make this section more attractive and easy to be read.**

**Response 19:** I will refine this section.

**Change 19:** Chapter 4.3.

**Point 20: Page 19: "Conclusions" and not "Conclusion".**

**Response 20:** It will be corrected.

**Change 20:** P26L9- Conclusions

**Point 21: And of course one simple question: Many researchers and among them Eleftheriou et al., use to call "thermal anomaly" the pixels with 4__Îd'(x,y) > 4. The fact that you selected a threshold equal to 2 instead of 4, makes the results comparable?**

**Response 21:** The criterion for determining TIR anomalies is relatively subjective, and there is no universal standard to judge whether there is TIR anomalies. For instance, $R_I$ values have been classified as 'anomalous' pixels for different threshold: >2.0, >2.5, >3.0 and >3.5(Tramutoli, Cuomo et al. 2005) ; $\geq 2.0$ and $\geq 3.0$ (Genzano, Aliano et al. 2007); $\geq 2.0$, $\geq 2.5$ and $\geq 3.0$ (Pergola, Aliano et al. 2010). In ELEFTHERIOU's study, the threshold was set to 4, however, from the point of view of mathematical statistics, when the value is greater than two times the standard deviation, it has already belonged to the abnormal category, so in this study, the threshold is set to 2.

Genzano, N., C. Aliano, C. Filizzola, N. Pergola and V. J. T. Tramutoli (2007). "A robust satellite technique for monitoring seismically active areas: The case of Bhuj–Gujarat earthquake."   431(1): 197-210.

Pergola, N., C. Aliano, I. Coviello and C. Filizzola (2010). "Using RST approach and EOS-MODIS radiances for monitoring seismically active regions: a study on the 6 April 2009 Abruzzo earthquake." Natural Hazards & Earth System Sciences 10(2): 239-249.

Tramutoli, V., V. Cuomo, C. Filizzola, N. Pergola and C. J. R. S. o. E. Pietrapertosa (2005). "Assessing the potential of thermal infrared satellite surveys for monitoring seismically active areas: The case of Kocaell (Izmit) earthquake, August 17, 1999."   96(3): 409-426.

**Change 21:** No changes.

**Reviewer 3:**

Sincerely thanks for your suggestions and questions.
I do have learned a lot from your comments.

**Main comments 1,2: Whole paper needs to be rewritten in a better English. In particular some sections, like the paragraph 3.2 (RST methodology) or paragraph 5 (Discussion), are not clear and only after various readings the paper can be understood. The major part of citations should be revised, both in the form (sometime given name is used instead of surname like at line 6 of page 2) and in content (some wrong citations have been used or some important citations miss, as for RETIRA index).**

**Response for main comments 1,2:** The major part of citations should be revised, both in the form (sometime given name is used instead of surname like at line 6 of page 2) and in content (some wrong citations have been used or some important citations miss, as for RETIRA index).

I will rewrite my paper in English and invite a native speaker to help me polish it. And I will

check my paper and revise the citations.

Changes for main comments 1,2: I have asked a professional English editing service to polish our manuscript. The citations have been revised .

**Main comments 3: To identify thermal anomalies possibly related to impending earthquakes, you used LST (Land Surface Temperature) products retrieved by the radiance collected by MODIS sensor on board of the polar satellites EOS/AQUA and EOS/TERRA.**
**Taking in mind that Authors who proposed the RST approach shown the advantages offered by the use of sensors onboard of geostationary platforms instead of sensor onboard of polar satellite packages (see the paper Filizzola, C., N. Pergola, C. Pietrapertosa, and V. Tramutoli (2004), Robust satellite techniques for seismically active areas monitoring: a sensitivity analysis on September 7, 1999 Athens's earthquake, Phys. Chem. Earth, 29(4–9), 517–527, doi:10.1016/j.pce.2003.11.019), since 2004 the major part of RST applications to thermal monitoring of seismogenic areas have been carried out using TIR satellite records acquired by sensors onboard of geostationary satellites (as also you have reported in the your paper). Now, my question is why you prefer to use EOS/MODIS data instead that TIR records collected by sensor on geostationary platforms (e.g. the Japanese MTSAT satellite)?**
**Moreover, LST (Land Surface Temperature) products have been take in account. LST products are very useful to reduce variability of atmospheric water vapor, but in the computation of LST several approximations are necessary (e.g. emissivity, total water vapour content, ecc.), which should produce errors (also of 4-5 K degree) in the satellite LST estimations. Taking in mind, that thermal anomalies possibly related to seismic activity are of low intensity, wrong LST estimation could mask and/or generate false anomalies. Have you an idea of the impact of this errors on the your analysis?**

**Response for main comments 3:** Firstly, we can compare the MODIS with other kinds of thermal infrared satellite receiving instruments (GMS-5 and AVHRR), in the following table some characteristics have been shown.

Table 1 The characteristics of three kinds of satellite thermal infrared data receiving instruments.

| Parameter | GMS-5 | AVHRR | MODIS |
|---|---|---|---|
| Spatial resolution/km | 5 | 1 | 1 |
| Bit | 8 | 10 | 12 |

| | | | |
|---|---|---|---|
| Bands | 4 | 5 | 36 |
| Thermal Infrared Bands | 2 | 3 | 6 |
| Band Width | | 100 | 20 |
| Scan method | Line-scanning | Line-scanning | CCD-scanning |
| SNR | | 9~20 | >=500 |

From the table, it can be concluded that MODIS outperforms other sensors in terms of spatial resolution, data accuracy and Thermal infrared bands. Moreover, it can provide global daily nighttime data which is also suitable for TIR anomalies extraction.

Secondly, some researchers have published papers about the TIR anomalies before earthquakes with use of MODIS LST data or other LST data. D. Ouzounov found evidence for a thermal anomaly LST pattern that is apparently related to pre-seismic activity with use of MODIS LST data(Ouzounov and Freund 2004). And that is a strong evidence to prove that the MODIS LST is also can be used for TIR anomalies study. Moreover, other researchers have also conducted the studies with t MODIS LST data(Choudhury 2005, Panda, Choudhury et al. 2007, Pergola, Aliano et al. 2010). So, I think that MODIS LST data is also suitable for this study.

In this paper, the structure is intact, and to compare different data extraction TIR anomalies is not the purpose, which can be conducted in the coming research.

**Main comments 4: Earthquake catalogue (China Seismic Information; http://www.csi.ac.cn/) used to verify possible correlation with TIR anomalies is inaccessible. Please provide a correct URL. Anyway, consulting a different seismic catalogue, i.e. UGSG catalogue (https://earthquake.usgs.gov/earthquakes/search/), using a similar criteria (M≥3.5; Depth >0; region from 25°N to 40°N and from 95°E to 110°E; time since August 1, 2002 up to April 15, 2018) I found 2369 earthquakes, respect to 3615 seismic events reported in the your paper. A comparable numbers of seismic events, i.e. 3828, is obtained when the USGS catalogue is**

consulted starting by 1965. Have you use seismic data from 2002 or from 1965? In the first case (i.e. 2002) how you explain this difference (2369 vs 3615)? In the second case (i.e. 1965), because MODIS data are available since 2002, how is possible found some relations among TIR anomalies and earthquakes (before 2002)? In this last case, please provide a correct analysis.

**Response for main comments 4:** http://www.csi.ac.cn is the Chinese Official website to publish the Earthquake information. And I am sorry for that I cannot get on this website either and I do not know why. And I refer to another website China Earthquake Datacenter (http://data.earthquake.cn), and the earthquake catalog is the same as before. I can attach my seismic catalog in the attachment. The catalog obtained from USGS is different from China, especially for the earthquakes with M<5.0. This is because the nearer to the epicenter, the more abundant and accurate information the stations can get, so the earthquake catalog provided by Chinese officials is more suitable for this study than USGS.

**Main comments 5: About the performed correlation analysis among the appearances of TIR anomalies and earthquake occurrences, you should be in mind that working in the optical band, a wide presence of meteorological clouds, as well as the lack of satellite data, do not allow to give continuity to the observations, which is necessary to identify possible TIR anomalies or to fully appreciate a possible space–time persistence of previously occurred TIR anomalies, producing in this way a possible overestimation of missed events. Please, consider this suggestion and provide a more convincing analysis. As consequence also your conclusions should be reconsidered.**

**Response for main comments 5:** A wide presence of meteorological clouds, lack of satellite data surely will influence the TPR or FNR. However, I think we should not to remove the earthquakes influenced by clouds or lack of data. I think this analysis is a simulated forecast, the prediction ability is to evaluate how many earthquakes can or cannot be predicted with this method and data. Surely, earthquakes are unable to be predicted when

there is large wide of clouds, but in other words, this is also a great limitation and defect of the RST method or Thermal Infrared data, and how to solve these problems is the key to enhance the ability of earthquake prediction. Some earthquakes cannot be predicted by this method is the truth, we should not to avoid or remove them, on the contrary, the earthquakes correspond to no TIR anomalies should be counted in missed rate. But the high missing rate does not mean that the RST or MODIS data are not valid for earthquake prediction study, it reflects some limitations of this method and data in other aspects.

**Point 1: Page 1 - Lines 18-19; In the abstract, you announced that a refined RST data analysis and Robust Estimator of TIR Anomalies (RETIRA) index were used, but in the text I have not read any new improvements to the RST methodology, if not those reported in Eleftheriou et al. (2016). Otherwise please explain better the refinements made to the RST technique. Moreover, add the reference of RETIRA index.**

**Response 1:** sorry for that, there is no refinement for the RST, but the statistical method, I add PPV, FDR and FNR to evaluate the earthquakes prediction ability. And I will add reference.

**Change 1:**

P9L3-The RETIRA (Robust Estimator of TIR Anomalies, Filizzola, 2004) must be computed

**Point 2: P1 - L25; Please provide the complete name of PPV, FDR, TPR and FNR.**

**Response 2:** I will add the full name of PPV, FDR, TPR and FNR.

**Change 2:** P1L24- This is the first time that the ability to predict earthquakes has been evaluated based on the positive predictive value (PPV), false discovery rate (FDR), true-positive rate (TPR) and false-negative rate (FNR).

**Point 3: P1 - L26; The sentence "the prediction ability of RST in Sichuan area is limited" is**

**too strong!**

**Response 3:** I will change the sentence. "The prediction ability of RST with use of MODIS in Sichuan area is limited."

**Change 3:** P1L26- The statistical results indicate that the prediction potential of RST with use of MODIS is limited with regard to the Sichuan region.

**Point 4: P4 - L34-36; I not understand the sense of this sentence "Moreover, Tronin indicated that the anomaly was sensitive to crustal 1000kmearthquakes with a magnitude more than 4.7 and for distance of up to 1000km" in this position.**

**Response 4:** That is the reason why I choose earthquakes with M>=3.5, because those earthquakes have covered the earthquakes with M>=4.7.

**Change 4:** No changes.

**Point 5: P5 - L14-24; Cloudy pixel, as well as pixels declared as edge clouds, should be exclude before the computation of ΔV(r,t) otherwise effects due to cloudiness are not removed and false TIR anomalies could be generate.**

**Response 5:** yes, I have eliminated the cloudy pixels and the edge clouds. And this have also been presented in this paper.

**Change 5:** No changes.

**Point 6: P5 - L25-31; How you identify the extreme weather events (e.g. blizzard)?**

**Response 6:** two ways. 1. The $k\sigma$-clipping processing can eliminate some influence of extreme weather. 2. By using satellite cloud map, wind direction map and meteorological knowledge, the meteorological data in this area are analyzed. For example, there is wide blizzard in this area from Jan. 2008 to Mar. 2008. And these works have been completed in the pre-processing and construction of background field, so this is a repeated information, I will delete it.

**Change 6:** This sentence has been deleted.

**Point 7: P6 - L9-13; The reference Saraf et al. (2009) is correct? I not found no mention about effects of cloudy pixels on ΔV in this publication.**

**Response 7:** Accepted, I have given the wrong reference, I will correct it.

**Change 7:** P7L28- As demonstrated by Aliano et al. and Genzano et al., the spatial distribution of clouds over a thermal heterogeneous scene can significantly change the value of $\Delta V$ in the cloud-free pixels(Aliano et al., 2008; Genzano et al., 2009).

**Point 8: P6 - L13-20; Cold spatial average effect as reported in Aliano et al. (2009), Genzano et al. (2010) and Eleftheriou et al. (2016) could affect the whole TIR scene. Rightly you have take in account this effect, but in opposite way of Genzano at al. (2015) or Eleftheriou et al. (2016) you work at pixel level instead of whole scene level. In this way, the above mentioned effect could be not removed in the computation of reference fields.**

**Response 8:** Maybe I have expressed it right, in this paper, when the cloudy fraction of the land portion of the scene is >80%, all the pixels in this scene will be removed from the calculation of background field.

**Change 8:** No changes.

**Point 9: P6 - L32-35; The sentence "This process should be paid more attention, because in the past papers, this process is always ignored." is wrong. In all applications of RST approach, kσ-clipping method (always applied) guarantee to remove outlier (i.e. extreme events) from the computation of reference fields. Please, consider to rewrite better the sentence.**

**Response 9:** I will delete this sentence.

**Change 9:** This sentence has been deleted.

**Point 10: P6-P7 (Change detection step); Although you have announced the computation of ALICE index, the index reported in the equation 13 should be the RETIRA index (correct equation can be found in Filizzola et al. 2004; Tramutoli et al. 2005). Please, correct it.**

**Response 10:** I have made a mistake, I used the RETIRA in this paper not the ALICE.

**Change 10:** All the ALICE have been replaced by RETIRA.

**Point 11: P7 L9-23; The criteria used to identify TIR anomalies are the same introduced for the first time in Genzano et al. (2015) and in Eleftheriou et al. (2016) in order to indentify Significant Sequences of TIR Anomalies (SSTAs). Please consider to call it in similar way, mentioning these two publications. Moreover, starting from a mathematical point of view you have consider to set a threshold K equal 2, if you have a normal distribution (Gaussian) a 2 times the standard deviation could be sufficient to identify anomalies. In addition, RETIRA index (as well as ALICE index) give the possibility to evaluate in term of SIgnal-Noise ratio (S/N) the intensity of anomalies (see Tramutoli at al. 2001 or Tramutoli et al. 2005 for more details). Please, take in mind this suggestions when choose threshold k.**

**Response 11:** Thanks for your advice, however, I think the expressions in my paper are

also OK, and they are also clear - TIR anomalies, TIR anomalies correspond to no

earthquakes.

**Change 11:** No changes.

**Point 12: P9 L3; Period B is the same of period A (i.e. from2002.09 and 2007.12)?**

**Response 12:** Period B is from 2008.01 to 2018.03, I will correct it.

**Change 12:**

P12L2-Further evidence is presented in Table 1, where earthquakes of $M \geq 5.0$ that

occurred during period A (from 2002.09 to 2007.12) are detailed. There were 229

earthquakes of $M \geq 5.0$, while the total number during period A was 24, which accounted

for 10.48% overall; the duration of period A accounted for 33.87% of the total timeframe

(i.e., period A + period B). Moreover, there were no earthquakes of $M \geq 6.5$ during period

A, but there were 14 earthquakes of $M \geq 6.5$ during period B (from 2008.01 to 2018.03).

All of this evidence indicates that seismic activity during period B was significantly more

violent and frequent.

**Point 13: P10 Fig.4; Figure shown are not a good example of TIR anomalies possibly associated to earthquakes. In the example on the right part (TIR map of 2010/10/22) earthquakes seems not satisfy the rules announced in chapter 3.3. Moreover, to show the whole sequence of TIR anomalies, not only one day with TIR anomalies, could help the reader to better understand the concept of Significant Sequence of TIR Anomalies.**

**Response 13:** I am sorry that I have made a mistake, right of the Fig.4 is the instance that

TIR anomaly do not correspond to earthquakes.

**Change 13:**

[Figure]

Fig .4 Two examples of the correlation between TIR anomalies and earthquakes: on the left is the TIR anomaly recorded on 2006.12.29 that corresponded to two earthquakes, and on the right is the TIR anomaly recorded on 2010.10.22 that did not correspond to earthquakes.

**Point 14: P11 Fig.5;** As reported in the caption "The cells in the blue rectangle mean that this day is affected by a large area of clouds, ...", now, some days with TIR anomalies belonging to several sequences of TIR anomalies (i.e. 2, 10, 27, 32, 35, 36, 37, 45, 50, 58, 59) are affected by a wide cloudy coverage, all this lets thinks that TIR anomalies due to meteorological effects are not removed from the analysis (as suggested in Eleftheriou et al., 2016).

**Response 14:** Thanks for your advice, yes I have made a mistake, and I have corrected all

the related results, figs and analysis.

**Change 14:**

Fig.5, Fig.6, Table2, Fig.9, Table3, Table4, Fig.10 and Fig.11 have been redone. And all the related contents in text have been corrected.

**Point 15: P12 L21-33. Rightly, you are reported that cloudy coverage could prevent to observe with continuity the presence of TIR anomalies, this is a intrinsically limitation of satellite technologies which work in the optical band, and not of RST methodology. Please revise your sentences.**

**Response 15:** As what I have mentioned at the beginning of the reply, I will revise the sentence.

**Change 15:**

P23L10-As such, the prediction potential of RST using MODIS LST data in the Sichuan area is limited. However, it doesn't indicate that the RST is not effective for earthquake prediction, on the contrary, many other cases prove that this method is very effective for extracting TIR anomalies. The low PPV and TPR may be caused by the limitation of RST, nature of MODIS LST data, special topographic and weather background of study area, or something else.

**Point 16: P13-16 Paragraph 4.3 (The evaluation of earthquake prediction ability for RST); The performed analysis not have any sense if carried out in this way. Mainly, the analysis on the rate of earthquakes which correspond ("TPR") or not ("FNR") to TIR anomalies it is very complicated to perform, because gaps in observations, due to the lack of satellite data or to a wide presence of meteorological clouds make impossible to give a continuity to the observations, which is necessary to identify possible TIR anomalies or to fully appreciate a possible space–time persistence of previously occurred TIR anomalies, as consequences the relation one to one (earthquake-TIR anomalies) that you are looking is corrupted by this limitation. Anyway, before to comment the results of a some kind of sensitivity analysis this circumstance should be announced.**

**Response 16:** As what I have explained at the beginning of the reply, this is simulated earthquake prediction, the limited prediction ability may be caused by lack of data, clouds cover, method itself, the characteristics of the data, the limitation of RST or other things. No matter what is reason that makes some certain earthquakes cannot be predicted, some of earthquakes haven't been correlated with TIR anomalies is the truth, so they have to be counted in the missing rate. Our paper is not to study why these earthquakes

are not corresponding to TIR anomalies, but just evaluate the prediction ability.

Another thing I want to explain that, in this study, earthquakes and TIR anomalies are not one-to-one but many-to-many.

Choudhury, S. J. I. J. o. R. S. (2005). "Cover: Satellite detects surface thermal anomalies associated with the Algerian earthquakes of May 2003." **26**(13): 2705-2713.

Ouzounov, D. and F. Freund (2004). "Mid-infrared emission prior to strong earthquakes analyzed by remote sensing data." Advances in Space Research **33**(3): 268-273.

Panda, S. K., S. Choudhury, A. K. Saraf and J. D. J. I. J. o. R. S. Das (2007). "MODIS land surface temperature data detects thermal anomaly preceding 8 October 2005 Kashmir earthquake." **28**(20): 4587-4596.

[revised manuscript text omitted]

month of September from 1995 to 2002, and; the false-positive rate (FPR) remained at zero when the earthquakes with M> > 4 or M>4.5 were considered, and the false positive rateFPR remained less thanunder 6% when thea threshold of  M > 5 iswas applied (Genzano, Filizzola et al., 2015). Tramutoli et al. studied the earthquakes with M> > 4.0 in Italythe southern Apennines in Italy's Po plain from July 2012 to June 2013 and the testing area was Italian southern Apennines, Po Plain, they found that the false positive rateFPR was lesserless than 33%%, while the missing rate is up towas as high as 67%% (Tramutoli, Corrado et al., 2015). Eleftheriou et al. studied the earthquakes that occurred in Greece in periodbetween 2004- and 2013 with use ofusing TIR images acquired bywith MSG/SEVIRI, and found that more than 93% of all identified TIR anomalies occurred in the prefixed space-time window around the time and location of occurrence of earthquakes (with M> > 4) and the, with an overall false positive rate is <FPR < 7%% (Eleftheriou, Filizzola et al., 2016). It seems that RST is an effect method to extracteffective means of extracting TIR anomalies before that occur as precursors to earthquakes, but there is no such study forhas hitherto been conducted on the Chinese mainland of China.

However, some researchersSeveral studies, however, have provedproven that some singleindividual earthquake results are unreliable. Some so-called TIR anomalies are caused by weathermeteorological anomalies, which that are not relatedunrelated to earthquakes. An instance is thatFor example, Matthew et al. studied the Gujarat (India) earthquake of 2001, and he found that the previous studystudies, which had indicated there was the presence of TIR anomalies beforeprior to the earthquake, was not reliable.were unreliable. They concluded that there was no robust evidence for the existence of LST anomalies prior to the 2001 Gujarat earthquake, and that cloud coveringcover was onea possible cause forof the anomalies (Blackett, Wooster et al., 2011). So, aAs such, rigorous rulestatistical analyses of thermal TIR anomaly judgement andanomalies over long period statistical analysisperiods are necessary.

In this paper, RST will beis applied for theto a mountainous area in China (27°N — 37°N, 97°E — 107°E) to study a mountainous area of China. The long). Long-term analysis (from SepSept. 2002 to Mar. 2018) will conformis used to verify the correspondencecorrelation between TIR anomalies and earthquakes. Based on the statistical results, the earthquake prediction abilitypotential of RST will be evaluated in this paper.

2. Study area

[Figure]

[Figure]

Comment [张1]: Because there are abundant faults in the study area, the figure will be in a mass and be confusing when the names of faults are added. Moreover, the names of faults are not necessary in our study.

[revised manuscript text omitted]

$$A_1(r, t) = \begin{cases} 0, & \text{...} \\ 1, & \text{...} \end{cases} \qquad (2)$$

➢ An outlier-mask is constructed.

This step is to determine the outliers, and these values should be excluded from the construction of the background field and the extraction of TIR anomalies.several other factors affect LST, for instance, the extreme weather (blizzard, foehne.g., blizzards or the Foehn), human activity, and forest fire. These factors across a wide range of LST data. These anomalies field and the extraction of TIR anomalies.

$$A_2(r, t, \tau) = \delta_1(r, t, \tau) * \delta_2(r, t, \tau) * \delta_3(r, t, \tau) \_\_\_$$

$$(3)A_2(r, t) = \begin{cases} 0, & \text{...} \\ 1, & \text{...} \end{cases} \qquad (3)$$

As it is shown in eq. (3), $\delta_1$, $\delta_2$, $\delta_3$ are three kinds of data that should be excluded from the construction of backfields. As demonstrated by Aliano et al. and Genzano et al., the spatial distribution of clouds over a thermal heterogeneous scene can significantly change the value of $\Delta V$ in the cloud-free pixels(Aliano et al., 2008; Genzano et al., 2009 #10). The large cloud cover area will introduce a cold spatial average effect to the computation of the RFs, so that when $V(r, t) < \mu_V \mu_V - 2 * \sigma_V$ (here, $\mu_V$ is the temporal average and the $\sigma_V$ is its

standard, these pixels' values will be excluded, Eleftheriou et al., 2016).

$$\delta_1(r, t, \tau) = \begin{cases} 0, & if \ V(r,t) - \mu_V(r,\tau,T) < -2*\sigma_V(r,t,\tau), t < \tau \\ 1, & other \end{cases} \quad (4)$$

Moreover, even where no cold spatial average effect is produced, extended cloud coverage can determine the $V(t)$ values and the values of the considered signal $\Delta V(r,t)$, scarcely representative of the actual conditions of cloud-free pixels, so that when the cloudy fraction of the land portion of the scene is $> 80\%$, all pixels must be excluded from the computation of the RFs (Eleftheriou et al., 2016).

$$\delta_2(r, t, \tau) = \begin{cases} 0, & if \ c \quad f \quad o \ l t \ p \quad o \ s \quad i > 80\% \\ 1, & other \end{cases} \quad (5)$$

$\delta_3$ is used to remove the outliers (where $k \ \ 2$), and its expression is as follows:

$$\delta_3(r, t, \tau) = \begin{cases} 1, & if \ |V(r,t) - \mu_V(r,\tau,T)| < k\sigma_V(r,t,\tau) \\ 0, & other \end{cases} \quad (6)$$

$\delta(r,t,\tau) = \delta_2\delta_1(r,t,\tau) * \delta_2(r,t,\tau) * \delta_3(r,t,\tau)$ computed using an iterative $k\sigma$-clipping technique, which begins by computing $\delta(r,t,\tau)$ based on the first determination of $\mu_V(r,\tau,T)(r,\tau,T)$ and $\sigma_V^2(r,\tau,T)$, and continues by updating their values using only space-time locations with $\delta(r,t,\tau) = 1$, as follows:

$$\mu'^2_{\Delta V}(r,\tau,T) \equiv \frac{\sum_{\forall t \in T}[\Delta V(r,t) \cdot A(r,t)]}{\sum_{\forall t \in T} A(r,t)} \quad (7)$$

$$\sigma'^2_{\Delta V}(r,\tau,T) \equiv \frac{\sum[\Delta V(r,t) \cdot A(r,t) - \mu_{\Delta V}(r,\tau)]^2}{\sum_{\forall t \in T} A(r,t)} \quad (8)$$

The process should be iterated until no further exclusions are determined, using the latest determination of $\delta$ (Tramutoli, 1998). And the final result of $\delta$ is the $A_3$ what we want.  are detailed below.

- Computing Reference Fields

  ➢ The $\mu_{\Delta V}(r,\tau,\Delta T)$ is the mean of location $r$ for time series $T$. The variance $\sigma_{\Delta V}^2(r,\tau,T)$ is applied at time $\tau$ using homogeneous historical records collected under the temporal constraint $t \in T$, $(t < )$  and the $V_R(r,\tau,\Delta T)$ is the background field.

  $A(r,T) = A_1(r,t) * A_2(r,t)$  (49)

  $$V_R(r,\tau,T) \equiv \frac{\sum_{\forall t \in T}[\Delta V(r,t) \cdot A(r,t)]}{\sum_{\forall t \in T} A(r,t)} \quad (510)$$

$$\sigma_{\Delta V}^2(r, t, T) \equiv \frac{\sum[\Delta V(r,t)\cdot A(r,t) - \mu_{\Delta V}(r,T)]^2}{\sum_{\forall t \in T} A(r,t)} \qquad (611)$$

> To compute the The outlier mask, and is then constructed; this method is aimeddesigned to eliminate the abnormal significant data caused by the non-seismic factors. FirstThe first step is to find reference fieldsthe RFs of minima $V_m^i(r,t,T)$ and maxima $V_m^i(r,t,T)$ to be used at the time .

$$\Delta V_m^i(r,t,T) \equiv min\{\Delta V(r,t_1),\cdots,\Delta V(r,t_N)\} \;\forall t \in T, A(r,T) = 1 \qquad (7)$$
$$\Delta V_m^i(r,t,T) \equiv MAX\{\Delta V(r,t_1),\cdots,\Delta V(r,t_N)\} \;\forall t \in T, A(r,T) = 1 \qquad (8)$$

Wherewhere a modified outlier mask $A_d(r,t) = A(r,T) \cdot \delta(r,t,t)$ havehas been introduced in order to avoid contributions from accidental minima or maxima. And because the blizzard, since blizzards, forest firefires, and the large areaareas of cloudscloud cover usually cause the abnormal increaseincreases or decreasedecreases in LST with a magnitude that far bigger thanexceeds the changechanges caused by earthquakes. These data should not be included forin the calculation of reference fieldthe RF. As showndemonstrated by ALIANOAliano et al. and GENZANOGenzano et al., the spatial distribution of clouds, over a thermal heterogeneous scene, can significantly change the value of $\Delta V$ in the cloud-free pixels (Saraf, Rawat et al., 2009). The large cloud cover area of clouds will bringintroduce a cold spatial average effect to the computation of reference fieldsthe RFs, so that when $V(r,t) < \mu_V \mu_V - 2 * \sigma_V$ (here, $\mu_V$ is the temporal average and the $\sigma_V$ is its standard), the value of these pixels pixels' values will be excluded (Eleftheriou, Filizzola et al., 2016). Moreover, even if not producing awhere no cold spatial average effect, an is produced, extended cloud coverage can determine values ofthe V(t) values and thenthe values of the considered signal $\Delta V(r,t)$, scarcely representative of the actual conditions of cloud-free pixels, so that when the cloudy fraction of the land portion of the scene is $> 80\%$, then all the pixels have tomust be excluded from the computation of reference fieldsthe RFs (Eleftheriou, Filizzola et al., 2016).

The termterms $\delta_1(r,t,\tau)$ a $\delta_2(r,t,\tau)$ are defined by the expression:

$$\delta_1(r,t,\tau) = \begin{cases} 0, i \quad V(r,t) - \mu_V(r,t,T) < -2 * \sigma_V(r,t,t) \\ 1, o\ he \end{cases} \qquad (9)$$
$$\delta_2(r,t,\tau) = \begin{cases} 0, i \;\epsilon \quad f \quad e \; l \; p \quad e \; s \quad i > 80\% \\ 1, o\ he \end{cases} \qquad (10)$$

For $\delta_1$ is used to remove the outliers caused by the forest fire orfires, blizzards and, or other factors, $\delta_2$ is used to remove them (with (where k >= 2,; in this study the, k is set to beat 4), and

its expression is as follows:

$$\delta_{\Delta V}(r,t,\tau) = \begin{cases} 1, & |V(r,t) - \mu_V(r,t,T)| < k\sigma_V(r,t,\tau) \\ 0, & other \end{cases}$$

$\delta(r,t,\tau) = \delta_{\Delta V}\delta_{\Delta V}(r,t,\tau) * \delta_{\Delta V}(r,t,\tau) * \delta_{\Delta V}(r,t,\tau)$ computed by using an iterative $k\sigma$-clipping technique, which startsbegins by computing $\delta(r,t,\tau)$ based on the base of the first determination of $\mu_V(r,t,T)(r,t,T)$ and $\sigma_V^{\Delta V}(r,t,T)$, and continues by updating their values by using only space/ time locations with $\delta(r,t,\tau) = 1$, as follows:

$$\mu_{\Delta V}^{\Delta V}(r,t,T) \equiv \frac{\sum_{v \in t}[\Delta V(r,t) \cdot A(r,t)]}{\sum_{v \in t} A(r,t)} \quad (11)$$

$$\sigma_{\Delta V}^{\Delta V}(r,t,T) \equiv \frac{\sum[\Delta V(r,t) \cdot A(r,t) - \mu_{\Delta V}(r,t)]^{\Delta V}}{\sum_{v \in t} \frac{}{A(r,t)}} \quad (12)$$

The process should be iterated until no further exclusions, are determined by the use of, using the latest determination of δ (Tramutoli, 1998).This process should be paidafforded more attention, becauseas it has been overlooked in the past papers, this process is always ignoredprevious studies.

- Change-detection step

- Change-detection

  ➢ To compute theThe RETIRA (Robust Estimator of TIR Anomalies, Filizzola, 2004) Absolutely Local Index of changeChange of the Environment (ALICE,)) must be computed, and the bigger the absolute value is, the more obviousevident the anomaly is. $\otimes_{\Delta V}(r,\ddagger,T)$ is the ALICERETIRA of location $r$ at time $t$, which is belongedbelongs to the time series $T$.

  $$\otimes_{\Delta V}(r,\ddagger,T) = \frac{[\Delta V(r,\ddagger) - V_{REF}(r,\ddagger,T)]}{\dagger_{\Delta V}(r,\ddagger,T)} \quad (1312)$$

  ➢ To determine whether theWhether $\otimes_{\Delta V}(r,\ddagger,T)$ is affected by cloud. In should be determined. From the results, it can bemay easily be concluded that some area in certain timeareas will lack of data at certain times, and for these situations we have to implementscenarios a special value to informmust be implemented to indicate that thisthese data isare affected by clouds and it should not be analyzed inexcluded from the

ensuing analyses.

**3.3 Identification of TIR anomalies**

After the calculation of $\otimes_{\Delta V}(r,\ddagger,T)$ , the next step is to identify the TIR anomalies and  correlate them with earthquake occurrences. In this paper, a $\otimes_{\Delta V}(r,\ddagger,T)$  that exceeds the threshold indicates the presence of a TIR anomaly; further conditions will be applied to confirm the correlation . For $\otimes_{\Delta V}(r,\ddagger,T)$  and $e(r,t)$, only in those cases where the following conditions are satisfied can it be concluded that the TIR anomaly is related to $e(r,t)$:

1)  The RETIRA $\otimes_{\Delta V}(r,\ddagger,T) > 2$. In Eleftheriou's study, the threshold was set at 4 (Eleftheriou et al., 2016) ; however, from a statistical perspective, when the value is greater than two times the standard deviation, it already falls within the abnormal category.  this study, therefore, the threshold is set at 2.

2)  The $V(r,t)$ is not  blocked by clouds or affected by other factors.

3)  Spatial persistence The TIR anomalies cluster together and  are not isolated being part of a group covering at least 150 $k^2$ within an area of $1° * 1°$ (400 pixels in the images).

4)  Temporal persistence : At least one more TIR anomaly appears within 7 days after the first TIR anomaly .

5)  The TIR anomalies appear 30 days before or 15 days after the $e(r,t)$ (Eleftheriou et al., 2016) .

6)  The shortest distance from a given point in the TIR anomalies group to the epicenter of $e(r,t)$ is less than $R_D = 10^{0.4 M}$.

Where the TIR anomalies satisfy conditions 1), 2), 3) and 4), but do not satisfy at least either 5) or 6), TIR anomalies are present with no corresponding earthquake. There are also cases wherein no TIR anomalies

occur.

4. Results and analysis

A comprehensive statistical analysis and the TIR extraction results are detailed in this chapter. In chapter 4.1, a statistical analysis is conducted to ascertain the basic seismological conditions in the study area. , while the statistical results for the correlation between earthquakes and TIR anomalies are presented and analyzed in chapter 4.2. Finally, an analysis of the earthquake prediction potential of RST is presented in chapter 4.3.

4.1 Statistical analysis of earthquake activity in the study area

Prior to investigating the correlation between TIR anomalies and earthquakes, a simple analysis of the temporal and spatial characteristics of the earthquakes is required.

First, the temporal distribution shows that the seismicity  from 2002 to 2018 was most active in 2008, and  it increased in frequency and violence from that time. The bottom of Fig. 2 indicates that there were 3,615 earthquakes in the study area (3.5 ≤ M ≤ 4 , 2,262; 4 ≤ M ≤ 5 , 1,124; 5 ≤ M ≤ 6  198; 6 ≤ M ≤ 7  26; and 7 ≤ M ≤ 8  5. Therefore, the study area is characterized by severe seismic activity. As may be seen from the upper part of Fig. 2, the average earthquake frequency during period A (from 2002.09 to 2007.12) was around 78. However, the total number of earthquakes in 2008 increased to 981 including the May 12 2008 Ms 8.0 Wenchuan Earthquake,  the most serious earthquake in China in recent years. Although the frequency decreased substantially after 2008 (the average frequency during this period was 243), it remained much higher than it had been during period A. The temporal distribution indicates that  seismic activity prior to 2008 had been relatively weak, but in 2008, the seismic activity was extremely intense and reached its peak . After 2008 , seismicity in this area continued to maintain this intensity.

[Figure]

Fig. 2 The temporal distribution from 2002.09- to 2018.03 of earthquakes with M ≥ 3.5 in the study area, and the distribution of seismic frequency along with theearthquake magnitude of the earthquakes..

**Table 1** Catalog of earthquakes with $M \geq 5.0$ prior to 2008.

| Date | Latitude\°N | Longitude\°E | Depth\km | Magnitude |
| --- | --- | --- | --- | --- |
| 2003.07.21 | 25.95 | 101.23 | 6 | 6.4 |
| 2003.10.16 | 25.92 | 101.30 | 5 | 6.2 |
| 2003.10.25 | 38.35 | 100.93 | 13 | 6.1 |
| 2003.08.18 | 29.57 | 95.60 | 33 | 6 |
| 2003.10.25 | 38.32 | 100.97 | 10 | 6 |
| 2006.07.19 | 33.03 | 96.35 | 30 | 5.9 |
| 2002.12.14 | 39.82 | 97.33 | 22 | 5.8 |
| 2005.08.05 | 26.55 | 103.15 | 21 | 5.6 |
| 2006.03.30 | 35.50 | 95.40 | 18 | 5.6 |
| 2003.11.13 | 34.75 | 103.93 | 10 | 5.5 |
| 2006.07.22 | 28.02 | 104.13 | 9 | 5.5 |
| 2006.08.25 | 28.03 | 104.01 | 7 | 5.5 |
| 2006.06.21 | 33.07 | 104.90 | 15 | 5.4 |
| 2006.07.18 | 33.07 | 96.28 | 20 | 5.4 |
| 2007.07.22 | 38.35 | 101.30 | 19 | 5.3 |
| 2004.09.07 | 34.73 | 103.92 | 19 | 5.2 |
| 2005.09.05 | 27.18 | 103.72 | 10 | 5.2 |
| 2003.08.21 | 27.42 | 101.27 | 5 | 5.1 |
| 2003.10.17 | 25.97 | 101.27 | 7 | 5.1 |
| 2003.11.01 | 25.93 | 101.22 | 3 | 5.1 |
| 2004.11.27 | 25.17 | 98.02 | 12 | 5 |
| 2005.01.05 | 32.28 | 101.55 | 5 | 5 |
| 2005.03.15 | 25.07 | 99.08 | 2 | 5 |
| 2006.07.23 | 33.03 | 96.05 | 30 | 5 |

Further evidence is presented in Table 1, where earthquakes of M 5.0 that occurred during period A (from

2002.092.09 andto 2007.1207.12).) are detailed. There arewere 229 earthquakes withof M 5.0, butwhile the total number induring period A iswas 24, which is accounted for 10.48%, while% overall; the duration of period A is accounted for 33.87% of the total time (timeframe (i.e., period A + period B). Moreover, there iswere no earthquakes withof M 6.5 induring period A, but there arewere 14 earthquakes withof M 6.5 induring period B (from 2008.01 to 2018.03). All theseof this evidence tell usindicates that the seismic activity induring period B is much was significantly more violent and frequent.

[Figure]

[Figure]

Fig. The spatial distribution of earthquakes in the study area. The  orange rectangle represents the study area ( $27°N$  $37°N$ , $97°E$  $107°E$ ). Earthquakes beyond the parameters of the study area are shown because earthquakes close to the study area may also cause TIR  anomalies within the study area.

Fig. 3 shows the spatial distribution of earthquakes.  within the study area. The results indicate that seismic events are clustered primarily in the west and center of the study area ( $25°N$ to $40°N$, $95°E$ to $110°E$ ) which are mountainous regions. The earthquakes are mainly aggregated along faults, with a much sparser spatial distribution in the east and in the Sichuan Basin . There is a clustering phenomenon centered on earthquakes of M 6,  since earthquakes usually occur along the fault lines of active geological movements.

The purpose of investigating the temporal and spatial characteristics of earthquakes is to acquire a general understanding of the seismic activities within the study area. There is another important reason, however, which is to avoid significant accumulation of earthquakes within a short timeframe, and concentrated within a small area, with the result that  the same TIR anomaly corresponds to numerous earthquakes; this phenomenon excessively distorts the statistical results presented above. Around 233 earthquakes were observed to occur after the May 12 2008 Ms 8.0 Wenchuan earthquake, in locations  close to the epicenter of Wenchuan event. In chapter 4.2, the statistical analysis will be divided into two sections: one dealing with the earthquakes that occurred during period C (from 2008.04 to 2008.07), and the other dealing with those that occurred outside of period C.

4.2 Statistical analysis of the correlation between TIR anomalies and earthquakes

In this section,  TIR anomalies are extracted and a statistical analysis of the correlation between TIR anomalies and earthquakes of M 4 is  conducted.

The way to judgeEvaluation of the TIR anomalies isconforms strictly conformedto the rules mentionedguidelines detailed in chapter 3.3.

[Figure]

Fig .4 Two instance forexamples of the correlation between TIR anomalies and earthquakes, and: on the left is the TIR anomaly inrecorded on 2006.12.29 correspondingthat corresponded to two earthquakes, and on the right is the TIR anomaly inrecorded on 2010-.10-.22 correspondingthat did not corresponded to three earthquakes.

[revised manuscript text omitted]

**Table 2 Statistical results of earthquakes with M     5.0**

| | TP1 | FP | TP2 | FN |
|---|---|---|---|---|
| M  5.0 | 15 | 43 | 27 | 223 |
| | PPV:          25.9% | | TPR:           10.8% | |
| | TP1/(TP1+FP) | | TP2/(TP2+FN) | |
| | FDR:          74.1% | | FNR: 89.2%  FN/(TP2+FN) | |
| | FP/(TP1+FP) | | | |

For a more accurate understanding of the eight parameters, an example is presented in Table 2. The example considered the earthquakes of M     5.0, and the results indicate that 58 (TP1+FP) TIR anomalies  appeared over the

duration of the study period, and  15 (TP1) of these correspond to earthquakes while the other  43 (FP) do not; as such, the probability of exact correspondence between  TIR anomalies and earthquakes is 25.9% (PPV), while the probability of no correspondence is 74.1% (FDR). Moreover,  250 (TP2+FN) earthquakes of M 5.0 were recorded in the study area  27 (TP2) of these correspond to TIR anomalies, while the other  223 (FN) do not; as such, the probability of exact correspondence between the earthquakes and TIR anomalies is 10.8% (TPR) while the probability of no correspondence is 89.2% (FNR). We have calculated the earthquakes with M m (m = {3.5, 3.6, 3.7,…,7.8, 7.9, 8.0}), and the experiments are conducted both with and without period C . The results show that these do not differ significantly, so in this section only  the results including period C are discussed.

[Figure]

[Figure]

Fig. 9 The statistical resultresults of earthquakes including the period C (from 2008.04 to 2008.07), and the curvecurves TP1, FP, TP2, FN, PPV and TPR are corresponding, which correspond to the exampleexamples presented in Table 2.

**Table. 2 3 The detailedDetailed results of earthquakes including the period C (from 2008.04 to 2008.07).)**

| M | TP1 | FP | TP2 | FN | PPV | FPRFDR | TPR | FNR |
|---|-----|-----|------|-----------|-----------|-------|-------|-------|
| 3.5 | 3740 | 2121 | 97111 | 35183504 | | 0.362 | 0.027 | 0.973 |
| | | | | | 0.638 0.656 | 0.344 | 0.031 | 0.969 |
| 3.6 | 3538 | 2323 | 87100 | 29462933 | 0.603 0.623 | 0.397 | 0.029 | 0.971 |

| | | | | | 0.377 | 0.033 | 0.967 |
|---|---|---|---|---|---|---|---|
| 3.7 | 34 | 24 | 79 | 2465 | 0.414 | 0.031 | 0.969 |
| | | | | 0.586  | 0.410 | 0.035 | 0.965 |
| 3.8 | 32 | 26 | 70 | 2094 | 0.448 | 0.032 | 0.968 |
| | | | | 0.552  | 0.426 | 0.037 | 0.963 |
| 3.9 | 30 | 28 | 66 | 1817 | 0.483 | 0.035 | 0.965 |
| | | | | 0.517  | 0.426 | 0.040 | 0.960 |
| 4 | 30 | 28 | 63 | 1574 | 0.483 | 0.038 | 0.962 |
| | | | | 0.517  | 0.426 | 0.045 | 0.955 |
| 4.1 | 29 | 29 | 59 | 1291 | 0.500 | 0.044 | 0.956 |
| | | | | 0.500  | 0.426 | 0.050 | 0.950 |
| 4.2 | 27 | 31 | 54 | 1076 | 0.534 | 0.048 | 0.952 |
| | | | | 0.466  | 0.459 | 0.055 | 0.945 |
| 4.3 | 26 | 32 | 51 | 917 | 0.552 | 0.053 | 0.947 |
| | | | | 0.448  | 0.475 | 0.061 | 0.939 |
| 4.4 | 26 | 32 | 49 | 762 | 0.552 | 0.060 | 0.940 |
| | | | | 0.448  | 0.475 | 0.069 | 0.931 |
| 4.5 | 23 | 35 | 48 | 691 | 0.603 | 0.065 | 0.935 |
| | | | | 0.397  | 0.508 | 0.073 | 0.927 |
| 4.6 | 23 | 35 | 45 | 556 | 0.603 | 0.075 | 0.925 |
| | | | | 0.397  | 0.525 | 0.085 | 0.915 |
| 4.7 | 20 | 38 | 40 | 451 | 0.655 | 0.081 | 0.919 |
| | | | | 0.345  | 0.590 | 0.096 | 0.904 |
| 4.8 | 19 | 39 | 37 | 382 | 0.672 | 0.088 | 0.912 |
| | | | | 0.328  | 0.639 | 0.095 | 0.905 |
| 4.9 | 18 | 40 | 33 | 279 | 0.690 | 0.106 | 0.894 |
| | | | | 0.310  | 0.656 | 0.115 | 0.885 |
| 5 | 15 | 43 | 27 | 223 | 0.741 | 0.108 | 0.892 |
| | | | | 0.259  | 0.689 | 0.128 | 0.872 |

Formatted ... [55]
Formatted ... [56]
Formatted ... [57]
Formatted ... [59]
Formatted ... [63]
Formatted ... [64]
Formatted ... [65]
Formatted ... [58]
Formatted ... [60]
Formatted ... [61]
Formatted ... [62]
Formatted ... [67]
Formatted ... [71]
Formatted ... [72]
Formatted ... [73]
Formatted ... [66]
Formatted ... [68]
Formatted ... [70]
Formatted ... [69]
Formatted ... [75]
Formatted ... [79]
Formatted ... [80]
Formatted ... [81]
Formatted ... [74]
Formatted ... [76]
Formatted ... [77]
Formatted ... [78]
Formatted ... [83]
Formatted ... [84]
Formatted ... [85]
Formatted ... [86]
Formatted ... [90]
Formatted ... [91]
Formatted ... [92]
Formatted ... [82]
Formatted ... [87]
Formatted ... [88]
Formatted ... [89]
Formatted ... [97]
Formatted ... [98]
Formatted ... [99]
Formatted ... [93]
Formatted ... [94]
Formatted ... [95]
Formatted ... [96]
Formatted ... [104]
Formatted ... [105]
Formatted ... [106]
Formatted ... [100]
Formatted ... [101]
Formatted ... [102]
Formatted ... [103]
Formatted ... [111]
Formatted ... [112]
Formatted ... [113]
Formatted ... [107]

| | | | | | | | |
|---|---|---|---|---|---|---|---|
| **5.1** | 13 17 | 45 44 | 25 30 | 188 183 | 0.776 | 0.117 | 0.883 |
| | | | 0.224 0.279 | | 0.721 | 0.141 | 0.859 |
| **5.2** | 13 16 | 45 45 | 24 28 | 156 152 | 0.776 | 0.133 | 0.867 |
| | | | 0.224 0.262 | | 0.738 | 0.156 | 0.844 |
| **5.3** | 12 16 | 46 45 | 20 27 | 117 127 | 0.793 | 0.146 | 0.854 |
| | | | 0.207 0.262 | | 0.738 | 0.175 | 0.825 |
| **5.4** | 12 15 | 46 46 | 20 23 | 114 111 | 0.793 | 0.149 | 0.851 |
| | | | 0.207 0.246 | | 0.754 | 0.172 | 0.828 |
| **5.5** | 12 15 | 46 46 | 19 19 | 75 75 | 0.793 | 0.202 | 0.798 |
| | | | 0.207 0.246 | | 0.754 | 0.202 | 0.798 |
| **5.6** | 10 14 | 48 47 | 15 17 | 58 56 | 0.828 | 0.205 | 0.795 |
| | | | 0.172 0.230 | | 0.770 | 0.233 | 0.767 |
| **5.7** | 9 13 | 49 48 | 14 16 | 41 39 | 0.845 | 0.255 | 0.745 |
| | | | 0.155 0.213 | | 0.787 | 0.291 | 0.709 |
| **5.8** | 9 12 | 49 49 | 14 15 | 33 32 | 0.845 | 0.298 | 0.702 |
| | | | 0.155 0.197 | | 0.803 | 0.319 | 0.681 |
| **5.9** | 9 12 | 49 49 | 14 15 | 31 30 | 0.845 | 0.311 | 0.689 |
| | | | 0.155 0.197 | | 0.803 | 0.333 | 0.667 |
| **6** | 9 12 | 49 49 | 12 13 | 26 25 | 0.845 | 0.316 | 0.684 |
| | | | 0.155 0.197 | | 0.803 | 0.342 | 0.658 |
| **6.1** | 7 10 | 51 51 | 10 11 | 19 18 | 0.879 | 0.345 | 0.655 |
| | | | 0.121 0.164 | | 0.836 | 0.379 | 0.621 |
| **6.2** | 7 9 | 51 52 | 10 10 | 17 17 | 0.879 | 0.370 | 0.630 |
| | | | 0.121 0.148 | | 0.852 | 0.370 | 0.630 |
| **6.3** | 7 8 | 51 53 | 8 8 | 12 12 | 0.879 | 0.400 | 0.600 |
| | | | 0.121 0.131 | | 0.869 | 0.400 | 0.600 |
| **6.4** | 7 8 | 51 53 | 8 8 | 10 10 | 0.879 | 0.444 | 0.556 |
| | | | 0.121 0.131 | | 0.869 | 0.444 | 0.556 |
| **6.5** | 6 6 | 52 55 | 6 6 | 7 7 | 0.103 0.098 | 0.897 | 0.462 | 0.538 |

Formatted ... [175]
Formatted ... [176]
Formatted ... [177]
Formatted ... [171]
Formatted ... [172]
Formatted ... [173]
Formatted ... [174]
Formatted ... [182]
Formatted ... [183]
Formatted ... [184]
Formatted ... [178]
Formatted ... [179]
Formatted ... [180]
Formatted ... [181]
Formatted ... [189]
Formatted ... [190]
Formatted ... [191]
Formatted ... [185]
Formatted ... [186]
Formatted ... [187]
Formatted ... [188]
Formatted ... [196]
Formatted ... [197]
Formatted ... [198]
Formatted ... [192]
Formatted ... [193]
Formatted ... [194]
Formatted ... [195]
Formatted ... [200]
Formatted ... [201]
Formatted ... [202]
Formatted ... [203]
Formatted ... [207]
Formatted ... [208]
Formatted ... [209]
Formatted ... [199]
Formatted ... [204]
Formatted ... [205]
Formatted ... [206]
Formatted ... [214]
Formatted ... [215]
Formatted ... [216]
Formatted ... [210]
Formatted ... [211]
Formatted ... [212]
Formatted ... [213]
Formatted ... [221]
Formatted ... [222]
Formatted ... [223]
Formatted ... [217]
Formatted ... [218]
Formatted ... [219]
Formatted ... [220]
Formatted ... [228]
Formatted ... [229]
Formatted ... [230]

| | | | | | | | |
|---|---|---|---|---|---|---|---|
| | | | | | 0.902 | 0.462 | 0.538 |
| 6.6 | 45 5456 | 45 | 65 | | 0.931 | 0.400 | 0.600 |
| | 0.069 0.082 | | | | 0.918 | 0.500 | 0.500 |
| 6.7 | 23 5658 | 23 | 54 | | 0.966 | 0.286 | 0.714 |
| | 0.034 0.049 | | | | 0.951 | 0.429 | 0.571 |
| 6.8 | 12 5759 | 12 | 43 | | 0.983 | 0.200 | 0.800 |
| | 0.017 0.033 | | | | 0.967 | 0.400 | 0.600 |
| 6.9 | 12 5759 | 12 | 43 | | 0.983 | 0.200 | 0.800 |
| | 0.017 0.033 | | | | 0.967 | 0.400 | 0.600 |
| 7 | 12 5759 | 12 | 43 | | 0.983 | 0.200 | 0.800 |
| | 0.017 0.033 | | | | 0.967 | 0.400 | 0.600 |
| 7.1 | 11 5760 | 11 | 33 | | 0.983 | 0.250 | 0.750 |
| | 0.017 0.017 | | | | 0.983 | 0.250 | 0.750 |
| 7.2 | 11 5760 | 11 | 22 | | 0.983 | 0.333 | 0.667 |
| | 0.017 0.017 | | | | 0.983 | 0.333 | 0.667 |
| 7.3 | 11 5760 | 11 | 11 | | 0.983 | 0.500 | 0.500 |
| | 0.017 0.017 | | | | 0983 | 0.500 | 0.500 |
| 7.4 | 00 5861 | 00 | 11 | | 1.000 | 0.000 | 1.000 |
| | 0.000 0.000 | | | | 1.000 | 0.000 | 1.000 |
| 7.5 | 00 5861 | 00 | 11 | | 1.000 | 0.000 | 1.000 |
| | 0.000 0.000 | | | | 1.000 | 0.000 | 1.000 |
| 7.6 | 00 5861 | 00 | 11 | | 1.000 | 0.000 | 1.000 |
| | 0.000 0.000 | | | | 1.000 | 0.000 | 1.000 |
| 7.7 | 00 5861 | 00 | 11 | | 1.000 | 0.000 | 1.000 |
| | 0.000 0.000 | | | | 1.000 | 0.000 | 1.000 |
| 7.8 | 00 5861 | 00 | 11 | | 1.000 | 0.000 | 1.000 |
| | 0.000 0.000 | | | | 1.000 | 0.000 | 1.000 |
| 7.9 | 00 5861 | 00 | 11 | | 1.000 | 0.000 | 1.000 |
| | 0.000 0.000 | | | | 1.000 | 0.000 | 1.000 |

Formatted ... [288]
Formatted ... [289]
Formatted ... [290]
Formatted ... [295]
Formatted ... [296]
Formatted ... [297]
Formatted ... [291]
Formatted ... [292]
Formatted ... [293]
Formatted ... [294]
Formatted ... [302]
Formatted ... [303]
Formatted ... [304]
Formatted ... [298]
Formatted ... [299]
Formatted ... [300]
Formatted ... [301]
Formatted ... [309]
Formatted ... [310]
Formatted ... [311]
Formatted ... [305]
Formatted ... [306]
Formatted ... [307]
Formatted ... [308]
Formatted ... [316]
Formatted ... [317]
Formatted ... [318]
Formatted ... [312]
Formatted ... [313]
Formatted ... [314]
Formatted ... [315]
Formatted ... [320]
Formatted ... [321]
Formatted ... [322]
Formatted ... [323]
Formatted ... [327]
Formatted ... [328]
Formatted ... [329]
Formatted ... [319]
Formatted ... [324]
Formatted ... [325]
Formatted ... [326]
Formatted ... [334]
Formatted ... [335]
Formatted ... [336]
Formatted ... [330]
Formatted ... [331]
Formatted ... [332]
Formatted ... [333]
Formatted ... [341]
Formatted ... [342]
Formatted ... [343]
Formatted ... [337]
Formatted ... [338]
Formatted ... [339]
Formatted ... [340]

[revised manuscript text omitted]

Font: Bold

| Page 17: [2] Formatted Table | Textcheck | 11/11/2018 21:52:00 |
|---|---|---|

Formatted Table

| Page 17: [3] Formatted | Textcheck | 11/11/2018 21:52:00 |
|---|---|---|

Font: Bold

| Page 17: [4] Formatted | Textcheck | 11/11/2018 21:52:00 |
|---|---|---|

Font: Bold

| Page 17: [5] Formatted | Textcheck | 11/11/2018 21:52:00 |
|---|---|---|

Font: Bold

| Page 17: [6] Formatted | Textcheck | 11/11/2018 21:52:00 |
|---|---|---|

Font: Bold

| Page 17: [7] Formatted | Textcheck | 11/11/2018 21:52:00 |
|---|---|---|

Line spacing:    1.5 lines

| Page 17: [8] Formatted | Textcheck | 11/11/2018 21:52:00 |
|---|---|---|

Font: Bold

| Page 17: [9] Formatted | Textcheck | 11/11/2018 21:52:00 |
|---|---|---|

Line spacing:    1.5 lines

| Page 17: [10] Formatted | Textcheck | 11/11/2018 21:52:00 |
|---|---|---|

Font: Bold

| Page 17: [11] Formatted | Textcheck | 11/11/2018 21:52:00 |
|---|---|---|

Line spacing:    1.5 lines

| Page 17: [12] Formatted | Textcheck | 11/11/2018 21:52:00 |
|---|---|---|

Font: Bold

| Page 17: [13] Formatted | Textcheck | 11/11/2018 21:52:00 |
|---|---|---|

Line spacing:    1.5 lines

| Page 17: [14] Formatted | Textcheck | 11/11/2018 21:52:00 |
|---|---|---|

Font: Bold

| Page 17: [15] Formatted | Textcheck | 11/11/2018 21:52:00 |
|---|---|---|

Line spacing:    1.5 lines

| Page 17: [16] Formatted | Textcheck | 11/11/2018 21:52:00 |
|---|---|---|

Font: Bold

| Page 17: [17] Formatted | Textcheck | 11/11/2018 21:52:00 |
|---|---|---|

Line spacing:    1.5 lines

| Page 17: [18] Formatted | Textcheck | 11/11/2018 21:52:00 |
|---|---|---|

Font: Bold

| Page 17: [19] Formatted | Textcheck | 11/11/2018 21:52:00 |
|---|---|---|

Line spacing:    1.5 lines

| Page 17: [20] Formatted | Textcheck | 11/11/2018 21:52:00 |
|---|---|---|

Font: Bold

| Page 17: [21] Formatted | Textcheck | 11/11/2018 21:52:00 |
|---|---|---|

Line spacing:    1.5 lines

| Page 17: [22] Formatted | Textcheck | 11/11/2018 21:52:00 |
|---|---|---|

Font: Bold

| Page 17: [23] Formatted | Textcheck | 11/11/2018 21:52:00 |
|---|---|---|

Line spacing:    1.5 lines

| Page 17: [24] Formatted | Textcheck | 11/11/2018 21:52:00 |
|---|---|---|

Font: Bold

| Page 17: [25] Formatted | Textcheck | 11/11/2018 21:52:00 |
|---|---|---|

Line spacing:    1.5 lines

| Page 17: [26] Formatted | Textcheck | 11/11/2018 21:52:00 |
|---|---|---|

Font: Bold

| Page 17: [27] Formatted | Textcheck | 11/11/2018 21:52:00 |
|---|---|---|

Line spacing:    1.5 lines

| Page 17: [28] Formatted | Textcheck | 11/11/2018 21:52:00 |
|---|---|---|

Font: Bold

| Page 17: [29] Formatted | Textcheck | 11/11/2018 21:52:00 |
|---|---|---|

Line spacing:    1.5 lines

| Page 17: [30] Formatted | Textcheck | 11/11/2018 21:52:00 |
|---|---|---|

Font: Bold

| Page 17: [31] Formatted | Textcheck | 11/11/2018 21:52:00 |
|---|---|---|

Line spacing:    1.5 lines

| Page 17: [32] Formatted | Textcheck | 11/11/2018 21:52:00 |
|---|---|---|

Font: Bold

| Page 17: [33] Formatted | Textcheck | 11/11/2018 21:52:00 |
|---|---|---|

Line spacing:    1.5 lines

| Page 17: [34] Formatted | Textcheck | 11/11/2018 21:52:00 |
|---|---|---|

Font: Bold

| Page 17: [35] Formatted | Textcheck | 11/11/2018 21:52:00 |
|---|---|---|

Line spacing:    1.5 lines

| Page 17: [36] Formatted | Textcheck | 11/11/2018 21:52:00 |
|---|---|---|

Font: Bold

| Page 17: [37] Formatted | Textcheck | 11/11/2018 21:52:00 |
|---|---|---|

Line spacing:    1.5 lines

| Page 17: [38] Formatted | Textcheck | 11/11/2018 21:52:00 |
|---|---|---|

Font: Bold

| Page 17: [39] Formatted | Textcheck | 11/11/2018 21:52:00 |
|---|---|---|

Line spacing:    1.5 lines

| Page 17: [40] Formatted | Textcheck | 11/11/2018 21:52:00 |
|---|---|---|

Font: Bold

| Page 17: [41] Formatted | Textcheck | 11/11/2018 21:52:00 |
|---|---|---|

Line spacing:    1.5 lines

| Page 17: [42] Formatted | Textcheck | 11/11/2018 21:52:00 |
|---|---|---|

Font: Bold

| Page 17: [43] Formatted | Textcheck | 11/11/2018 21:52:00 |
|---|---|---|

Line spacing: 1.5 lines

| Page 17: [44] Formatted | Textcheck | 11/11/2018 21:52:00 |
|---|---|---|

Font: Bold

| Page 17: [45] Formatted | Textcheck | 11/11/2018 21:52:00 |
|---|---|---|

Line spacing: 1.5 lines

| Page 17: [46] Formatted | Textcheck | 11/11/2018 21:52:00 |
|---|---|---|

Font: Bold

| Page 17: [47] Formatted | Textcheck | 11/11/2018 21:52:00 |
|---|---|---|

Line spacing: 1.5 lines

| Page 17: [48] Formatted | Textcheck | 11/11/2018 21:52:00 |
|---|---|---|

Font: Bold

| Page 17: [49] Formatted | Textcheck | 11/11/2018 21:52:00 |
|---|---|---|

Line spacing: 1.5 lines

| Page 17: [50] Formatted | Textcheck | 11/11/2018 21:52:00 |
|---|---|---|

Font: Bold

| Page 17: [51] Formatted | Textcheck | 11/11/2018 21:52:00 |
|---|---|---|

Line spacing: 1.5 lines

| Page 17: [52] Formatted | Textcheck | 11/11/2018 21:52:00 |
|---|---|---|

Font: Bold

| Page 17: [53] Formatted | Textcheck | 11/11/2018 21:52:00 |
|---|---|---|

Line spacing: 1.5 lines

| Page 17: [54] Formatted | Textcheck | 11/11/2018 21:52:00 |
|---|---|---|

Line spacing: 1.5 lines

| Page 33: [55] Formatted | 张 颖 | 21/11/2018 15:24:00 |
|---|---|---|

Font: 10.5 pt

| Page 33: [56] Formatted | 张 颖 | 21/11/2018 15:24:00 |
|---|---|---|

Font: 10.5 pt

| Page 33: [57] Formatted | 张 颖 | 21/11/2018 15:24:00 |
|---|---|---|

Font: 10.5 pt

| Page 33: [58] Formatted | Textcheck | 11/11/2018 21:52:00 |
|---|---|---|

Line spacing: 1.5 lines

| Page 33: [59] Formatted | 张 颖 | 21/11/2018 15:24:00 |
|---|---|---|

Font color: Black

| Page 33: [60] Formatted | 张 颖 | 21/11/2018 15:24:00 |
|---|---|---|

Font: (Default) Calibri, 10.5 pt

| Page 33: [61] Formatted | 张 颖 | 21/11/2018 15:17:00 |
|---|---|---|

Line spacing: 1.5 lines

| Page 33: [62] Formatted | 张 颖 | 21/11/2018 15:24:00 |
|---|---|---|

Font: 10.5 pt

| Page 33: [63] Formatted | 张 颖 | 21/11/2018 15:24:00 |
|---|---|---|

Font: (Default) Calibri, 10.5 pt

| Page 33: [63] Formatted | 张 颖 | 21/11/2018 15:24:00 |
|---|---|---|

Font: (Default) Calibri, 10.5 pt

| Page 33: [64] Formatted | 张 颖 | 21/11/2018 15:24:00 |
|---|---|---|

Font: (Default) Calibri, 10.5 pt

| Page 33: [64] Formatted | 张 颖 | 21/11/2018 15:24:00 |
|---|---|---|

Font: (Default) Calibri, 10.5 pt

| Page 33: [65] Formatted | 张 颖 | 21/11/2018 15:24:00 |
|---|---|---|

Font: (Default) Calibri, 10.5 pt

| Page 33: [65] Formatted | 张 颖 | 21/11/2018 15:24:00 |
|---|---|---|

Font: (Default) Calibri, 10.5 pt

| Page 33: [66] Formatted | Textcheck | 11/11/2018 21:52:00 |
|---|---|---|

Line spacing:    1.5 lines

| Page 33: [67] Formatted | 张 颖 | 21/11/2018 15:24:00 |
|---|---|---|

Font color: Black

| Page 33: [68] Formatted | 张 颖 | 21/11/2018 15:24:00 |
|---|---|---|

Font: (Default) Calibri, 10.5 pt

| Page 33: [69] Formatted | 张 颖 | 21/11/2018 15:17:00 |
|---|---|---|

Line spacing:    1.5 lines

| Page 33: [70] Formatted | 张 颖 | 21/11/2018 15:24:00 |
|---|---|---|

Font: 10.5 pt

| Page 33: [71] Formatted | 张 颖 | 21/11/2018 15:24:00 |
|---|---|---|

Font: (Default) Calibri, 10.5 pt

| Page 33: [71] Formatted | 张 颖 | 21/11/2018 15:24:00 |
|---|---|---|

Font: (Default) Calibri, 10.5 pt

| Page 33: [72] Formatted | 张 颖 | 21/11/2018 15:24:00 |
|---|---|---|

Font: (Default) Calibri, 10.5 pt

| Page 33: [72] Formatted | 张 颖 | 21/11/2018 15:24:00 |
|---|---|---|

Font: (Default) Calibri, 10.5 pt

| Page 33: [73] Formatted | 张 颖 | 21/11/2018 15:24:00 |
|---|---|---|

Font: (Default) Calibri, 10.5 pt

| Page 33: [73] Formatted | 张 颖 | 21/11/2018 15:24:00 |
|---|---|---|

Font: (Default) Calibri, 10.5 pt

| Page 33: [74] Formatted | Textcheck | 11/11/2018 21:52:00 |
|---|---|---|

Line spacing:    1.5 lines

| Page 33: [75] Formatted | 张 颖 | 21/11/2018 15:24:00 |
|---|---|---|

Font color: Black

| Page 33: [76] Formatted | 张 颖 | 21/11/2018 15:24:00 |
|---|---|---|

Font: (Default) Calibri, 10.5 pt

| Page 33: [77] Formatted | 张 颖 | 21/11/2018 15:17:00 |
|---|---|---|

Line spacing:    1.5 lines

| Page 33: [78] Formatted | 张 颖 | 21/11/2018 15:24:00 |
|---|---|---|

Font: 10.5 pt

| Page 33: [79] Formatted | 张 颖 | 21/11/2018 15:24:00 |
|---|---|---|

Font: (Default) Calibri, 10.5 pt

| Page 33: [79] Formatted | 张 颖 | 21/11/2018 15:24:00 |
|---|---|---|

Font: (Default) Calibri, 10.5 pt

| Page 33: [80] Formatted | 张 颖 | 21/11/2018 15:24:00 |
|---|---|---|

Font: (Default) Calibri, 10.5 pt

| Page 33: [80] Formatted | 张 颖 | 21/11/2018 15:24:00 |
|---|---|---|

Font: (Default) Calibri, 10.5 pt

| Page 33: [81] Formatted | 张 颖 | 21/11/2018 15:24:00 |
|---|---|---|

Font: (Default) Calibri, 10.5 pt

| Page 33: [81] Formatted | 张 颖 | 21/11/2018 15:24:00 |
|---|---|---|

Font: (Default) Calibri, 10.5 pt

| Page 33: [82] Formatted | Textcheck | 11/11/2018 21:52:00 |
|---|---|---|

Line spacing:    1.5 lines

| Page 33: [83] Formatted | 张 颖 | 20/11/2018 14:27:00 |
|---|---|---|

Font color: Red

| Page 33: [84] Formatted | 张 颖 | 20/11/2018 14:27:00 |
|---|---|---|

Font color: Red

| Page 33: [85] Formatted | 张 颖 | 20/11/2018 14:27:00 |
|---|---|---|

Font color: Red

| Page 33: [86] Formatted | 张 颖 | 20/11/2018 14:27:00 |
|---|---|---|

Font color: Red

| Page 33: [87] Formatted | 张 颖 | 21/11/2018 15:17:00 |
|---|---|---|

Font: (Default) Calibri, 10.5 pt, Font color: Red

| Page 33: [88] Formatted | 张 颖 | 21/11/2018 15:17:00 |
|---|---|---|

Line spacing:    1.5 lines

| Page 33: [89] Formatted | 张 颖 | 21/11/2018 15:17:00 |
|---|---|---|

Font: 10.5 pt

| Page 33: [89] Formatted | 张 颖 | 21/11/2018 15:17:00 |
|---|---|---|

Font: 10.5 pt

| Page 33: [90] Formatted | 张 颖 | 21/11/2018 15:17:00 |
|---|---|---|

Font: (Default) Calibri, 10.5 pt, Font color: Red

| Page 33: [90] Formatted | 张 颖 | 21/11/2018 15:17:00 |
|---|---|---|

Font: (Default) Calibri, 10.5 pt, Font color: Red

| Page 33: [90] Formatted | 张 颖 | 21/11/2018 15:17:00 |
|---|---|---|

Font: (Default) Calibri, 10.5 pt, Font color: Red

| Page 33: [91] Formatted | 张 颖 | 21/11/2018 15:17:00 |
|---|---|---|

Font: (Default) Calibri, 10.5 pt, Font color: Red

| Page 33: [91] Formatted | 张 颖 | 21/11/2018 15:17:00 |
|---|---|---|

Font: (Default) Calibri, 10.5 pt, Font color: Red

| Page 33: [91] Formatted | 张 颖 | 21/11/2018 15:17:00 |
|---|---|---|

Font: (Default) Calibri, 10.5 pt, Font color: Red

| Page 33: [92] Formatted | 张 颖 | 21/11/2018 15:17:00 |
|---|---|---|

Font: (Default) Calibri, 10.5 pt, Font color: Red

| Page 33: [92] Formatted | 张 颖 | 21/11/2018 15:17:00 |
|---|---|---|

Font: (Default) Calibri, 10.5 pt, Font color: Red

| Page 33: [92] Formatted | 张 颖 | 21/11/2018 15:17:00 |
|---|---|---|

Font: (Default) Calibri, 10.5 pt, Font color: Red

| Page 33: [93] Formatted | Textcheck | 11/11/2018 21:52:00 |
|---|---|---|

Line spacing:    1.5 lines

| Page 33: [94] Formatted | 张 颖 | 21/11/2018 15:24:00 |
|---|---|---|

Font: (Default) Calibri, 10.5 pt

| Page 33: [95] Formatted | 张 颖 | 21/11/2018 15:17:00 |
|---|---|---|

Line spacing:    1.5 lines

| Page 33: [96] Formatted | 张 颖 | 21/11/2018 15:24:00 |
|---|---|---|

Font: 10.5 pt

| Page 33: [97] Formatted | 张 颖 | 21/11/2018 15:24:00 |
|---|---|---|

Font: (Default) Calibri, 10.5 pt

| Page 33: [97] Formatted | 张 颖 | 21/11/2018 15:24:00 |
|---|---|---|

Font: (Default) Calibri, 10.5 pt

| Page 33: [98] Formatted | 张 颖 | 21/11/2018 15:24:00 |
|---|---|---|

Font: (Default) Calibri, 10.5 pt

| Page 33: [98] Formatted | 张 颖 | 21/11/2018 15:24:00 |
|---|---|---|

Font: (Default) Calibri, 10.5 pt

| Page 33: [99] Formatted | 张 颖 | 21/11/2018 15:24:00 |
|---|---|---|

Font: (Default) Calibri, 10.5 pt

| Page 33: [99] Formatted | 张 颖 | 21/11/2018 15:24:00 |
|---|---|---|

Font: (Default) Calibri, 10.5 pt

| Page 33: [100] Formatted | Textcheck | 11/11/2018 21:52:00 |
|---|---|---|

Line spacing:    1.5 lines

| Page 33: [101] Formatted | 张 颖 | 21/11/2018 15:24:00 |
|---|---|---|

Font: (Default) Calibri, 10.5 pt

| Page 33: [102] Formatted | 张 颖 | 21/11/2018 15:17:00 |
|---|---|---|

Line spacing:    1.5 lines

| Page 33: [103] Formatted | 张 颖 | 21/11/2018 15:24:00 |
|---|---|---|

Font: 10.5 pt

| Page 33: [104] Formatted | 张 颖 | 21/11/2018 15:24:00 |

Font: (Default) Calibri, 10.5 pt

| Page 33: [104] Formatted | 张 颖 | 21/11/2018 15:24:00 |

Font: (Default) Calibri, 10.5 pt

| Page 33: [105] Formatted | 张 颖 | 21/11/2018 15:24:00 |

Font: (Default) Calibri, 10.5 pt

| Page 33: [105] Formatted | 张 颖 | 21/11/2018 15:24:00 |

Font: (Default) Calibri, 10.5 pt

| Page 33: [106] Formatted | 张 颖 | 21/11/2018 15:24:00 |

Font: (Default) Calibri, 10.5 pt

| Page 33: [106] Formatted | 张 颖 | 21/11/2018 15:24:00 |

Font: (Default) Calibri, 10.5 pt

| Page 33: [107] Formatted | Textcheck | 11/11/2018 21:52:00 |

Line spacing:    1.5 lines

| Page 33: [108] Formatted | 张 颖 | 21/11/2018 15:24:00 |

Font: (Default) Calibri, 10.5 pt

| Page 33: [109] Formatted | 张 颖 | 21/11/2018 15:17:00 |

Line spacing:    1.5 lines

| Page 33: [110] Formatted | 张 颖 | 21/11/2018 15:24:00 |

Font: 10.5 pt

| Page 33: [111] Formatted | 张 颖 | 21/11/2018 15:24:00 |

Font: (Default) Calibri, 10.5 pt

| Page 33: [111] Formatted | 张 颖 | 21/11/2018 15:24:00 |

Font: (Default) Calibri, 10.5 pt

| Page 33: [112] Formatted | 张 颖 | 21/11/2018 15:24:00 |

Font: (Default) Calibri, 10.5 pt

| Page 33: [112] Formatted | 张 颖 | 21/11/2018 15:24:00 |

Font: (Default) Calibri, 10.5 pt

| Page 33: [113] Formatted | 张 颖 | 21/11/2018 15:24:00 |

Font: (Default) Calibri, 10.5 pt

| Page 33: [113] Formatted | 张 颖 | 21/11/2018 15:24:00 |

Font: (Default) Calibri, 10.5 pt

| Page 33: [114] Formatted | Textcheck | 11/11/2018 21:52:00 |

Line spacing:    1.5 lines

| Page 33: [115] Formatted | 张 颖 | 21/11/2018 15:24:00 |

Font: (Default) Calibri, 10.5 pt

| Page 33: [116] Formatted | 张 颖 | 21/11/2018 15:17:00 |

Line spacing:    1.5 lines

| Page 33: [117] Formatted | 张 颖 | 21/11/2018 15:24:00 |

Font: 10.5 pt

| Page 33: [118] Formatted | 张 颖 | 21/11/2018 15:24:00 |
|---|---|---|

Font: (Default) Calibri, 10.5 pt

| Page 33: [118] Formatted | 张 颖 | 21/11/2018 15:24:00 |
|---|---|---|

Font: (Default) Calibri, 10.5 pt

| Page 33: [119] Formatted | 张 颖 | 21/11/2018 15:24:00 |
|---|---|---|

Font: (Default) Calibri, 10.5 pt

| Page 33: [119] Formatted | 张 颖 | 21/11/2018 15:24:00 |
|---|---|---|

Font: (Default) Calibri, 10.5 pt

| Page 33: [120] Formatted | 张 颖 | 21/11/2018 15:24:00 |
|---|---|---|

Font: (Default) Calibri, 10.5 pt

| Page 33: [120] Formatted | 张 颖 | 21/11/2018 15:24:00 |
|---|---|---|

Font: (Default) Calibri, 10.5 pt

| Page 33: [121] Formatted | Textcheck | 11/11/2018 21:52:00 |
|---|---|---|

Line spacing:    1.5 lines

| Page 33: [122] Formatted | 张 颖 | 20/11/2018 14:27:00 |
|---|---|---|

Font color: Red

| Page 33: [123] Formatted | 张 颖 | 20/11/2018 14:27:00 |
|---|---|---|

Font color: Red

| Page 33: [124] Formatted | 张 颖 | 20/11/2018 14:27:00 |
|---|---|---|

Font color: Red

| Page 33: [125] Formatted | 张 颖 | 20/11/2018 14:27:00 |
|---|---|---|

Font color: Red

| Page 33: [126] Formatted | 张 颖 | 21/11/2018 15:17:00 |
|---|---|---|

Font: (Default) Calibri, 10.5 pt, Font color: Red

| Page 33: [127] Formatted | 张 颖 | 21/11/2018 15:17:00 |
|---|---|---|

Line spacing:    1.5 lines

| Page 33: [128] Formatted | 张 颖 | 21/11/2018 15:17:00 |
|---|---|---|

Font: 10.5 pt

| Page 33: [129] Formatted | 张 颖 | 21/11/2018 15:17:00 |
|---|---|---|

Font: (Default) Calibri, 10.5 pt, Font color: Red

| Page 33: [129] Formatted | 张 颖 | 21/11/2018 15:17:00 |
|---|---|---|

Font: (Default) Calibri, 10.5 pt, Font color: Red

| Page 33: [130] Formatted | 张 颖 | 21/11/2018 15:17:00 |
|---|---|---|

Font: (Default) Calibri, 10.5 pt, Font color: Red

| Page 33: [130] Formatted | 张 颖 | 21/11/2018 15:17:00 |
|---|---|---|

Font: (Default) Calibri, 10.5 pt, Font color: Red

| Page 33: [131] Formatted | 张 颖 | 21/11/2018 15:17:00 |
|---|---|---|

Font: (Default) Calibri, 10.5 pt, Font color: Red

| Page 33: [131] Formatted | 张 颖 | 21/11/2018 15:17:00 |
|---|---|---|

Font: (Default) Calibri, 10.5 pt, Font color: Red

| Page 33: [132] Formatted | Textcheck | 11/11/2018 21:52:00 |
|---|---|---|

Line spacing: 1.5 lines

| Page 33: [133] Formatted | 张 颖 | 21/11/2018 15:24:00 |
|---|---|---|

Font: (Default) Calibri, 10.5 pt

| Page 33: [134] Formatted | 张 颖 | 21/11/2018 15:17:00 |
|---|---|---|

Line spacing: 1.5 lines

| Page 33: [135] Formatted | 张 颖 | 21/11/2018 15:24:00 |
|---|---|---|

Font: 10.5 pt

| Page 33: [136] Formatted | 张 颖 | 21/11/2018 15:24:00 |
|---|---|---|

Font: (Default) Calibri, 10.5 pt

| Page 33: [136] Formatted | 张 颖 | 21/11/2018 15:24:00 |
|---|---|---|

Font: (Default) Calibri, 10.5 pt

| Page 33: [137] Formatted | 张 颖 | 21/11/2018 15:24:00 |
|---|---|---|

Font: (Default) Calibri, 10.5 pt

| Page 33: [137] Formatted | 张 颖 | 21/11/2018 15:24:00 |
|---|---|---|

Font: (Default) Calibri, 10.5 pt

| Page 33: [138] Formatted | 张 颖 | 21/11/2018 15:24:00 |
|---|---|---|

Font: (Default) Calibri, 10.5 pt

| Page 33: [138] Formatted | 张 颖 | 21/11/2018 15:24:00 |
|---|---|---|

Font: (Default) Calibri, 10.5 pt

| Page 33: [139] Formatted | Textcheck | 11/11/2018 21:52:00 |
|---|---|---|

Line spacing: 1.5 lines

| Page 33: [140] Formatted | 张 颖 | 21/11/2018 15:24:00 |
|---|---|---|

Font: (Default) Calibri, 10.5 pt

| Page 33: [141] Formatted | 张 颖 | 21/11/2018 15:17:00 |
|---|---|---|

Line spacing: 1.5 lines

| Page 33: [142] Formatted | 张 颖 | 21/11/2018 15:24:00 |
|---|---|---|

Font: 10.5 pt

| Page 33: [143] Formatted | 张 颖 | 21/11/2018 15:24:00 |
|---|---|---|

Font: (Default) Calibri, 10.5 pt

| Page 33: [143] Formatted | 张 颖 | 21/11/2018 15:24:00 |
|---|---|---|

Font: (Default) Calibri, 10.5 pt

| Page 33: [144] Formatted | 张 颖 | 21/11/2018 15:24:00 |
|---|---|---|

Font: (Default) Calibri, 10.5 pt

| Page 33: [144] Formatted | 张 颖 | 21/11/2018 15:24:00 |
|---|---|---|

Font: (Default) Calibri, 10.5 pt

| Page 33: [145] Formatted | 张 颖 | 21/11/2018 15:24:00 |
|---|---|---|

Font: (Default) Calibri, 10.5 pt

| Page 33: [145] Formatted | 张 颖 | 21/11/2018 15:24:00 |
|---|---|---|

Font: (Default) Calibri, 10.5 pt

| Page 33: [146] Formatted | Textcheck | 11/11/2018 21:52:00 |
|---|---|---|

Line spacing:    1.5 lines

| Page 33: [147] Formatted | 张 颖 | 21/11/2018 15:24:00 |
|---|---|---|

| Page 33: [148] Formatted | 张 颖 | 21/11/2018 15:17:00 |
|---|---|---|

Line spacing:    1.5 lines

| Page 33: [149] Formatted | 张 颖 | 21/11/2018 15:24:00 |
|---|---|---|

Font: 10.5 pt

| Page 33: [150] Formatted | 张 颖 | 21/11/2018 15:24:00 |
|---|---|---|

Font: (Default) Calibri, 10.5 pt

| Page 33: [150] Formatted | 张 颖 | 21/11/2018 15:24:00 |
|---|---|---|

Font: (Default) Calibri, 10.5 pt

| Page 33: [151] Formatted | 张 颖 | 21/11/2018 15:24:00 |
|---|---|---|

Font: (Default) Calibri, 10.5 pt

| Page 33: [151] Formatted | 张 颖 | 21/11/2018 15:24:00 |
|---|---|---|

Font: (Default) Calibri, 10.5 pt

| Page 33: [152] Formatted | 张 颖 | 21/11/2018 15:24:00 |
|---|---|---|

Font: (Default) Calibri, 10.5 pt

| Page 33: [152] Formatted | 张 颖 | 21/11/2018 15:24:00 |
|---|---|---|

Font: (Default) Calibri, 10.5 pt

| Page 33: [153] Formatted | Textcheck | 11/11/2018 21:52:00 |
|---|---|---|

Line spacing:    1.5 lines

| Page 33: [154] Formatted | 张 颖 | 21/11/2018 15:24:00 |
|---|---|---|

Font: (Default) Calibri, 10.5 pt

| Page 33: [155] Formatted | 张 颖 | 21/11/2018 15:17:00 |
|---|---|---|

Line spacing:    1.5 lines

| Page 33: [156] Formatted | 张 颖 | 21/11/2018 15:24:00 |
|---|---|---|

Font: 10.5 pt

| Page 33: [157] Formatted | 张 颖 | 21/11/2018 15:24:00 |
|---|---|---|

Font: (Default) Calibri, 10.5 pt

| Page 33: [157] Formatted | 张 颖 | 21/11/2018 15:24:00 |
|---|---|---|

Font: (Default) Calibri, 10.5 pt

| Page 33: [158] Formatted | 张 颖 | 21/11/2018 15:24:00 |
|---|---|---|

Font: (Default) Calibri, 10.5 pt

| Page 33: [158] Formatted | 张 颖 | 21/11/2018 15:24:00 |
|---|---|---|

Font: (Default) Calibri, 10.5 pt

| Page 33: [159] Formatted | 张 颖 | 21/11/2018 15:24:00 |
|---|---|---|

Font: (Default) Calibri, 10.5 pt

| Page 33: [159] Formatted | 张 颖 | 21/11/2018 15:24:00 |
|---|---|---|

Font: (Default) Calibri, 10.5 pt

| Page 33: [160] Formatted | Textcheck | 11/11/2018 21:52:00 |
|---|---|---|

Line spacing:    1.5 lines

| Page 33: [161] Formatted | 张 颖 | 20/11/2018 14:27:00 |
|---|---|---|

Font color: Red

| Page 33: [162] Formatted | 张 颖 | 20/11/2018 14:27:00 |
|---|---|---|

Font color: Red

| Page 33: [163] Formatted | 张 颖 | 20/11/2018 14:27:00 |
|---|---|---|

Font color: Red

| Page 33: [164] Formatted | 张 颖 | 20/11/2018 14:27:00 |
|---|---|---|

Font color: Red

| Page 33: [165] Formatted | 张 颖 | 21/11/2018 15:17:00 |
|---|---|---|

Font: (Default) Calibri, 10.5 pt, Font color: Red

| Page 33: [166] Formatted | 张 颖 | 21/11/2018 15:17:00 |
|---|---|---|

Line spacing:    1.5 lines

| Page 33: [167] Formatted | 张 颖 | 21/11/2018 15:17:00 |
|---|---|---|

Font: 10.5 pt

| Page 33: [167] Formatted | 张 颖 | 21/11/2018 15:17:00 |
|---|---|---|

Font: 10.5 pt

| Page 33: [168] Formatted | 张 颖 | 21/11/2018 15:17:00 |
|---|---|---|

Font: (Default) Calibri, 10.5 pt, Font color: Red

| Page 33: [168] Formatted | 张 颖 | 21/11/2018 15:17:00 |
|---|---|---|

Font: (Default) Calibri, 10.5 pt, Font color: Red

| Page 33: [168] Formatted | 张 颖 | 21/11/2018 15:17:00 |
|---|---|---|

Font: (Default) Calibri, 10.5 pt, Font color: Red

| Page 33: [169] Formatted | 张 颖 | 21/11/2018 15:17:00 |
|---|---|---|

Font: (Default) Calibri, 10.5 pt, Font color: Red

| Page 33: [169] Formatted | 张 颖 | 21/11/2018 15:17:00 |
|---|---|---|

Font: (Default) Calibri, 10.5 pt, Font color: Red

| Page 33: [169] Formatted | 张 颖 | 21/11/2018 15:17:00 |
|---|---|---|

Font: (Default) Calibri, 10.5 pt, Font color: Red

| Page 33: [170] Formatted | 张 颖 | 21/11/2018 15:17:00 |
|---|---|---|

Font: (Default) Calibri, 10.5 pt, Font color: Red

| Page 33: [170] Formatted | 张 颖 | 21/11/2018 15:17:00 |
|---|---|---|

Font: (Default) Calibri, 10.5 pt, Font color: Red

| Page 33: [170] Formatted | 张 颖 | 21/11/2018 15:17:00 |
|---|---|---|

Font: (Default) Calibri, 10.5 pt, Font color: Red

| Page 34: [171] Formatted | Textcheck | 11/11/2018 21:52:00 |
|---|---|---|

Line spacing:    1.5 lines

| Page 34: [172] Formatted | 张 颖 | 21/11/2018 15:25:00 |
|---|---|---|

Font: (Default) Calibri, 10.5 pt

| Page 34: [173] Formatted | 张 颖 | 21/11/2018 15:17:00 |

Line spacing:    1.5 lines

| Page 34: [174] Formatted | 张 颖 | 21/11/2018 15:25:00 |

Font: 10.5 pt

| Page 34: [175] Formatted | 张 颖 | 21/11/2018 15:25:00 |

Font: (Default) Calibri, 10.5 pt

| Page 34: [175] Formatted | 张 颖 | 21/11/2018 15:25:00 |

Font: (Default) Calibri, 10.5 pt

| Page 34: [176] Formatted | 张 颖 | 21/11/2018 15:25:00 |

Font: (Default) Calibri, 10.5 pt

| Page 34: [176] Formatted | 张 颖 | 21/11/2018 15:25:00 |

Font: (Default) Calibri, 10.5 pt

| Page 34: [177] Formatted | 张 颖 | 21/11/2018 15:25:00 |

Font: (Default) Calibri, 10.5 pt

| Page 34: [177] Formatted | 张 颖 | 21/11/2018 15:25:00 |

Font: (Default) Calibri, 10.5 pt

| Page 34: [178] Formatted | Textcheck | 11/11/2018 21:52:00 |

Line spacing:    1.5 lines

| Page 34: [179] Formatted | 张 颖 | 21/11/2018 15:25:00 |

Font: (Default) Calibri, 10.5 pt

| Page 34: [180] Formatted | 张 颖 | 21/11/2018 15:17:00 |

Line spacing:    1.5 lines

| Page 34: [181] Formatted | 张 颖 | 21/11/2018 15:25:00 |

Font: 10.5 pt

| Page 34: [182] Formatted | 张 颖 | 21/11/2018 15:25:00 |

Font: (Default) Calibri, 10.5 pt

| Page 34: [182] Formatted | 张 颖 | 21/11/2018 15:25:00 |

Font: (Default) Calibri, 10.5 pt

| Page 34: [183] Formatted | 张 颖 | 21/11/2018 15:25:00 |

Font: (Default) Calibri, 10.5 pt

| Page 34: [183] Formatted | 张 颖 | 21/11/2018 15:25:00 |

Font: (Default) Calibri, 10.5 pt

| Page 34: [184] Formatted | 张 颖 | 21/11/2018 15:25:00 |

Font: (Default) Calibri, 10.5 pt

| Page 34: [184] Formatted | 张 颖 | 21/11/2018 15:25:00 |

Font: (Default) Calibri, 10.5 pt

| Page 34: [185] Formatted | Textcheck | 11/11/2018 21:52:00 |

Line spacing:    1.5 lines

| Page 34: [186] Formatted | 张 颖 | 21/11/2018 15:25:00 |

Font: (Default) Calibri, 10.5 pt

| Page 34: [187] Formatted | 张 颖 | 21/11/2018 15:17:00 |
|---|---|---|

Line spacing:    1.5 lines

| Page 34: [188] Formatted | 张 颖 | 21/11/2018 15:25:00 |
|---|---|---|

Font: 10.5 pt

| Page 34: [189] Formatted | 张 颖 | 21/11/2018 15:25:00 |
|---|---|---|

Font: (Default) Calibri, 10.5 pt

| Page 34: [189] Formatted | 张 颖 | 21/11/2018 15:25:00 |
|---|---|---|

Font: (Default) Calibri, 10.5 pt

| Page 34: [190] Formatted | 张 颖 | 21/11/2018 15:25:00 |
|---|---|---|

Font: (Default) Calibri, 10.5 pt

| Page 34: [190] Formatted | 张 颖 | 21/11/2018 15:25:00 |
|---|---|---|

Font: (Default) Calibri, 10.5 pt

| Page 34: [191] Formatted | 张 颖 | 21/11/2018 15:25:00 |
|---|---|---|

Font: (Default) Calibri, 10.5 pt

| Page 34: [191] Formatted | 张 颖 | 21/11/2018 15:25:00 |
|---|---|---|

Font: (Default) Calibri, 10.5 pt

| Page 34: [192] Formatted | Textcheck | 11/11/2018 21:52:00 |
|---|---|---|

Line spacing:    1.5 lines

| Page 34: [193] Formatted | 张 颖 | 21/11/2018 15:25:00 |
|---|---|---|

Font: (Default) Calibri, 10.5 pt

| Page 34: [194] Formatted | 张 颖 | 21/11/2018 15:17:00 |
|---|---|---|

Line spacing:    1.5 lines

| Page 34: [195] Formatted | 张 颖 | 21/11/2018 15:25:00 |
|---|---|---|

Font: 10.5 pt

| Page 34: [196] Formatted | 张 颖 | 21/11/2018 15:25:00 |
|---|---|---|

Font: (Default) Calibri, 10.5 pt

| Page 34: [196] Formatted | 张 颖 | 21/11/2018 15:25:00 |
|---|---|---|

Font: (Default) Calibri, 10.5 pt

| Page 34: [197] Formatted | 张 颖 | 21/11/2018 15:25:00 |
|---|---|---|

Font: (Default) Calibri, 10.5 pt

| Page 34: [197] Formatted | 张 颖 | 21/11/2018 15:25:00 |
|---|---|---|

Font: (Default) Calibri, 10.5 pt

| Page 34: [198] Formatted | 张 颖 | 21/11/2018 15:25:00 |
|---|---|---|

Font: (Default) Calibri, 10.5 pt

| Page 34: [198] Formatted | 张 颖 | 21/11/2018 15:25:00 |
|---|---|---|

Font: (Default) Calibri, 10.5 pt

| Page 34: [199] Formatted | Textcheck | 11/11/2018 21:52:00 |
|---|---|---|

Line spacing:    1.5 lines

| Page 34: [200] Formatted | 张 颖 | 20/11/2018 14:27:00 |
|---|---|---|

Font color: Red

| Page 34: [201] Formatted | 张 颖 | 20/11/2018 14:27:00 |
|---|---|---|

Font color: Red

| Page 34: [202] Formatted | 张 颖 | 20/11/2018 14:27:00 |
|---|---|---|

Font color: Red

| Page 34: [203] Formatted | 张 颖 | 20/11/2018 14:27:00 |
|---|---|---|

Font color: Red

| Page 34: [204] Formatted | 张 颖 | 21/11/2018 15:17:00 |
|---|---|---|

Font: (Default) Calibri, 10.5 pt, Font color: Red

| Page 34: [205] Formatted | 张 颖 | 21/11/2018 15:17:00 |
|---|---|---|

Line spacing:    1.5 lines

| Page 34: [206] Formatted | 张 颖 | 21/11/2018 15:17:00 |
|---|---|---|

Font: 10.5 pt

| Page 34: [207] Formatted | 张 颖 | 21/11/2018 15:17:00 |
|---|---|---|

Font: (Default) Calibri, 10.5 pt, Font color: Red

| Page 34: [207] Formatted | 张 颖 | 21/11/2018 15:17:00 |
|---|---|---|

Font: (Default) Calibri, 10.5 pt, Font color: Red

| Page 34: [208] Formatted | 张 颖 | 21/11/2018 15:17:00 |
|---|---|---|

Font: (Default) Calibri, 10.5 pt, Font color: Red

| Page 34: [208] Formatted | 张 颖 | 21/11/2018 15:17:00 |
|---|---|---|

Font: (Default) Calibri, 10.5 pt, Font color: Red

| Page 34: [209] Formatted | 张 颖 | 21/11/2018 15:17:00 |
|---|---|---|

Font: (Default) Calibri, 10.5 pt, Font color: Red

| Page 34: [209] Formatted | 张 颖 | 21/11/2018 15:17:00 |
|---|---|---|

Font: (Default) Calibri, 10.5 pt, Font color: Red

| Page 34: [210] Formatted | Textcheck | 11/11/2018 21:52:00 |
|---|---|---|

Line spacing:    1.5 lines

| Page 34: [211] Formatted | 张 颖 | 21/11/2018 15:25:00 |
|---|---|---|

Font: (Default) Calibri, 10.5 pt

| Page 34: [212] Formatted | 张 颖 | 21/11/2018 15:17:00 |
|---|---|---|

Line spacing:    1.5 lines

| Page 34: [213] Formatted | 张 颖 | 21/11/2018 15:25:00 |
|---|---|---|

Font: 10.5 pt

| Page 34: [214] Formatted | 张 颖 | 21/11/2018 15:25:00 |
|---|---|---|

Font: (Default) Calibri, 10.5 pt

| Page 34: [214] Formatted | 张 颖 | 21/11/2018 15:25:00 |
|---|---|---|

Font: (Default) Calibri, 10.5 pt

| Page 34: [215] Formatted | 张 颖 | 21/11/2018 15:25:00 |
|---|---|---|

Font: (Default) Calibri, 10.5 pt

| Page 34: [215] Formatted | 张 颖 | 21/11/2018 15:25:00 |
|---|---|---|

Font: (Default) Calibri, 10.5 pt

| Page 34: [216] Formatted | 张 颖 | 21/11/2018 15:25:00 |
|---|---|---|

Font: (Default) Calibri, 10.5 pt

| Page 34: [216] Formatted | 张 颖 | 21/11/2018 15:25:00 |
|---|---|---|

Font: (Default) Calibri, 10.5 pt

| Page 34: [217] Formatted | Textcheck | 11/11/2018 21:52:00 |
|---|---|---|

Line spacing:    1.5 lines

| Page 34: [218] Formatted | 张 颖 | 21/11/2018 15:25:00 |
|---|---|---|

Font: (Default) Calibri, 10.5 pt

| Page 34: [219] Formatted | 张 颖 | 21/11/2018 15:17:00 |
|---|---|---|

Line spacing:    1.5 lines

| Page 34: [220] Formatted | 张 颖 | 21/11/2018 15:25:00 |
|---|---|---|

Font: 10.5 pt

| Page 34: [221] Formatted | 张 颖 | 21/11/2018 15:25:00 |
|---|---|---|

Font: (Default) Calibri, 10.5 pt

| Page 34: [221] Formatted | 张 颖 | 21/11/2018 15:25:00 |
|---|---|---|

Font: (Default) Calibri, 10.5 pt

| Page 34: [222] Formatted | 张 颖 | 21/11/2018 15:25:00 |
|---|---|---|

Font: (Default) Calibri, 10.5 pt

| Page 34: [222] Formatted | 张 颖 | 21/11/2018 15:25:00 |
|---|---|---|

Font: (Default) Calibri, 10.5 pt

| Page 34: [223] Formatted | 张 颖 | 21/11/2018 15:25:00 |
|---|---|---|

Font: (Default) Calibri, 10.5 pt

| Page 34: [223] Formatted | 张 颖 | 21/11/2018 15:25:00 |
|---|---|---|

Font: (Default) Calibri, 10.5 pt

| Page 34: [224] Formatted | Textcheck | 11/11/2018 21:52:00 |
|---|---|---|

Line spacing:    1.5 lines

| Page 34: [225] Formatted | 张 颖 | 21/11/2018 15:25:00 |
|---|---|---|

Font: (Default) Calibri, 10.5 pt

| Page 34: [226] Formatted | 张 颖 | 21/11/2018 15:17:00 |
|---|---|---|

Line spacing:    1.5 lines

| Page 34: [227] Formatted | 张 颖 | 21/11/2018 15:25:00 |
|---|---|---|

Font: 10.5 pt

| Page 34: [228] Formatted | 张 颖 | 21/11/2018 15:25:00 |
|---|---|---|

Font: (Default) Calibri, 10.5 pt

| Page 34: [228] Formatted | 张 颖 | 21/11/2018 15:25:00 |
|---|---|---|

Font: (Default) Calibri, 10.5 pt

| Page 34: [229] Formatted | 张 颖 | 21/11/2018 15:25:00 |
|---|---|---|

Font: (Default) Calibri, 10.5 pt

| Page 34: [229] Formatted | 张 颖 | 21/11/2018 15:25:00 |
|---|---|---|

Font: (Default) Calibri, 10.5 pt

| Page 34: [230] Formatted | 张 颖 | 21/11/2018 15:25:00 |
|---|---|---|

Font: (Default) Calibri, 10.5 pt

| Page 34: [230] Formatted | 张 颖 | 21/11/2018 15:25:00 |
|---|---|---|

Font: (Default) Calibri, 10.5 pt

| Page 34: [231] Formatted | Textcheck | 11/11/2018 21:52:00 |
|---|---|---|

Line spacing:    1.5 lines

| Page 34: [232] Formatted | 张 颖 | 21/11/2018 15:25:00 |
|---|---|---|

Font: (Default) Calibri, 10.5 pt

| Page 34: [233] Formatted | 张 颖 | 21/11/2018 15:17:00 |
|---|---|---|

Line spacing:    1.5 lines

| Page 34: [234] Formatted | 张 颖 | 21/11/2018 15:25:00 |
|---|---|---|

Font: 10.5 pt

| Page 34: [235] Formatted | 张 颖 | 21/11/2018 15:25:00 |
|---|---|---|

Font: (Default) Calibri, 10.5 pt

| Page 34: [235] Formatted | 张 颖 | 21/11/2018 15:25:00 |
|---|---|---|

Font: (Default) Calibri, 10.5 pt

| Page 34: [236] Formatted | 张 颖 | 21/11/2018 15:25:00 |
|---|---|---|

Font: (Default) Calibri, 10.5 pt

| Page 34: [236] Formatted | 张 颖 | 21/11/2018 15:25:00 |
|---|---|---|

Font: (Default) Calibri, 10.5 pt

| Page 34: [237] Formatted | 张 颖 | 21/11/2018 15:25:00 |
|---|---|---|

Font: (Default) Calibri, 10.5 pt

| Page 34: [237] Formatted | 张 颖 | 21/11/2018 15:25:00 |
|---|---|---|

Font: (Default) Calibri, 10.5 pt

| Page 34: [238] Formatted | Textcheck | 11/11/2018 21:52:00 |
|---|---|---|

Line spacing:    1.5 lines

| Page 34: [239] Formatted | 张 颖 | 20/11/2018 14:27:00 |
|---|---|---|

Font color: Red

| Page 34: [240] Formatted | 张 颖 | 20/11/2018 14:27:00 |
|---|---|---|

Font color: Red

| Page 34: [241] Formatted | 张 颖 | 20/11/2018 14:27:00 |
|---|---|---|

Font color: Red

| Page 34: [242] Formatted | 张 颖 | 20/11/2018 14:27:00 |
|---|---|---|

Font color: Red

| Page 34: [243] Formatted | 张 颖 | 21/11/2018 15:17:00 |
|---|---|---|

Font: (Default) Calibri, 10.5 pt, Font color: Red

| Page 34: [244] Formatted | 张 颖 | 21/11/2018 15:17:00 |
|---|---|---|

Line spacing:    1.5 lines

| Page 34: [245] Formatted | 张 颖 | 21/11/2018 15:17:00 |
|---|---|---|

Font: 10.5 pt

| Page 34: [245] Formatted | 张 颖 | 21/11/2018 15:17:00 |
|---|---|---|

Font: 10.5 pt

| Page 34: [246] Formatted | 张 颖 | 21/11/2018 15:17:00 |
|---|---|---|

Font: (Default) Calibri, 10.5 pt, Font color: Red

| Page 34: [246] Formatted | 张 颖 | 21/11/2018 15:17:00 |
|---|---|---|

Font: (Default) Calibri, 10.5 pt, Font color: Red

| Page 34: [246] Formatted | 张 颖 | 21/11/2018 15:17:00 |
|---|---|---|

Font: (Default) Calibri, 10.5 pt, Font color: Red

| Page 34: [247] Formatted | 张 颖 | 21/11/2018 15:17:00 |
|---|---|---|

Font: (Default) Calibri, 10.5 pt, Font color: Red

| Page 34: [247] Formatted | 张 颖 | 21/11/2018 15:17:00 |
|---|---|---|

Font: (Default) Calibri, 10.5 pt, Font color: Red

| Page 34: [247] Formatted | 张 颖 | 21/11/2018 15:17:00 |
|---|---|---|

Font: (Default) Calibri, 10.5 pt, Font color: Red

| Page 34: [248] Formatted | 张 颖 | 21/11/2018 15:17:00 |
|---|---|---|

Font: (Default) Calibri, 10.5 pt, Font color: Red

| Page 34: [248] Formatted | 张 颖 | 21/11/2018 15:17:00 |
|---|---|---|

Font: (Default) Calibri, 10.5 pt, Font color: Red

| Page 34: [248] Formatted | 张 颖 | 21/11/2018 15:17:00 |
|---|---|---|

Font: (Default) Calibri, 10.5 pt, Font color: Red

| Page 34: [249] Formatted | Textcheck | 11/11/2018 21:52:00 |
|---|---|---|

Line spacing:    1.5 lines

| Page 34: [250] Formatted | 张 颖 | 21/11/2018 15:25:00 |
|---|---|---|

Font: (Default) Calibri, 10.5 pt

| Page 34: [251] Formatted | 张 颖 | 21/11/2018 15:17:00 |
|---|---|---|

Line spacing:    1.5 lines

| Page 34: [252] Formatted | 张 颖 | 21/11/2018 15:25:00 |
|---|---|---|

Font: 10.5 pt

| Page 34: [253] Formatted | 张 颖 | 21/11/2018 15:25:00 |
|---|---|---|

Font: (Default) Calibri, 10.5 pt

| Page 34: [253] Formatted | 张 颖 | 21/11/2018 15:25:00 |
|---|---|---|

Font: (Default) Calibri, 10.5 pt

| Page 34: [254] Formatted | 张 颖 | 21/11/2018 15:25:00 |
|---|---|---|

Font: (Default) Calibri, 10.5 pt

| Page 34: [254] Formatted | 张 颖 | 21/11/2018 15:25:00 |
|---|---|---|

Font: (Default) Calibri, 10.5 pt

| Page 34: [255] Formatted | 张 颖 | 21/11/2018 15:25:00 |
|---|---|---|

Font: (Default) Calibri, 10.5 pt

| Page 34: [255] Formatted | 张 颖 | 21/11/2018 15:25:00 |
|---|---|---|

Font: (Default) Calibri, 10.5 pt

| Page 34: [256] Formatted | Textcheck | 11/11/2018 21:52:00 |
|---|---|---|

Line spacing:    1.5 lines

| Page 34: [257] Formatted | 张 颖 | 21/11/2018 15:25:00 |
|---|---|---|

| Page 34: [258] Formatted | 张 颖 | 21/11/2018 15:17:00 |
|---|---|---|

Line spacing:    1.5 lines

| Page 34: [259] Formatted | 张 颖 | 21/11/2018 15:25:00 |
|---|---|---|

Font: 10.5 pt

| Page 34: [260] Formatted | 张 颖 | 21/11/2018 15:25:00 |
|---|---|---|

Font: (Default) Calibri, 10.5 pt

| Page 34: [260] Formatted | 张 颖 | 21/11/2018 15:25:00 |
|---|---|---|

Font: (Default) Calibri, 10.5 pt

| Page 34: [261] Formatted | 张 颖 | 21/11/2018 15:25:00 |
|---|---|---|

Font: (Default) Calibri, 10.5 pt

| Page 34: [261] Formatted | 张 颖 | 21/11/2018 15:25:00 |
|---|---|---|

Font: (Default) Calibri, 10.5 pt

| Page 34: [262] Formatted | 张 颖 | 21/11/2018 15:25:00 |
|---|---|---|

Font: (Default) Calibri, 10.5 pt

| Page 34: [262] Formatted | 张 颖 | 21/11/2018 15:25:00 |
|---|---|---|

Font: (Default) Calibri, 10.5 pt

| Page 34: [263] Formatted | Textcheck | 11/11/2018 21:52:00 |
|---|---|---|

Line spacing:    1.5 lines

| Page 34: [264] Formatted | 张 颖 | 21/11/2018 15:25:00 |
|---|---|---|

Font: (Default) Calibri, 10.5 pt

| Page 34: [265] Formatted | 张 颖 | 21/11/2018 15:17:00 |
|---|---|---|

Line spacing:    1.5 lines

| Page 34: [266] Formatted | 张 颖 | 21/11/2018 15:25:00 |
|---|---|---|

Font: 10.5 pt

| Page 34: [267] Formatted | 张 颖 | 21/11/2018 15:25:00 |
|---|---|---|

Font: (Default) Calibri, 10.5 pt

| Page 34: [267] Formatted | 张 颖 | 21/11/2018 15:25:00 |
|---|---|---|

Font: (Default) Calibri, 10.5 pt

| Page 34: [268] Formatted | 张 颖 | 21/11/2018 15:25:00 |
|---|---|---|

Font: (Default) Calibri, 10.5 pt

| Page 34: [268] Formatted | 张 颖 | 21/11/2018 15:25:00 |
|---|---|---|

Font: (Default) Calibri, 10.5 pt

| Page 34: [269] Formatted | 张 颖 | 21/11/2018 15:25:00 |
|---|---|---|

Font: (Default) Calibri, 10.5 pt

| Page 34: [269] Formatted | 张 颖 | 21/11/2018 15:25:00 |
|---|---|---|

Font: (Default) Calibri, 10.5 pt

| Page 34: [270] Formatted | Textcheck | 11/11/2018 21:52:00 |
|---|---|---|

Line spacing:    1.5 lines

| Page 34: [271] Formatted | 张 颖 | 21/11/2018 15:25:00 |
|---|---|---|

Font: (Default) Calibri, 10.5 pt

| Page 34: [272] Formatted | 张 颖 | 21/11/2018 15:17:00 |
|---|---|---|

Line spacing:    1.5 lines

| Page 34: [273] Formatted | 张 颖 | 21/11/2018 15:25:00 |
|---|---|---|

Font: 10.5 pt

| Page 34: [274] Formatted | 张 颖 | 21/11/2018 15:25:00 |
|---|---|---|

Font: (Default) Calibri, 10.5 pt

| Page 34: [274] Formatted | 张 颖 | 21/11/2018 15:25:00 |
|---|---|---|

Font: (Default) Calibri, 10.5 pt

| Page 34: [275] Formatted | 张 颖 | 21/11/2018 15:25:00 |
|---|---|---|

Font: (Default) Calibri, 10.5 pt

| Page 34: [275] Formatted | 张 颖 | 21/11/2018 15:25:00 |
|---|---|---|

Font: (Default) Calibri, 10.5 pt

| Page 34: [276] Formatted | 张 颖 | 21/11/2018 15:25:00 |
|---|---|---|

Font: (Default) Calibri, 10.5 pt

| Page 34: [276] Formatted | 张 颖 | 21/11/2018 15:25:00 |
|---|---|---|

Font: (Default) Calibri, 10.5 pt

| Page 34: [277] Formatted | Textcheck | 11/11/2018 21:52:00 |
|---|---|---|

Line spacing:    1.5 lines

| Page 34: [278] Formatted | 张 颖 | 20/11/2018 14:28:00 |
|---|---|---|

Font color: Red

| Page 34: [279] Formatted | 张 颖 | 20/11/2018 14:28:00 |
|---|---|---|

Font color: Red

| Page 34: [280] Formatted | 张 颖 | 20/11/2018 14:28:00 |
|---|---|---|

Font color: Red

| Page 34: [281] Formatted | 张 颖 | 20/11/2018 14:28:00 |
|---|---|---|

Font color: Red

| Page 34: [282] Formatted | 张 颖 | 21/11/2018 15:17:00 |
|---|---|---|

Font: (Default) Calibri, 10.5 pt, Font color: Red

| Page 34: [283] Formatted | 张 颖 | 21/11/2018 15:17:00 |
|---|---|---|

Line spacing:    1.5 lines

| Page 34: [284] Formatted | 张 颖 | 21/11/2018 15:17:00 |
|---|---|---|

Font: 10.5 pt

| Page 34: [285] Formatted | 张 颖 | 21/11/2018 15:17:00 |
|---|---|---|

Font: (Default) Calibri, 10.5 pt, Font color: Red

| Page 34: [286] Formatted | 张 颖 | 21/11/2018 15:17:00 |
|---|---|---|

Font: (Default) Calibri, 10.5 pt, Font color: Red

| Page 34: [287] Formatted | 张 颖 | 21/11/2018 15:17:00 |
|---|---|---|

Font: (Default) Calibri, 10.5 pt, Font color: Red

| Page 35: [288] Formatted | 张 颖 | 21/11/2018 15:17:00 |
|---|---|---|

Font: 10.5 pt

| Page 35: [289] Formatted | 张 颖 | 21/11/2018 15:17:00 |
|---|---|---|

Font: 10.5 pt

| Page 35: [290] Formatted | 张 颖 | 21/11/2018 15:17:00 |
|---|---|---|

Font: 10.5 pt

| Page 35: [291] Formatted | Textcheck | 11/11/2018 21:52:00 |
|---|---|---|

Line spacing: 1.5 lines

| Page 35: [292] Formatted | 张 颖 | 21/11/2018 15:25:00 |
|---|---|---|

Font: (Default) Calibri, 10.5 pt

| Page 35: [293] Formatted | 张 颖 | 21/11/2018 15:17:00 |
|---|---|---|

Line spacing: 1.5 lines

| Page 35: [294] Formatted | 张 颖 | 21/11/2018 15:25:00 |
|---|---|---|

Font: 10.5 pt

| Page 35: [295] Formatted | 张 颖 | 21/11/2018 15:25:00 |
|---|---|---|

Font: (Default) Calibri, 10.5 pt

| Page 35: [295] Formatted | 张 颖 | 21/11/2018 15:25:00 |
|---|---|---|

Font: (Default) Calibri, 10.5 pt

| Page 35: [296] Formatted | 张 颖 | 21/11/2018 15:25:00 |
|---|---|---|

Font: (Default) Calibri, 10.5 pt

| Page 35: [296] Formatted | 张 颖 | 21/11/2018 15:25:00 |
|---|---|---|

Font: (Default) Calibri, 10.5 pt

| Page 35: [297] Formatted | 张 颖 | 21/11/2018 15:25:00 |
|---|---|---|

Font: (Default) Calibri, 10.5 pt

| Page 35: [297] Formatted | 张 颖 | 21/11/2018 15:25:00 |
|---|---|---|

Font: (Default) Calibri, 10.5 pt

| Page 35: [298] Formatted | Textcheck | 11/11/2018 21:52:00 |
|---|---|---|

Line spacing: 1.5 lines

| Page 35: [299] Formatted | 张 颖 | 21/11/2018 15:25:00 |
|---|---|---|

Font: (Default) Calibri, 10.5 pt

| Page 35: [300] Formatted | 张 颖 | 21/11/2018 15:17:00 |
|---|---|---|

Line spacing: 1.5 lines

| Page 35: [301] Formatted | 张 颖 | 21/11/2018 15:25:00 |
|---|---|---|

Font: 10.5 pt

| Page 35: [302] Formatted | 张 颖 | 21/11/2018 15:25:00 |
|---|---|---|

Font: (Default) Calibri, 10.5 pt

| Page 35: [302] Formatted | 张 颖 | 21/11/2018 15:25:00 |
|---|---|---|

Font: (Default) Calibri, 10.5 pt

| Page 35: [303] Formatted | 张 颖 | 21/11/2018 15:25:00 |
|---|---|---|

Font: (Default) Calibri, 10.5 pt

| Page 35: [303] Formatted | 张 颖 | 21/11/2018 15:25:00 |
|---|---|---|

Font: (Default) Calibri, 10.5 pt

| Page 35: [304] Formatted | 张 颖 | 21/11/2018 15:25:00 |
|---|---|---|

Font: (Default) Calibri, 10.5 pt

| Page 35: [304] Formatted | 张 颖 | 21/11/2018 15:25:00 |
|---|---|---|

Font: (Default) Calibri, 10.5 pt

| Page 35: [305] Formatted | Textcheck | 11/11/2018 21:52:00 |
|---|---|---|

Line spacing:    1.5 lines

| Page 35: [306] Formatted | 张 颖 | 21/11/2018 15:25:00 |
|---|---|---|

Font: (Default) Calibri, 10.5 pt

| Page 35: [307] Formatted | 张 颖 | 21/11/2018 15:17:00 |
|---|---|---|

Line spacing:    1.5 lines

| Page 35: [308] Formatted | 张 颖 | 21/11/2018 15:25:00 |
|---|---|---|

Font: 10.5 pt

| Page 35: [309] Formatted | 张 颖 | 21/11/2018 15:25:00 |
|---|---|---|

Font: (Default) Calibri, 10.5 pt

| Page 35: [309] Formatted | 张 颖 | 21/11/2018 15:25:00 |
|---|---|---|

Font: (Default) Calibri, 10.5 pt

| Page 35: [310] Formatted | 张 颖 | 21/11/2018 15:25:00 |
|---|---|---|

Font: (Default) Calibri, 10.5 pt

| Page 35: [310] Formatted | 张 颖 | 21/11/2018 15:25:00 |
|---|---|---|

Font: (Default) Calibri, 10.5 pt

| Page 35: [311] Formatted | 张 颖 | 21/11/2018 15:25:00 |
|---|---|---|

Font: (Default) Calibri, 10.5 pt

| Page 35: [311] Formatted | 张 颖 | 21/11/2018 15:25:00 |
|---|---|---|

Font: (Default) Calibri, 10.5 pt

| Page 35: [312] Formatted | Textcheck | 11/11/2018 21:52:00 |
|---|---|---|

Line spacing:    1.5 lines

| Page 35: [313] Formatted | 张 颖 | 21/11/2018 15:25:00 |
|---|---|---|

Font: (Default) Calibri, 10.5 pt

| Page 35: [314] Formatted | 张 颖 | 21/11/2018 15:17:00 |
|---|---|---|

Line spacing:    1.5 lines

| Page 35: [315] Formatted | 张 颖 | 21/11/2018 15:25:00 |
|---|---|---|

Font: 10.5 pt

| Page 35: [316] Formatted | 张 颖 | 21/11/2018 15:25:00 |
|---|---|---|

Font: (Default) Calibri, 10.5 pt

| Page 35: [316] Formatted | 张 颖 | 21/11/2018 15:25:00 |
|---|---|---|

Font: (Default) Calibri, 10.5 pt

| Page 35: [317] Formatted | 张 颖 | 21/11/2018 15:25:00 |
|---|---|---|

Font: (Default) Calibri, 10.5 pt

| Page 35: [317] Formatted | 张 颖 | 21/11/2018 15:25:00 |
|---|---|---|

Font: (Default) Calibri, 10.5 pt

| Page 35: [318] Formatted | 张 颖 | 21/11/2018 15:25:00 |
|---|---|---|

Font: (Default) Calibri, 10.5 pt

| Page 35: [318] Formatted | 张 颖 | 21/11/2018 15:25:00 |
|---|---|---|

Font: (Default) Calibri, 10.5 pt

| Page 35: [319] Formatted | Textcheck | 11/11/2018 21:52:00 |
|---|---|---|

Line spacing:    1.5 lines

| Page 35: [320] Formatted | 张 颖 | 20/11/2018 14:28:00 |
|---|---|---|

Font color: Red

| Page 35: [321] Formatted | 张 颖 | 20/11/2018 14:28:00 |
|---|---|---|

Font color: Red

| Page 35: [322] Formatted | 张 颖 | 20/11/2018 14:28:00 |
|---|---|---|

Font color: Red

| Page 35: [323] Formatted | 张 颖 | 20/11/2018 14:28:00 |
|---|---|---|

Font color: Red

| Page 35: [324] Formatted | 张 颖 | 21/11/2018 15:17:00 |
|---|---|---|

Font: (Default) Calibri, 10.5 pt, Font color: Red

| Page 35: [325] Formatted | 张 颖 | 21/11/2018 15:17:00 |
|---|---|---|

Line spacing:    1.5 lines

| Page 35: [326] Formatted | 张 颖 | 21/11/2018 15:17:00 |
|---|---|---|

Font: 10.5 pt

| Page 35: [326] Formatted | 张 颖 | 21/11/2018 15:17:00 |
|---|---|---|

Font: 10.5 pt

| Page 35: [327] Formatted | 张 颖 | 21/11/2018 15:17:00 |
|---|---|---|

Font: (Default) Calibri, 10.5 pt, Font color: Red

| Page 35: [327] Formatted | 张 颖 | 21/11/2018 15:17:00 |
|---|---|---|

Font: (Default) Calibri, 10.5 pt, Font color: Red

| Page 35: [327] Formatted | 张 颖 | 21/11/2018 15:17:00 |
|---|---|---|

Font: (Default) Calibri, 10.5 pt, Font color: Red

| Page 35: [328] Formatted | 张 颖 | 21/11/2018 15:17:00 |
|---|---|---|

Font: (Default) Calibri, 10.5 pt, Font color: Red

| Page 35: [328] Formatted | 张 颖 | 21/11/2018 15:17:00 |
|---|---|---|

Font: (Default) Calibri, 10.5 pt, Font color: Red

| Page 35: [328] Formatted | 张 颖 | 21/11/2018 15:17:00 |
|---|---|---|

Font: (Default) Calibri, 10.5 pt, Font color: Red

| Page 35: [329] Formatted | 张 颖 | 21/11/2018 15:17:00 |
|---|---|---|

Font: (Default) Calibri, 10.5 pt, Font color: Red

| Page 35: [329] Formatted | 张 颖 | 21/11/2018 15:17:00 |
|---|---|---|

Font: (Default) Calibri, 10.5 pt, Font color: Red

| Page 35: [329] Formatted | 张 颖 | 21/11/2018 15:17:00 |
|---|---|---|

Font: (Default) Calibri, 10.5 pt, Font color: Red

| Page 35: [330] Formatted | Textcheck | 11/11/2018 21:52:00 |
|---|---|---|

Line spacing:    1.5 lines

| Page 35: [331] Formatted | 张 颖 | 21/11/2018 15:25:00 |
|---|---|---|

Font: (Default) Calibri, 10.5 pt

| Page 35: [332] Formatted | 张 颖 | 21/11/2018 15:17:00 |
|---|---|---|

Line spacing:    1.5 lines

| Page 35: [333] Formatted | 张 颖 | 21/11/2018 15:25:00 |
|---|---|---|

Font: 10.5 pt

| Page 35: [334] Formatted | 张 颖 | 21/11/2018 15:25:00 |
|---|---|---|

Font: (Default) Calibri, 10.5 pt

| Page 35: [334] Formatted | 张 颖 | 21/11/2018 15:25:00 |
|---|---|---|

Font: (Default) Calibri, 10.5 pt

| Page 35: [335] Formatted | 张 颖 | 21/11/2018 15:25:00 |
|---|---|---|

Font: (Default) Calibri, 10.5 pt

| Page 35: [335] Formatted | 张 颖 | 21/11/2018 15:25:00 |
|---|---|---|

Font: (Default) Calibri, 10.5 pt

| Page 35: [336] Formatted | 张 颖 | 21/11/2018 15:25:00 |
|---|---|---|

Font: (Default) Calibri, 10.5 pt

| Page 35: [336] Formatted | 张 颖 | 21/11/2018 15:25:00 |
|---|---|---|

Font: (Default) Calibri, 10.5 pt

| Page 35: [337] Formatted | Textcheck | 11/11/2018 21:52:00 |
|---|---|---|

Line spacing:    1.5 lines

| Page 35: [338] Formatted | 张 颖 | 21/11/2018 15:25:00 |
|---|---|---|

Font: (Default) Calibri, 10.5 pt

| Page 35: [339] Formatted | 张 颖 | 21/11/2018 15:17:00 |
|---|---|---|

Line spacing:    1.5 lines

| Page 35: [340] Formatted | 张 颖 | 21/11/2018 15:25:00 |
|---|---|---|

Font: 10.5 pt

| Page 35: [341] Formatted | 张 颖 | 21/11/2018 15:25:00 |
|---|---|---|

Font: (Default) Calibri, 10.5 pt

| Page 35: [341] Formatted | 张 颖 | 21/11/2018 15:25:00 |
|---|---|---|

Font: (Default) Calibri, 10.5 pt

| Page 35: [342] Formatted | 张 颖 | 21/11/2018 15:25:00 |
|---|---|---|

Font: (Default) Calibri, 10.5 pt

| Page 35: [342] Formatted | 张 颖 | 21/11/2018 15:25:00 |
|---|---|---|

Font: (Default) Calibri, 10.5 pt

| Page 35: [343] Formatted | 张 颖 | 21/11/2018 15:25:00 |
|---|---|---|

Font: (Default) Calibri, 10.5 pt

| Page 35: [343] Formatted | 张 颖 | 21/11/2018 15:25:00 |
|---|---|---|

Font: (Default) Calibri, 10.5 pt

| Page 35: [344] Formatted | Textcheck | 11/11/2018 21:52:00 |
|---|---|---|

Line spacing:    1.5 lines

| Page 35: [345] Formatted | 张 颖 | 21/11/2018 15:25:00 |
|---|---|---|

Font: (Default) Calibri, 10.5 pt

| Page 35: [346] Formatted | 张 颖 | 21/11/2018 15:17:00 |
|---|---|---|

Line spacing:    1.5 lines

| Page 35: [347] Formatted | 张 颖 | 21/11/2018 15:25:00 |
|---|---|---|

Font: 10.5 pt

| Page 35: [348] Formatted | 张 颖 | 21/11/2018 15:25:00 |
|---|---|---|

Font: (Default) Calibri, 10.5 pt

| Page 35: [348] Formatted | 张 颖 | 21/11/2018 15:25:00 |
|---|---|---|

Font: (Default) Calibri, 10.5 pt

| Page 35: [349] Formatted | 张 颖 | 21/11/2018 15:25:00 |
|---|---|---|

Font: (Default) Calibri, 10.5 pt

| Page 35: [349] Formatted | 张 颖 | 21/11/2018 15:25:00 |
|---|---|---|

Font: (Default) Calibri, 10.5 pt

| Page 35: [350] Formatted | 张 颖 | 21/11/2018 15:25:00 |
|---|---|---|

Font: (Default) Calibri, 10.5 pt

| Page 35: [350] Formatted | 张 颖 | 21/11/2018 15:25:00 |
|---|---|---|

Font: (Default) Calibri, 10.5 pt

| Page 35: [351] Formatted | Textcheck | 11/11/2018 21:52:00 |
|---|---|---|

Line spacing:    1.5 lines

| Page 35: [352] Formatted | 张 颖 | 21/11/2018 15:25:00 |
|---|---|---|

Font: (Default) Calibri, 10.5 pt

| Page 35: [353] Formatted | 张 颖 | 21/11/2018 15:17:00 |
|---|---|---|

Line spacing:    1.5 lines

| Page 35: [354] Formatted | 张 颖 | 21/11/2018 15:25:00 |
|---|---|---|

Font: 10.5 pt

| Page 35: [355] Formatted | 张 颖 | 21/11/2018 15:25:00 |
|---|---|---|

Font: (Default) Calibri, 10.5 pt

| Page 35: [355] Formatted | 张 颖 | 21/11/2018 15:25:00 |
|---|---|---|

Font: (Default) Calibri, 10.5 pt

| Page 35: [356] Formatted | 张 颖 | 21/11/2018 15:25:00 |
|---|---|---|

Font: (Default) Calibri, 10.5 pt

| Page 35: [356] Formatted | 张 颖 | 21/11/2018 15:25:00 |
|---|---|---|

Font: (Default) Calibri, 10.5 pt

| Page 35: [357] Formatted | 张 颖 | 21/11/2018 15:25:00 |
|---|---|---|

Font: (Default) Calibri, 10.5 pt

| Page 35: [357] Formatted | 张 颖 | 21/11/2018 15:25:00 |
|---|---|---|

Font: (Default) Calibri, 10.5 pt

| Page 35: [358] Formatted | Textcheck | 11/11/2018 21:52:00 |
|---|---|---|

Line spacing:    1.5 lines

| Page 35: [359] Formatted | 张 颖 | 20/11/2018 14:28:00 |
|---|---|---|

Font color: Red

| Page 35: [360] Formatted | 张 颖 | 20/11/2018 14:28:00 |
|---|---|---|

Font color: Red

| Page 35: [361] Formatted | 张 颖 | 20/11/2018 14:28:00 |
|---|---|---|

Font color: Red

| Page 35: [362] Formatted | 张 颖 | 20/11/2018 14:28:00 |
|---|---|---|

Font color: Red

| Page 35: [363] Formatted | 张 颖 | 21/11/2018 15:17:00 |
|---|---|---|

Font: (Default) Calibri, 10.5 pt, Font color: Red

| Page 35: [364] Formatted | 张 颖 | 21/11/2018 15:17:00 |
|---|---|---|

Line spacing:    1.5 lines

| Page 35: [365] Formatted | 张 颖 | 21/11/2018 15:17:00 |
|---|---|---|

Font: 10.5 pt

| Page 35: [366] Formatted | 张 颖 | 21/11/2018 15:17:00 |
|---|---|---|

Font: (Default) Calibri, 10.5 pt, Font color: Red

| Page 35: [366] Formatted | 张 颖 | 21/11/2018 15:17:00 |
|---|---|---|

Font: (Default) Calibri, 10.5 pt, Font color: Red

| Page 35: [367] Formatted | 张 颖 | 21/11/2018 15:17:00 |
|---|---|---|

Font: (Default) Calibri, 10.5 pt, Font color: Red

| Page 35: [367] Formatted | 张 颖 | 21/11/2018 15:17:00 |
|---|---|---|

Font: (Default) Calibri, 10.5 pt, Font color: Red

| Page 35: [368] Formatted | 张 颖 | 21/11/2018 15:17:00 |
|---|---|---|

Font: (Default) Calibri, 10.5 pt, Font color: Red

| Page 35: [368] Formatted | 张 颖 | 21/11/2018 15:17:00 |
|---|---|---|

Font: (Default) Calibri, 10.5 pt, Font color: Red

| Page 35: [369] Formatted | Textcheck | 11/11/2018 21:52:00 |
|---|---|---|

Line spacing:    1.5 lines

| Page 35: [370] Formatted | 张 颖 | 21/11/2018 15:26:00 |
|---|---|---|

Font: (Default) Calibri, 10.5 pt

| Page 35: [371] Formatted | 张 颖 | 21/11/2018 15:17:00 |
|---|---|---|

Line spacing:    1.5 lines

| Page 35: [372] Formatted | 张 颖 | 21/11/2018 15:26:00 |
|---|---|---|

Font: 10.5 pt

| Page 35: [373] Formatted | 张 颖 | 21/11/2018 15:26:00 |
|---|---|---|

Font: (Default) Calibri, 10.5 pt

| Page 35: [373] Formatted | 张 颖 | 21/11/2018 15:26:00 |
|---|---|---|

Font: (Default) Calibri, 10.5 pt

| Page 35: [374] Formatted | 张 颖 | 21/11/2018 15:26:00 |
|---|---|---|

Font: (Default) Calibri, 10.5 pt

| Page 35: [374] Formatted | 张 颖 | 21/11/2018 15:26:00 |
|---|---|---|

Font: (Default) Calibri, 10.5 pt

| Page 35: [375] Formatted | 张 颖 | 21/11/2018 15:26:00 |
|---|---|---|

Font: (Default) Calibri, 10.5 pt

| Page 35: [375] Formatted | 张 颖 | 21/11/2018 15:26:00 |
|---|---|---|

Font: (Default) Calibri, 10.5 pt

| Page 35: [376] Formatted | Textcheck | 11/11/2018 21:52:00 |
|---|---|---|

Line spacing:    1.5 lines

| Page 35: [377] Formatted | 张 颖 | 21/11/2018 15:26:00 |
|---|---|---|

Font: (Default) Calibri, 10.5 pt

| Page 35: [378] Formatted | 张 颖 | 21/11/2018 15:17:00 |
|---|---|---|

Line spacing:    1.5 lines

| Page 35: [379] Formatted | 张 颖 | 21/11/2018 15:26:00 |
|---|---|---|

Font: 10.5 pt

| Page 35: [380] Formatted | 张 颖 | 21/11/2018 15:26:00 |
|---|---|---|

Font: (Default) Calibri, 10.5 pt

| Page 35: [380] Formatted | 张 颖 | 21/11/2018 15:26:00 |
|---|---|---|

Font: (Default) Calibri, 10.5 pt

| Page 35: [381] Formatted | 张 颖 | 21/11/2018 15:26:00 |
|---|---|---|

Font: (Default) Calibri, 10.5 pt

| Page 35: [381] Formatted | 张 颖 | 21/11/2018 15:26:00 |
|---|---|---|

Font: (Default) Calibri, 10.5 pt

| Page 35: [382] Formatted | 张 颖 | 21/11/2018 15:26:00 |
|---|---|---|

Font: (Default) Calibri, 10.5 pt

| Page 35: [382] Formatted | 张 颖 | 21/11/2018 15:26:00 |
|---|---|---|

Font: (Default) Calibri, 10.5 pt

| Page 35: [383] Formatted | Textcheck | 11/11/2018 21:52:00 |
|---|---|---|

Line spacing:    1.5 lines

| Page 35: [384] Formatted | 张 颖 | 21/11/2018 15:26:00 |
|---|---|---|

Font: (Default) Calibri, 10.5 pt

| Page 35: [385] Formatted | 张 颖 | 21/11/2018 15:17:00 |
|---|---|---|

Line spacing:    1.5 lines

| Page 35: [386] Formatted | 张 颖 | 21/11/2018 15:26:00 |
|---|---|---|

Font: 10.5 pt

| Page 35: [387] Formatted | 张 颖 | 21/11/2018 15:26:00 |
|---|---|---|

Font: (Default) Calibri, 10.5 pt

| Page 35: [387] Formatted | 张 颖 | 21/11/2018 15:26:00 |
|---|---|---|

Font: (Default) Calibri, 10.5 pt

| Page 35: [388] Formatted | 张 颖 | 21/11/2018 15:26:00 |
|---|---|---|

Font: (Default) Calibri, 10.5 pt

| Page 35: [388] Formatted | 张 颖 | 21/11/2018 15:26:00 |
|---|---|---|

Font: (Default) Calibri, 10.5 pt

| Page 35: [389] Formatted | 张 颖 | 21/11/2018 15:26:00 |
|---|---|---|

Font: (Default) Calibri, 10.5 pt

| Page 35: [389] Formatted | 张 颖 | 21/11/2018 15:26:00 |
|---|---|---|

Font: (Default) Calibri, 10.5 pt

| Page 35: [390] Formatted | Textcheck | 11/11/2018 21:52:00 |
|---|---|---|

Line spacing:    1.5 lines

| Page 35: [391] Formatted | 张 颖 | 21/11/2018 15:26:00 |
|---|---|---|

Font: (Default) Calibri, 10.5 pt

| Page 35: [392] Formatted | 张 颖 | 21/11/2018 15:17:00 |
|---|---|---|

Line spacing:    1.5 lines

| Page 35: [393] Formatted | 张 颖 | 21/11/2018 15:26:00 |
|---|---|---|

Font: 10.5 pt

| Page 35: [394] Formatted | 张 颖 | 21/11/2018 15:26:00 |
|---|---|---|

Font: (Default) Calibri, 10.5 pt

| Page 35: [394] Formatted | 张 颖 | 21/11/2018 15:26:00 |
|---|---|---|

Font: (Default) Calibri, 10.5 pt

| Page 35: [395] Formatted | 张 颖 | 21/11/2018 15:26:00 |
|---|---|---|

Font: (Default) Calibri, 10.5 pt

| Page 35: [395] Formatted | 张 颖 | 21/11/2018 15:26:00 |
|---|---|---|

Font: (Default) Calibri, 10.5 pt

| Page 35: [396] Formatted | 张 颖 | 21/11/2018 15:26:00 |
|---|---|---|

Font: (Default) Calibri, 10.5 pt

| Page 35: [396] Formatted | 张 颖 | 21/11/2018 15:26:00 |
|---|---|---|

Font: (Default) Calibri, 10.5 pt